# Marginal Density Ratio for Off-Policy Evaluation in Contextual Bandits

**Muhammad Faaiz Taufiq**[*]
Department of Statistics
University of Oxford

**Arnaud Doucet**
Department of Statistics
University of Oxford

**Rob Cornish**
Department of Statistics
University of Oxford

**Jean-François Ton**
ByteDance Research
ByteDance

## Abstract

Off-Policy Evaluation (OPE) in contextual bandits is crucial for assessing new policies using existing data without costly experimentation. However, current OPE methods, such as Inverse Probability Weighting (IPW) and Doubly Robust (DR) estimators, suffer from high variance, particularly in cases of low overlap between target and behavior policies or large action and context spaces. In this paper, we introduce a new OPE estimator for contextual bandits, the Marginal Ratio (MR) estimator, which focuses on the shift in the marginal distribution of outcomes $Y$ instead of the policies themselves. Through rigorous theoretical analysis, we demonstrate the benefits of the MR estimator compared to conventional methods like IPW and DR in terms of variance reduction. Additionally, we establish a connection between the MR estimator and the state-of-the-art Marginalized Inverse Propensity Score (MIPS) estimator, proving that MR achieves lower variance among a generalized family of MIPS estimators. We further illustrate the utility of the MR estimator in causal inference settings, where it exhibits enhanced performance in estimating Average Treatment Effects (ATE). Our experiments on synthetic and real-world datasets corroborate our theoretical findings and highlight the practical advantages of the MR estimator in OPE for contextual bandits.

## 1 Introduction

In contextual bandits, the objective is to select an action $A$, guided by contextual information $X$, to maximize the resulting outcome $Y$. This paradigm is prevalent in many real-world applications such as healthcare, personalized recommendation systems, or online advertising [1–3]. The objective is to perform actions, such as prescribing medication or recommending items, which lead to desired outcomes like improved patient health or higher click-through rates. Nonetheless, updating the policy presents challenges, as naïvely implementing a new, untested policy may raise ethical or financial concerns. For instance, prescribing a drug based on a new policy poses risks, as it may result in unexpected side effects. As a result, recent research [4–11] has concentrated on evaluating the performance of new policies (target policy) using only existing data that was generated using the current policy (behaviour policy). This problem is known as Off-Policy Evaluation (OPE).

Current OPE methods in contextual bandits, such as the Inverse Probability Weighting (IPW) [12] and Doubly Robust (DR) [13] estimators primarily account for the policy shift by re-weighting the

---

[*]Part of the work was done during an internship at ByteDance Research.
Correspondence to `muhammad.taufiq@stats.ox.ac.uk` and `jeanfrancois@bytedance.com`

37th Conference on Neural Information Processing Systems (NeurIPS 2023).

data using the ratio of the target and behaviour polices to estimate the target policy value. This can be problematic as it may lead to high variance in the estimators in cases of substantial policy shifts. The issue is further exacerbated in situations with large action or context spaces [14], since in these cases the estimation of policy ratios is even more difficult leading to extreme bias and variance.

In this work we show that this problem of high variance in OPE can be alleviated by using methods which directly consider the shift in the marginal distribution of the outcome $Y$ resulting from the policy shift, instead of considering the policy shift itself (as in IPW and DR). To this end, we propose a new OPE estimator for contextual bandits called the Marginal Ratio (MR) estimator, which weights the data directly based on the shift in the marginal distribution of outcomes $Y$ and consequently is much more robust to increasing sizes of action and context spaces than existing methods like IPW or DR. Our extensive theoretical analyses show that MR enjoys better variance properties than the existing methods making it highly attractive for a variety of applications in addition to OPE. One such application is the estimation of Average Treatment Effect (ATE) in causal inference, for which we show that MR provides greater sample efficiency than the most commonly used methods.

Our contributions in this paper are as follows:

- Firstly, we introduce MR, an OPE estimator for contextual bandits, that focuses on the shift in the marginal distribution of $Y$ rather than the joint distribution of $(X, A, Y)$. We show that MR has favourable theoretical properties compared to existing methods like IPW and DR. Our analysis also encompasses theory on the approximation errors of our estimator.
- Secondly, we explicitly lay out the connection between MR and Marginalized Inverse Propensity Score (MIPS) [14], a recent state-of-the-art contextual bandits OPE method, and prove that MR attains lowest variance among a generalized family of MIPS estimators.
- Thirdly, we show that the MR estimator can be applied in the setting of causal inference to estimate average treatment effects (ATE), and theoretically prove that the resulting estimator is more data-efficient with higher accuracy and lower variance than commonly used methods.
- Finally, we verify all our theoretical analyses through a variety of experiments on synthetic and real-world datasets and empirically demonstrate that the MR estimator achieves better overall performance compared to current state-of-the-art methods.

## 2 Background

### 2.1 Setup and Notation

We consider the standard contextual bandit setting. Let $X \in \mathcal{X}$ be a context vector (e.g., user features), $A \in \mathcal{A}$ denote an action (e.g., recommended website to the user), and $Y \in \mathcal{Y}$ denote a scalar reward or outcome (e.g., whether the user clicks on the website). The outcome and context are sampled from unknown probability distributions $p(y \mid x, a)$ and $p(x)$ respectively. Let $\mathcal{D} := \{(x_i, a_i, y_i)\}_{i=1}^n$ be a historically logged dataset with $n$ observations, generated by a (possibly unknown) *behaviour policy* $\pi^b(a \mid x)$. Specifically, $\mathcal{D}$ consists of i.i.d. samples from the joint density under *behaviour policy*,

$$p_{\pi^b}(x, a, y) := p(y \mid x, a)\, \pi^b(a \mid x)\, p(x). \tag{1}$$

We denote the joint density of $(X, A, Y)$ under the *target policy* as

$$p_{\pi^*}(x, a, y) := p(y \mid x, a)\, \pi^*(a \mid x)\, p(x). \tag{2}$$

Moreover, we use $p_{\pi^b}(y)$ to denote the marginal density of $Y$ under the behaviour policy,

$$p_{\pi^b}(y) = \int_{\mathcal{A} \times \mathcal{X}} p_{\pi^b}(x, a, y)\, \mathrm{d}a\, \mathrm{d}x,$$

and likewise for the target policy $\pi^*$. Similarly, we use $\mathbb{E}_{\pi^b}$ and $\mathbb{E}_{\pi^*}$ to denote the expectations under the joint densities $p_{\pi^b}(x, a, y)$ and $p_{\pi^*}(x, a, y)$ respectively.

**Off-policy evaluation (OPE)** The main objective of OPE is to estimate the expectation of the outcome $Y$ under a given target policy $\pi^*$, i.e., $\mathbb{E}_{\pi^*}[Y]$, using only the logged data $\mathcal{D}$.

Throughout this work, we assume that the support of the target policy $\pi^*$ is included in the support of the behaviour policy $\pi^b$. This is to ensure that importance sampling yields unbiased off-policy estimators, and is satisfied for exploratory behaviour policies such as the $\epsilon$-greedy policies.

**Assumption 2.1** (Support). For any $x \in \mathcal{X}, a \in \mathcal{A}, \pi^*(a \mid x) > 0 \implies \pi^b(a \mid x) > 0$.

## 2.2 Existing off-policy evaluation methodologies

Next, we will present some of the most commonly used OPE estimators before outlining the limitations of these methodologies. This motivates our proposal of an alternative OPE estimator.

The value of the target policy can be expressed as the expectation of outcome $Y$ under the target data distribution $p_{\pi^*}(x, a, y)$. However in most cases, we do not have access to samples from this target distribution and hence we have to resort to importance sampling methods.

**Inverse Probability Weighting (IPW) estimator**   One way to compute the target policy value, $\mathbb{E}_{\pi^*}[Y]$, when only given data generated from $p_{\pi^b}(x, a, y)$ is to rewrite the policy value as follows:

$$\mathbb{E}_{\pi^*}[Y] = \int y\, p_{\pi^*}(x, a, y)\, \mathrm{d}y\, \mathrm{d}a\, \mathrm{d}x = \int y\, \underbrace{\frac{p_{\pi^*}(x, a, y)}{p_{\pi^b}(x, a, y)}}_{\rho(a,x)}\, p_{\pi^b}(x, a, y)\, \mathrm{d}y\, \mathrm{d}a\, \mathrm{d}x = \mathbb{E}_{\pi^b}\left[Y\, \rho(A, X)\right],$$

where $\rho(a, x) := \frac{p_{\pi^*}(x,a,y)}{p_{\pi^b}(x,a,y)} = \frac{\pi^*(a|x)}{\pi^b(a|x)}$, given the factorizations in Eqns. (1) and (2). This leads to the commonly used *Inverse Probability Weighting (IPW)* [12] estimator:

$$\hat{\theta}_{\text{IPW}} := \frac{1}{n} \sum_{i=1}^{n} \rho(a_i, x_i)\, y_i.$$

When the behaviour policy is known, IPW is an unbiased and consistent estimator. However, it can suffer from high variance, especially as the overlap between the behaviour and target policies decreases.

**Doubly Robust (DR) estimator**   To alleviate the high variance of IPW, [13] proposed a *Doubly Robust (DR)* estimator for OPE. DR uses an estimate of the conditional mean $\hat{\mu}(a, x) \approx \mathbb{E}[Y \mid X = x, A = a]$ (*outcome model*), as a control variate to decrease the variance of IPW. It is also doubly robust in that it yields accurate value estimates if either the importance weights $\rho(a, x)$ or the outcome model $\hat{\mu}(a, x)$ is well estimated [13, 15]. The DR estimator for $\mathbb{E}_{\pi^*}[Y]$ can be written as follows:

$$\hat{\theta}_{\text{DR}} = \frac{1}{n} \sum_{i=1}^{n} \rho(a_i, x_i)\, (y_i - \hat{\mu}(a_i, x_i)) + \hat{\eta}(\pi^*),$$

where $\hat{\eta}(\pi^*) = \frac{1}{n} \sum_{i=1}^{n} \sum_{a' \in \mathcal{A}} \hat{\mu}(a', x_i)\pi^*(a' \mid x_i) \approx \mathbb{E}_{\pi^*}[\hat{\mu}(A, X)]$. Here, $\hat{\eta}(\pi^*)$ is referred to as the Direct Method (DM) as it uses $\hat{\mu}(a, x)$ directly to estimate target policy value.

## 2.3 Limitation of existing methodologies

To estimate the value of the target policy $\pi^*$, the existing methodologies consider the shift in the joint distribution of $(X, A, Y)$ as a result of the policy shift (by weighting samples by policy ratios). As we show in Section 3.1, considering the joint shift can lead to inefficient policy evaluation and high variance especially as the policy shift increases [16]. Since our goal is to estimate $\mathbb{E}_{\pi^*}[Y]$, we will show in the next section that considering only the shift in the marginal distribution of the outcomes $Y$ from $p_{\pi^b}(Y)$ to $p_{\pi^*}(Y)$, leads to a more efficient OPE methodology compared to existing approaches.

To better comprehend why only considering the shift in the marginal distribution is advantageous, let us examine an extreme example where we assume that $Y \perp\!\!\!\perp A \mid X$, i.e., the outcome $Y$ for a user $X$ is independent of the action $A$ taken. In this specific instance, $\mathbb{E}_{\pi^*}[Y] = \mathbb{E}_{\pi^b}[Y] \approx 1/n \sum_{i=1}^{n} y_i$, indicating that an unweighted empirical mean serves as a suitable unbiased estimator of $\mathbb{E}_{\pi^*}[Y]$. However, IPW and DR estimators use policy ratios $\rho(a, x) = \frac{\pi^*(a|x)}{\pi^b(a|x)}$ as importance weights. In case of large policy shifts, these ratios may vary significantly, leading to high variance in IPW and DR.

In this particular example, the shift in policies is inconsequential as it does not impact the distribution of outcomes $Y$. Hence, IPW and DR estimators introduce additional variance due to the policy ratios when they are not actually required. This limitation is not exclusive to this special case; in general, methodologies like IPW and DR exhibit high variance when there is low overlap between target and behavior policies [16] even if the resulting shift in marginals of the outcome $Y$ is not significant.

Therefore, we propose the *Marginal Ratio (MR)* OPE estimator for contextual bandits in the subsequent section, which circumvents these issues by focusing on the shift in the marginal distribution of the outcomes $Y$. Additionally, we provide extensive theoretical insights on the comparison of MR to existing state-of-the-art methods, such as IPW and DR.

## 3    Marginal Ratio (MR) estimator

Our method's key insight involves weighting outcomes by the marginal density ratio of outcome $Y$:

$$\mathbb{E}_{\pi^*}[Y] = \int_{\mathcal{Y}} y \, p_{\pi^*}(y) \, \mathrm{d}y = \int_{\mathcal{Y}} y \, \frac{p_{\pi^*}(y)}{p_{\pi^b}(y)} \, p_{\pi^b}(y) \, \mathrm{d}y = \mathbb{E}_{\pi^b}\left[Y \, w(Y)\right],$$

where $w(y) := \frac{p_{\pi^*}(y)}{p_{\pi^b}(y)}$. This leads to the Marginal Ratio OPE estimator:

$$\hat{\theta}_{\mathrm{MR}} := \frac{1}{n} \sum_{i=1}^n w(y_i) \, y_i.$$

In Section 3.1 we prove that by only considering the shift in the marginal distribution of outcomes, the MR estimator achieves a lower variance than the standard OPE methods. In fact, this estimator does not depend on the shift between target and behaviour policies directly. Instead, it depends on the shift between the marginals $p_{\pi^b}(y)$ and $p_{\pi^*}(y)$.

**Estimation of $w(y)$**    When the weights $w(y)$ are known exactly, the MR estimator is unbiased and consistent. However, in practice the weights $w(y)$ are often not known and must be estimated using the logged data $\mathcal{D}$. Here, we outline an efficient way to estimate $w(y)$ by first representing it as a conditional expectation, which can subsequently be expressed as the solution to a regression problem.

**Lemma 3.1.** *Let $w(y) = \frac{p_{\pi^*}(y)}{p_{\pi^b}(y)}$ and $\rho(a,x) = \frac{\pi^*(a|x)}{\pi^b(a|x)}$, then $w(y) = \mathbb{E}_{\pi^b}[\rho(A,X) \mid Y = y]$, and,*

$$w = \arg\min_f \mathbb{E}_{\pi^b}\left[(\rho(A,X) - f(Y))^2\right]. \tag{3}$$

Lemma 3.1 allows us to approximate $w(y)$ using a parametric family $\{f_\phi : \mathbb{R} \to \mathbb{R} \mid \phi \in \Phi\}$ (e.g. neural networks) and defining $\hat{w}(y) := f_{\phi^*}(y)$, where $\phi^*$ solves the regression problem in Eq. (3).

Note that MR can also be estimated alternatively by directly estimating $h(y) := w(y) \, y$ using a similar regression technique as above and computing $\hat{\theta}_{\mathrm{MR}} = 1/n \sum_{i=1}^n h(y_i)$. We include additional details along with empirical comparisons in Appendix F.1.1.

### 3.1    Theoretical analysis

Recall that the traditional OPE estimators like IPW and DR use importance weights which account for the the shift in the joint distributions of $(X, A, Y)$. In this section, we prove that by considering only the shift in the marginal distribution of $Y$ instead, MR achieves better variance properties than these estimators. Our analysis in this subsection assumes that the ratios $\rho(a,x)$ and $w(y)$ are known exactly. Since the OPE estimators considered are unbiased in this case, our analysis of variance is analogous to that of the mean squared error (MSE) here. We address the case where the weights are not known exactly in Section 3.1.2. First, we make precise our intuition that the shift in the joint distribution of $(X, A, Y)$ is 'greater' than the shift in the marginal distribution of outcomes $Y$. We formalise this using the notion of $f$-divergences.

**Proposition 3.2.** *Let $f : [0, \infty) \to \mathbb{R}$ be a convex function with $f(1) = 0$, and $\mathrm{D}_f(P||Q)$ denotes the $f$-divergence between distributions $P$ and $Q$. Then,*

$$\mathrm{D}_f\left(p_{\pi^*}(x,a,y) \,||\, p_{\pi^b}(x,a,y)\right) \geq \mathrm{D}_f\left(p_{\pi^*}(y) \,||\, p_{\pi^b}(y)\right).$$

**Intuition**    Proposition 3.2 shows that the shift in the joint distributions is at least as 'large' as the shift in the marginals of the outcome $Y$. Traditional OPE estimators, therefore take into consideration more of a distribution shift than needed, and consequently lead to inefficient estimators. In contrast, the MR estimator mitigates this problem by only considering the shift in the marginal distributions of outcomes resulting from the policy shift. This provides further intuition on why the MR estimator has lower variance compared to existing methods.

**Proposition 3.3** (Variance comparison with IPW estimator). *When the weights $\rho(a,x)$ and $w(y)$ are known exactly, we have that $\mathrm{Var}_{\pi^b}[\hat{\theta}_{\mathrm{MR}}] \leq \mathrm{Var}_{\pi^b}[\hat{\theta}_{\mathrm{IPW}}]$. In particular,*

$$\mathrm{Var}_{\pi^b}[\hat{\theta}_{\mathrm{IPW}}] - \mathrm{Var}_{\pi^b}[\hat{\theta}_{\mathrm{MR}}] = \frac{1}{n}\mathbb{E}_{\pi^b}\left[\mathrm{Var}_{\pi^b}\left[\rho(A,X)\mid Y\right]Y^2\right] \geq 0.$$

**Intuition** Proposition 3.3 shows that the variance of MR estimator is smaller than that of the IPW estimator when the weights are known exactly. Moreover, the proposition also shows that the difference between the two variances will increases as the variance $\mathrm{Var}_{\pi^b}\left[\rho(A,X)\mid Y\right]$ increases. This variance is likely to be large when the policy shift between $\pi^b$ and $\pi^*$ is large, or when the dimensions of contexts $X$ and/or the actions $A$ is large, and therefore in these cases the MR estimator will perform increasingly better than the IPW estimator. A similar phenomenon occurs for DR as we show next, even though in this case the variance of MR is not in general smaller than that of DR.

**Proposition 3.4** (Variance comparison with DR estimator). *When the weights $\rho(a,x)$ and $w(y)$ are known exactly and $\mu(A,X) := \mathbb{E}[Y\mid X,A]$, we have that,*

$$\mathrm{Var}_{\pi^b}[\hat{\theta}_{\mathrm{DR}}] - \mathrm{Var}_{\pi^b}[\hat{\theta}_{\mathrm{MR}}] \geq \frac{1}{n}\mathbb{E}_{\pi^b}\left[\mathrm{Var}_{\pi^b}\left[\rho(A,X)\,Y\mid Y\right] - \mathrm{Var}_{\pi^b}\left[\rho(A,X)\,\mu(A,X)\mid X\right]\right].$$

**Intuition** Proposition 3.4 shows that if $\mathrm{Var}_{\pi^b}\left[\rho(A,X)\,Y\mid Y\right]$ is greater than $\mathrm{Var}_{\pi^b}\left[\rho(A,X)\,\mu(A,X)\mid X\right]$ on average, the variance of the MR estimator will be less than that of the DR estimator. Intuitively, this will occur when the dimension of context space $\mathcal{X}$ is high because in this case the conditional variance over $X$ and $A$, $\mathrm{Var}_{\pi^b}\left[\rho(A,X)\,Y\mid Y\right]$ is likely to be greater than the conditional variance over $A$, $\mathrm{Var}_{\pi^b}\left[\rho(A,X)\,\mu(A,X)\mid X\right]$. Our empirical results in Appendix F.2 are consistent with this intuition. Additionally, we also provide theoretical comparisons with other extensions of DR, such as Switch-DR [5] and DR with Optimistic Shrinkage (DRos) [17] in Appendix B, and show that a similar intuition applies for these results. We emphasise that the well known results in [5] which show that IPW and DR estimators achieve the optimal *worst case* variance (where the worst case is taken over a class of possible outcome distributions $Y\mid X,A$) are not at odds with our results presented here (as the distribution of $Y\mid X,A$ is fixed in our setting).

### 3.1.1 Comparison with Marginalised Inverse Propensity Score (MIPS) [14]

In this section, we compare MR against the recently proposed Marginalised Inverse Propensity Score (MIPS) estimator [14], which uses a marginalisation technique to reduce variance and provides a robust OPE estimate specifically in contextual bandits with large action spaces. We prove that the MR estimator achieves lower variance than the MIPS estimator and doesn't require new assumptions.

**MIPS estimator** As we mentioned earlier, the variance of the IPW estimator may be high when the action $A$ is high dimensional. To mitigate this, the MIPS estimator assumes the existence of a (potentially lower dimensional) action embedding $E$, which summarises all 'relevant' information about the action $A$. Formally, this assumption can be written as follows:

**Assumption 3.5.** The action $A$ has no direct effect on the outcome $Y$, i.e., $Y \perp\!\!\!\perp A \mid X, E$.

For example, in the setting of a recommendation system where $A$ corresponds to the items recommended, $E$ may correspond to the item categories. Assumption 3.5 then intuitively means that item category $E$ encodes all relevant information about the item $A$ which determines the outcome $Y$. Assuming that such action embedding $E$ exists, [14] prove that the MIPS estimator $\hat{\theta}_{\mathrm{MIPS}}$, defined as

$$\hat{\theta}_{\mathrm{MIPS}} := \frac{1}{n}\sum_{i=1}^{n}\frac{p_{\pi^*}(e_i,x_i)}{p_{\pi^b}(e_i,x_i)}\,y_i = \frac{1}{n}\sum_{i=1}^{n}\frac{p_{\pi^*}(e_i\mid x_i)}{p_{\pi^b}(e_i\mid x_i)}\,y_i,$$

provides an unbiased estimator of target policy value $\mathbb{E}_{\pi^*}[Y]$. Moreover, $\mathrm{Var}_{\pi^b}[\hat{\theta}_{\mathrm{MIPS}}] \leq \mathrm{Var}_{\pi^b}[\hat{\theta}_{\mathrm{IPW}}]$.

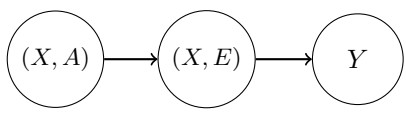

Figure 1: Bayesian network corresponding to Assumption 3.5.

**Intuition** The context-embedding pair $(X,E)$ can be seen as a representation of the context-action pair $(X,A)$ which contains less 'redundant information' regarding the outcome $Y$. Intuitively, the MIPS estimator, which only considers the shift in the distribution of $(X,E)$ is therefore more efficient than the IPW estimator (which considers the shift in the distribution of $(X,A)$ instead).

**MR achieves lower variance than MIPS**  Given the intuition above, we should achieve greater variance reduction as the amount of redundant information in the representation $(X, E)$ decreases. We formalise this in Appendix D and show that the variance of MIPS estimator decreases as the representation gets closer to $Y$ in terms of information content. As a result, we achieve the greatest variance reduction by considering the marginal shift in the outcome $Y$ itself (as in MR) rather than the shift in the representation $(X, E)$ (as in MIPS). The following result formalizes this finding.

**Theorem 3.6.** *When the weights* $w(y)$, $\frac{p_{\pi^*}(e,x)}{p_{\pi^b}(e,x)}$ *and* $\rho(a, x)$ *are known exactly, then under Assumption 3.5,* $\mathbb{E}_{\pi^b}[\hat{\theta}_{\mathrm{MR}}] = \mathbb{E}_{\pi^b}[\hat{\theta}_{\mathrm{MIPS}}] = \mathbb{E}_{\pi^*}[Y]$, *and* $\mathrm{Var}_{\pi^b}[\hat{\theta}_{\mathrm{MR}}] \leq \mathrm{Var}_{\pi^b}[\hat{\theta}_{\mathrm{MIPS}}] \leq \mathrm{Var}_{\pi^b}[\hat{\theta}_{\mathrm{IPW}}]$.

This analysis provides a link between the MR and MIPS estimators in the framework of contextual bandits, and shows that the MR estimator achieves lower variance than MIPS estimator while not requiring any additional assumptions (e.g. Assumption 3.5 as in MIPS). We also verify this empirically in Section 5.1 by reproducing the experimental setup in [14] along with the MR baseline.

### 3.1.2   Weight estimation error

Our analysis so far assumes prior knowledge of the behavior policy $\pi^b$ and the marginal ratios $w(y)$. However, in practice, both quantities are often unknown and must be estimated from data. To this end, we assume access to an additional training dataset $\mathcal{D}_{\mathrm{tr}} = \{(x_i^{\mathrm{tr}}, a_i^{\mathrm{tr}}, y_i^{\mathrm{tr}})\}_{i=1}^m$ (for weight estimation), in addition to the evaluation dataset $\mathcal{D} = \{(x_i, a_i, y_i)\}_{i=1}^n$ (for computing the OPE estimate). The estimation of $\hat{w}(y)$ involves a two-step process that exclusively utilizes data from $\mathcal{D}_{\mathrm{tr}}$:

(i) First, we estimate the policy ratio $\hat{\rho}(a, x) \approx \frac{\pi^*(a|x)}{\pi^b(a|x)}$. This can be achieved by estimating the behaviour policy $\hat{\pi}^b$, and defining $\hat{\rho}(a, x) := \frac{\pi^*(a|x)}{\hat{\pi}^b(a|x)}$. Alternatively, $\hat{\rho}(a, x)$ can also be estimated directly by using density ratio estimation techniques as in [18].

(ii) Secondly, we estimate the weights $\hat{w}(y)$ using Eq. (3) with $\hat{\rho}$ instead of $\rho$.

In practice, one may consider splitting $\mathcal{D}_{\mathrm{tr}}$ for each estimation step outlined above. Moreover, each approximation step may introduce bias and therefore, the MR estimator may have two sources of bias. While classical OPE methods like IPW and DR also suffer from bias because of $\hat{\rho}$ estimation, the estimation of $\hat{w}(y)$ is specific to MR. However, we show below that given any policy ratio estimate $\hat{\rho}$, if $\hat{w}(y)$ approximates $\mathbb{E}_{\pi^b}[\hat{\rho}(A, X) \mid Y = y]$ 'well enough' (i.e., the estimation step (ii) shown above is 'accurate enough'), then MR achieves a lower variance than IPW and incurs little extra bias.

**Proposition 3.7.** *Suppose that the IPW and MR estimators are defined as,*

$$\tilde{\theta}_{\mathrm{IPW}} := \frac{1}{n} \sum_{i=1}^n \hat{\rho}(a_i, x_i)\, y_i, \quad \text{and} \quad \tilde{\theta}_{\mathrm{MR}} := \frac{1}{n} \sum_{i=1}^n \hat{w}(y_i)\, y_i,$$

*and define the approximation error as* $\epsilon := \hat{w}(Y) - \tilde{w}(Y)$, *where* $\tilde{w}(Y) := \mathbb{E}_{\pi^b}[\hat{\rho}(A, X) \mid Y]$. *Then we have that,* $\mathrm{Bias}(\tilde{\theta}_{\mathrm{MR}}) - \mathrm{Bias}(\tilde{\theta}_{\mathrm{IPW}}) = \mathbb{E}_{\pi^b}[\epsilon\, Y]$. *Moreover,*

$$\mathrm{Var}_{\pi^b}[\tilde{\theta}_{\mathrm{IPW}}] - \mathrm{Var}_{\pi^b}[\tilde{\theta}_{\mathrm{MR}}] = \frac{1}{n}(\underbrace{\mathbb{E}_{\pi^b}[\mathrm{Var}_{\pi^b}[\hat{\rho}(A, X)\, Y \mid Y]]}_{\geq 0} - \mathrm{Var}_{\pi^b}[\epsilon\, Y] - 2\,\mathrm{Cov}(\tilde{w}(Y)\, Y, \epsilon\, Y)). \quad (4)$$

**Intuition**  The $\epsilon$ term defined in Proposition 3.7 denotes the error of the second approximation step outlined above. As a direct consequence of this result, we show in Appendix C that as the error $\epsilon$ becomes small (specifically as $\mathbb{E}_{\pi^b}[\epsilon^2] \to 0$), the difference between biases of MR and IPW estimator becomes negligible. Likewise, the terms $\mathrm{Var}_{\pi^b}[\epsilon\, Y]$ and $\mathrm{Cov}(\tilde{w}(Y)\, Y, \epsilon\, Y)$ in Eq. (4) will also be small and as a result the variance of MR will be lower than that of IPW (as the first term is positive).

In fact, using recent results regarding the generalisation error of neural networks [19], we show that when using 2-layer wide neural networks to approximate the weights $\hat{w}(y)$, the estimation error $\epsilon$ declines with increasing training data size $m$. Specifically, under certain regularity assumptions we obtain $\mathbb{E}_{\pi^b}[\epsilon^2] = O(m^{-2/3})$. Using this we show that as the training data size $m$ increases, the biases of MR and IPW estimators become roughly equal with a high probability, and

$$\mathrm{Var}_{\pi^b}[\tilde{\theta}_{\mathrm{IPW}}] - \mathrm{Var}_{\pi^b}[\tilde{\theta}_{\mathrm{MR}}] = \frac{1}{n}\mathbb{E}_{\pi^b}[\mathrm{Var}_{\pi^b}[\hat{\rho}(A, X)\, Y \mid Y]] + O(m^{-1/3}).$$

Therefore the variance of MR estimator falls below that of IPW for large enough $m$. The empirical results shown in Appendix F.2 are consistent with this result. Due to space constraints, the main technical result has been included in Appendix C.

## 3.2 Application to causal inference

Beyond contextual bandits, the variance reduction properties of the MR estimator make it highly useful in a wide variety of other applications. Here, we show one such application in the field of causal inference, where MR can be used for the estimation of average treatment effect (ATE) [20] and leads to some desirable properties in comparison to the conventional ATE estimation approaches. Specifically, we illustrate that the MR estimator for ATE utilizes the evaluation data $\mathcal{D}$ more efficiently and achieves lower variance than state-of-the-art ATE estimators and consequently provides more accurate ATE estimates. To be concrete, the goal in this setting is to estimate ATE, defined as follows:

$$\text{ATE} := \mathbb{E}[Y(1) - Y(0)].$$

Here $Y(a)$ corresponds to the outcome under a deterministic policy $\pi_a(a' \mid x) := \mathbb{1}(a' = a)$. Hence any OPE estimator can be used to estimate $\mathbb{E}[Y(a)]$ (and therefore ATE) by considering target policy $\pi^* = \pi_a$. An important distinction between MR and existing approaches (like IPW or DR) is that, when estimating $\mathbb{E}[Y(a)]$, the existing approaches only use datapoints in $\mathcal{D}$ with $A = a$. To see why this is the case, we note that the policy ratios $\frac{\pi^*(A|X)}{\pi^b(A|X)} = \frac{\mathbb{1}(A=a)}{\pi^b(A|X)}$ are zero when $A \neq a$. In contrast, the MR weights $\frac{p_{\pi^*}(Y)}{p_{\pi^b}(Y)}$ are not necessarily zero for datapoints with $A \neq a$, and therefore the MR estimator uses all evaluation datapoints when estimating $\mathbb{E}[Y(a)]$.

As such we show that MR applied to ATE estimation leads to a smaller variance than the existing approaches. Moreover, because MR is able to use all datapoints when estimating $\mathbb{E}[Y(a)]$, MR will generally be more accurate than the existing methods especially in the setting where the data is imbalanced, i.e., the number of datapoints with $A = a$ is small for a specific action $a$. In Appendix E, we formalise this variance reduction of the MR ATE estimator compared to IPW and DR estimators, by deriving analogous results to Propositions 3.3 and 3.4. In addition, we also show empirically in Section 5.3 that the MR ATE estimator outperforms the most commonly used ATE estimators.

# 4 Related Work

Off-Policy evaluation is a central problem both in contextual bandits [13, 5, 21, 6, 7, 17, 22, 8, 23] and in RL [24–27]. Existing OPE methodologies can be broadly categorised into Direct Method (DM), Inverse Probability Weighting (IPW), and Doubly Robust (DR). While DM typically has a low variance, it suffers from high bias when the reward model is misspecified [28]. On the other hand, IPW [12] and DR [13, 5, 17] use policy ratios as importance weights when estimating policy value and suffer from high variance as overlap between behaviour and target policies increases or as the action/context space gets larger [29, 14]. To circumvent this problem, techniques like weight clipping or normalisation [4, 30, 31] are often employed, however, these can often increase bias.

In contrast to these approaches, [14] propose MIPS, which considers the marginal shift in the distribution of a lower dimensional embedding of the action space. While this approach reduces the variance associated with IPW, we show in Section 3.1.1 that the MR estimator achieves a lower variance than MIPS while not requiring any additional assumptions (like Assumption 3.5).

In the context of Reinforcement Learning (RL), various marginalisation techniques of importance weights have been used to propose OPE methodologies. [21, 25, 26] use methods which considers the shift in the marginal distribution of the states, and applies importance weighting with respect to this marginal shift rather than the trajectory distribution. Similarly, [32] use marginalisation for OPE in deep RL, where the goal is to consider the shift in marginal distributions of state and action. Although marginalization is a key trick of these estimators, these techniques do not consider the marginal shift in reward as in MR and are aimed at resolving the curse of horizon, a problem specific to RL. Apart from this, [33] propose a general framework of OPE based on conditional expectations of importance ratios for variance reduction. While their proposed framework includes reward conditioned importance ratios, this is not the main focus and there is little theoretical and empirical comparison of their proposed methodology with existing state-of-the-art methods like DR.

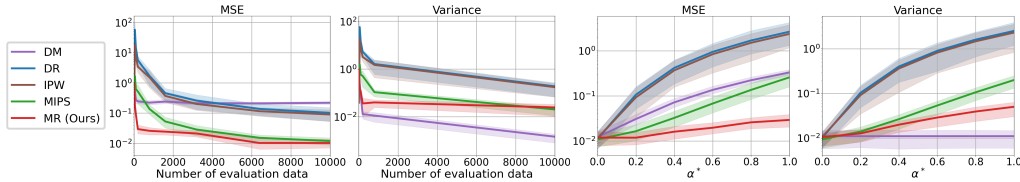

(a) Results with varying size of evaluation dataset $n$.

(b) Results with varying $\alpha^*$.

Figure 2: Results for synthetic data experiment. In 2a we have $\alpha^* = 0.8$ and in 2b we have $n = 800$.

Finally we note that the idea of approximating the ratio of intractable marginal densities via leveraging the fact that this ratio can be reformulated as the conditional expectation of a ratio of tractable densities is a standard idea in computational statistics [34] and has been exploited more recently to perform likelihood-free inference [35]. In particular, while [34] typically approximates this expectation through Markov chain Monte Carlo, [35] uses regression instead, however without any theory.

## 5 Empirical Evaluation

In this section, we provide empirical evidence to support our theoretical results by investigating the performance of our MR estimator against the current state-of-the-art OPE methods. The code to reproduce our experiments has been made available at: github.com/faaizT/MR-OPE.

### 5.1 Experiments on synthetic data

For our synthetic data experiment, we reproduce the experimental setup for the synthetic data experiment in [14] by reusing their code with minor modifications. Specifically, $\mathcal{X} \subseteq \mathbb{R}^d$, for various values of $d$ as described below. Likewise, the action space $\mathcal{A} = \{0, \ldots, n_a - 1\}$, with $n_a$ taking a range of different values. Additional details regarding the reward function, behaviour policy $\pi^b$, and the estimation of weights $\hat{w}(y)$ have been included in Appendix F.2 for completeness.

**Target policies**  To investigate the effect of increasing policy shift, we define a class of policies,

$$\pi^{\alpha^*}(a|x) = \alpha^* \mathbb{1}(a = \arg\max_{a' \in \mathcal{A}} q(x, a')) + \frac{1 - \alpha^*}{|\mathcal{A}|} \quad \text{where} \quad q(x, a) := \mathbb{E}[Y \mid X = x, A = a],$$

where $\alpha^* \in [0, 1]$ allows us to control the shift between $\pi^b$ and $\pi^*$. In particular, as we show later, the shift between $\pi^b$ and $\pi^*$ increases as $\alpha^* \to 1$. Using the ground truth behaviour policy $\pi^b$, we generate a dataset which is split into training and evaluation datasets of sizes $m$ and $n$ respectively.

**Baselines**  We compare our estimator with DM, IPW, DR and MIPS estimators. Our setup includes action embeddings $E$ satisfying Assumption 3.5, and so MIPS is unbiased. Additional baselines have been considered in Appendix F.2. For MR, we split the training data to estimate $\hat{\pi}^b$ and $\hat{w}(y)$, whereas for all other baselines we use the entire training data to estimate $\hat{\pi}^b$ for a fair comparison.

**Results**  We compute the target policy value using the $n$ evaluation datapoints. Here, the MSE of the estimators is computed over 10 different sets of logged data replicated with different seeds. The results presented have context dimension $d = 1000$, number of actions $n_a = 100$ and training data size $m = 5000$. More experiments for a variety of parameter values are included in Appendix F.2.

**Varying number of evaluation data $n$**  In Figure 2a we plot the results with increasing size of evaluation data $n$ increases. MR achieves the smallest MSE among all the baselines considered when $n$ is small, with the MSE of MR being at least an order of magnitude smaller than every baseline for $n \leq 500$. This shows that MR is significantly more accurate than the baselines when the size of the evaluation data is small. As $n \to \infty$, the difference between the results for MR and MIPS decreases. However, MR attains smaller variance and MSE than MIPS generally, verifying our analysis in Section 3.1.1. Moreover, Figure 2a shows that while the variance of MR is greater than that of DM, it still achieves the lowest MSE overall, owing to the high bias of DM.

**Varying** $\alpha^*$    As $\alpha^*$ parameter of the target policy increases, so does the shift between the policies $\pi^b$ and $\pi^{\alpha^*}$ as illustrated by the figure on the right, which plots the KL-divergence $D_{\text{KL}}(\pi^b \,||\, \pi^{\alpha^*})$ as a function of $\alpha$. Figure 2b plots the results for increasing policy shift. Overall, the MSE of MR estimator is lowest among all the baselines. Moreover, while the MSE and variance of all estimators increase with increasing $\alpha^*$ the increase in these quantities is lower for the MR estimator than for the other baselines. Therefore, the relative performance of MR estimator improves with increasing policy shift and MR remains robust to increase in policy shift.

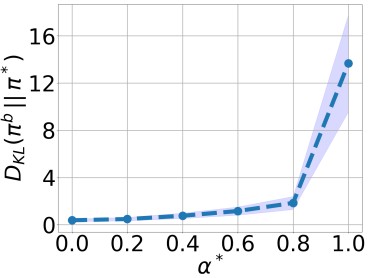

**Additional ablation studies**    In Appendix F.2, we investigate the effect of varying context dimensions $d$, number of actions $n_a$ and number of training data $m$. In every case, we observe that the MR estimator has a smaller MSE than all other baselines considered. In particular, MR remains robust to increasing $n_a$ whereas the MSE and variance of IPW and DR estimators degrade substantially when $n_a \geq 2000$. Likewise, MR outperforms the baselines even when the training data size $m$ is small.

Table 1: Mean squared error of target policy value with standard errors over 10 different seeds for different classification datasets. Here, number of evaluation data $n = 1000$, and $\alpha^* = 0.6$.

| Dataset | Digits | Letter | OptDigits | PenDigits | SatImage | Mnist | CIFAR-100 |
|---|---|---|---|---|---|---|---|
| DM | 0.1508±0.0015 | 0.0886±0.0026 | 0.0485±0.0016 | 0.0520±0.0016 | 0.0208±0.0009 | 0.1109±0.0014 | 0.0020±0.0001 |
| DR | 0.1334±0.0400 | 35.085±17.768 | 0.0464±0.0061 | 0.2343±0.1404 | 0.0560±0.0395 | 0.2617±0.0139 | 3823.9±2023.2 |
| DRos | 0.0847±0.0025 | 0.2363±0.0586 | 0.0384±0.0025 | 0.0138±0.0029 | 0.0078±0.0008 | 0.2151±0.0061 | 0.2628±0.1087 |
| IPW | 0.1632±0.0462 | 45.253±22.057 | 0.0844±0.0056 | 0.1342±0.0531 | 0.0900±0.0676 | 0.3359±0.0118 | 4116.9±2097.9 |
| SwitchDR | 0.0982±0.0032 | 0.2387±0.0507 | 0.0557±0.0047 | 0.0342±0.0090 | 0.0136±0.0012 | 0.2750±0.0102 | 1.1644±0.8227 |
| MR (Ours) | **0.0034±0.0001** | **0.0018±0.0004** | **0.0006±0.0002** | **0.0008±0.0002** | **0.0016±0.0003** | **0.0121±0.0009** | **0.0007±0.0002** |

## 5.2    Experiments on classification datasets

Following previous works on OPE in contextual bandits [13, 22, 36, 5], we transform classification datasets into contextual bandit feedback data in this experiment. We consider five UCI classification datasets [37] as well as Mnist [38] and CIFAR-100 [39] datasets, each of which comprises $\{(x_i, a_i^{\text{gt}})\}_i$, where $x_i \in \mathcal{X}$ are feature vectors and $a_i^{\text{gt}} \in \mathcal{A}$ are the ground-truth labels. In the contextual bandits setup, the feature vectors $x_i$ are considered to be the contexts, whereas the actions correspond to the possible class of labels. For the context vector $x_i$ and the action $a_i$, the reward $y_i$ is defined as $y_i := \mathbb{1}(a_i = a_i^{\text{gt}})$, i.e., the reward is 1 when the action is the same as the ground truth label and 0 otherwise. Here, the baselines considered include the DM, IPW and DR estimators as well as Switch-DR [5] and DR with Optimistic Shrinkage (DRos) [17]. We do not consider a MIPS baseline here as there is no natural embedding $E$ of $\mathcal{A}$. Additional details are provided in Appendix F.3.

In Table 1, we present the results with number of evaluation data $n = 1000$ and number of training data $m = 500$. The table shows that across all datasets, MR achieves the lowest MSE among all methods. Moreover, for the Letter and CIFAR-100 datasets the IPW and DR yield large bias and variance arising from poor policy estimates $\widehat{\pi}^b$. Despite this, the MR estimator which utilizes the *same* $\widehat{\pi}^b$ for the estimation of $\hat{w}(y)$ leads to much more accurate results. We also verify that MR outperforms the baselines for increasing policy shift and evaluation data $n$ in Appendix F.3.

## 5.3    Application to ATE estimation

In this experiment, we investigate the empirical performance of the MR estimator for ATE estimation.

**Twins dataset**    We use the Twins dataset studied in [40], which comprises data from twin births in the USA between 1989-1991. The treatment $a = 1$ corresponds to being born the heavier twin and the outcome $Y$ corresponds to the mortality of each of the twins in their first year of life. Specifically, $Y(1)$ corresponds to the mortality of the heavier twin (and likewise for $Y(0)$). To simulate the observational study, we follow a similar strategy as in [40] to selectively hide one of the two twins as explained in Appendix F.4. We obtain a total of 11,984 datapoints, of which 5000 datapoints are used to train the behaviour policy $\widehat{\pi}^b$ and outcome model $\hat{q}(x, a)$.

Table 2: Mean absolute ATE estimation error $\epsilon_{\text{ATE}}$ with standard errors over 10 different seeds, for increasing number of evaluation data $n$.

| $n$ | 50 | 200 | 1600 | 3200 |
|---|---|---|---|---|
| DM | 0.092±0.003 | 0.092±0.003 | 0.092±0.004 | 0.092±0.004 |
| DR | 0.101±0.024 | **0.065±0.009** | 0.071±0.005 | 0.069±0.004 |
| DRos | 0.100±0.017 | 0.089±0.006 | 0.093±0.004 | 0.087±0.004 |
| IPW | 0.092±0.024 | 0.088±0.014 | 0.067±0.007 | 0.067±0.007 |
| SwitchDR | 0.101±0.024 | **0.065±0.009** | 0.071±0.005 | 0.069±0.004 |
| MR (Ours) | **0.062±0.007** | **0.065±0.007** | **0.061±0.005** | **0.061±0.006** |

Here, we consider the same baselines as the classification data experiments in previous section. For our evaluation, we consider the absolute error in ATE estimation, $\epsilon_{\text{ATE}}$, defined as: $\epsilon_{\text{ATE}} := |\hat{\theta}_{\text{ATE}}^{(n)} - \theta_{\text{ATE}}|$. Here, $\hat{\theta}_{\text{ATE}}^{(n)}$ denotes the value of the ATE estimated using $n$ evaluation datapoints. We compute the ATE value using the $n$ evaluation datapoints, over 10 different sets of observational data (using different seeds). Table 2 shows that MR achieves the lowest estimation error $\epsilon_{\text{ATE}}$ for all values of $n$ considered here. While the performance of other baselines improves with increasing $n$, MR outperforms them all.

# 6  Discussion

In this paper, we proposed an OPE method for contextual bandits called marginal ratio (MR) estimator, which considers only the shift in the marginal distribution of the outcomes resulting from the policy shift. Our theoretical and empirical analysis showed that MR achieves better variance and MSE compared to the current state-of-the-art methods and is more data efficient overall. Additionally, we demonstrated that MR applied to ATE estimation provides more accurate results than most commonly used methods. Next, we discuss limitations of our methodology and possible avenues for future work.

**Limitations**    The MR estimator requires the additional step of estimating $\hat{w}(y)$ which may introduce an additional source of bias in the value estimation. However, $\hat{w}(y)$ can be estimated by solving a simple 1d regression problem, and as we show empirically in Appendix F, MR achieves the smallest bias among all baselines considered in most cases. Most notably, our ablation study in Appendix F.2 shows that even when the training data is reasonably small, MR outperforms the baselines considered.

**Future work**    The MR estimator can also be applied to policy optimisation problems, where the data collected using an 'old' policy is used to learn a new policy. This approach has been used in Proximal Policy Optimisation (PPO) [41] for example, which has gained immense popularity and has been applied to reinforcement learning with human feedback (RLHF) [42]. We believe that the MR estimator applied to these methodologies could lead to improvements in the stability and convergence of these optimisation schemes, given its favourable variance properties.

# Acknowledgements

We would like to thank Jake Fawkes, Siu Lun Chau, Shahine Bouabid and Robert Hu for their useful feedback. We also appreciate the insights and constructive criticisms provided by the anonymous reviewers. MFT acknowledges his PhD funding from Google DeepMind.

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

## A  Proofs

*Proof of Lemma 3.1.* First, we express the weights $w(y)$ as the conditional expectation as follows:

$$
\begin{aligned}
w(y) &= \frac{p_{\pi^*}(y)}{p_{\pi^b}(y)} \\
&= \int_{\mathcal{X},\mathcal{A}} \frac{p_{\pi^*}(x,a,y)}{p_{\pi^b}(y)} \, \mathrm{d}a \, \mathrm{d}x \\
&= \int_{\mathcal{X},\mathcal{A}} \frac{p_{\pi^*}(x,a,y)}{p_{\pi^b}(y)} \frac{p_{\pi^b}(x,a\mid y)}{p_{\pi^b}(x,a\mid y)} \, \mathrm{d}a \, \mathrm{d}x \\
&= \int_{\mathcal{X},\mathcal{A}} \frac{p_{\pi^*}(x,a,y)}{p_{\pi^b}(x,a,y)} \, p_{\pi^b}(x,a\mid y) \, \mathrm{d}a \, \mathrm{d}x \\
&= \int_{\mathcal{X},\mathcal{A}} \rho(a,x) \, p_{\pi^b}(x,a\mid y) \, \mathrm{d}a \, \mathrm{d}x \\
&= \mathbb{E}_{\pi^b}[\rho(A,X)\mid Y=y],
\end{aligned}
$$

where $\rho(a,x) = \frac{p_{\pi^*}(x,a,y)}{p_{\pi^b}(x,a,y)} = \frac{\pi^*(a\mid x)}{\pi^b(a\mid x)}$. Since conditional expectations can be defined as the solution of regression problem, the result follows. $\qquad\square$

*Proof of Proposition 3.2.* We have

$$
\begin{aligned}
\mathrm{D}_f\left(p_{\pi^*}(x,a,y)\,\|\,p_{\pi^b}(x,a,y)\right) &= \mathbb{E}_{\pi^b}\left[f\left(\frac{p_{\pi^*}(X,A,Y)}{p_{\pi^b}(X,A,Y)}\right)\right] \\
&= \mathbb{E}_{\pi^b}\left[f\left(\frac{\pi^*(A\mid X)}{\pi^b(A\mid X)}\right)\right] \\
&= \mathbb{E}_{\pi^b}\left[\mathbb{E}_{\pi^b}\left[f\left(\frac{\pi^*(A\mid X)}{\pi^b(A\mid X)}\right)\bigg|Y\right]\right] \\
&\geq \mathbb{E}_{\pi^b}\left[f\left(\mathbb{E}_{\pi^b}\left[\frac{\pi^*(A\mid X)}{\pi^b(A\mid X)}\bigg|Y\right]\right)\right] \quad\text{(Jensen's inequality)} \\
&= \mathbb{E}_{\pi^b}\left[f\left(\frac{p_{\pi^*}(Y)}{p_{\pi^b}(Y)}\right)\right] \\
&= \mathrm{D}_f\left(p_{\pi^*}(y)\,\|\,p_{\pi^b}(y)\right).
\end{aligned}
$$

$\qquad\square$

*Proof of Proposition 3.3.* Since $\mathbb{E}_{\pi^b}[\hat\theta_{\mathrm{IPW}}] = \mathbb{E}_{\pi^b}[\hat\theta_{\mathrm{MR}}] = \mathbb{E}_{\pi^*}[Y]$, we have that,

$$
\begin{aligned}
\mathrm{Var}_{\pi^b}[\hat\theta_{\mathrm{IPW}}] - \mathrm{Var}_{\pi^b}[\hat\theta_{\mathrm{MR}}] &= \mathbb{E}_{\pi^b}[\hat\theta_{\mathrm{IPW}}]^2 - \mathbb{E}_{\pi^b}[\hat\theta_{\mathrm{MR}}]^2 \\
&= \frac{1}{n}\left(\mathbb{E}_{\pi^b}\left[\rho(A,X)^2\,Y^2\right] - \mathbb{E}_{\pi^b}\left[w(Y)^2\,Y^2\right]\right) \\
&= \frac{1}{n}\left(\mathbb{E}_{\pi^b}\left[\mathbb{E}_{\pi^b}[\rho(A,X)^2\mid Y]\,Y^2\right] - \mathbb{E}_{\pi^b}\left[w(Y)^2\,Y^2\right]\right) \\
&= \frac{1}{n}\left(\mathbb{E}_{\pi^b}\left[\mathbb{E}_{\pi^b}[\rho(A,X)^2\mid Y]\,Y^2\right] - \mathbb{E}_{\pi^b}\left[\mathbb{E}_{\pi^b}[\rho(A,X)\mid Y]^2\,Y^2\right]\right) \\
&= \frac{1}{n}\mathbb{E}_{\pi^b}\left[\mathrm{Var}_{\pi^b}\left[\rho(A,X)\mid Y\right]\,Y^2\right].
\end{aligned}
$$

In the second last step above, we use the fact that $w(y) = \mathbb{E}_{\pi^b}[\rho(A,X)\mid Y=y]$. $\qquad\square$

*Proof of Proposition 3.4.* Let $\hat{\mu}(a, x) \approx \mathbb{E}[Y \mid X = x, A = a]$ denote the outcome model in DR estimator. Then, using multiple applications of the law of total variance we get that

$$n \operatorname{Var}_{\pi^b}[\hat{\theta}_{\text{DR}}] = \operatorname{Var}_{\pi^b}\left[\rho(A, X)\left(Y - \hat{\mu}(A, X)\right) + \sum_{a' \in \mathcal{A}} \hat{\mu}(a', X)\, \pi^*(a' \mid X)\right]$$

$$= \operatorname{Var}_{\pi^b}\left[\rho(A, X)\left(Y - \hat{\mu}(A, X)\right) + \mathbb{E}_{\pi^*}[\hat{\mu}(A, X) \mid X]\right]$$

$$= \mathbb{E}_{\pi^b}\left[\operatorname{Var}_{\pi^b}[\rho(A, X)\left(Y - \hat{\mu}(A, X)\right) + \mathbb{E}_{\pi^*}[\hat{\mu}(A, X) \mid X] \mid X, A]\right]$$
$$+ \operatorname{Var}_{\pi^b}\left[\mathbb{E}_{\pi^b}[\rho(A, X)\left(Y - \hat{\mu}(A, X)\right) + \mathbb{E}_{\pi^*}[\hat{\mu}(A, X) \mid X] \mid X, A]\right]$$

$$= \mathbb{E}_{\pi^b}\left[\rho(A, X)^2 \operatorname{Var}[Y \mid X, A]\right]$$
$$+ \operatorname{Var}_{\pi^b}\left[\mathbb{E}_{\pi^b}[\rho(A, X)\left(Y - \hat{\mu}(A, X)\right) + \mathbb{E}_{\pi^b}[\rho(A, X)\,\hat{\mu}(A, X) \mid X] \mid X, A]\right]$$

$$= \mathbb{E}_{\pi^b}\left[\rho(A, X)^2 \operatorname{Var}[Y \mid X, A]\right]$$
$$+ \operatorname{Var}_{\pi^b}\left[\rho(A, X)\left(\mu(A, X) - \hat{\mu}(A, X)\right) + \mathbb{E}_{\pi^b}[\rho(A, X)\,\hat{\mu}(A, X) \mid X]\right]$$

$$= \mathbb{E}_{\pi^b}\left[\rho(A, X)^2 \operatorname{Var}[Y \mid X, A]\right]$$
$$+ \operatorname{Var}_{\pi^b}\left[\mathbb{E}_{\pi^b}[\rho(A, X)\left(\mu(A, X) - \hat{\mu}(A, X)\right) + \mathbb{E}_{\pi^b}[\rho(A, X)\,\hat{\mu}(A, X) \mid X] \mid X]\right]$$
$$+ \mathbb{E}_{\pi^b}\left[\operatorname{Var}_{\pi^b}[\rho(A, X)\left(\mu(A, X) - \hat{\mu}(A, X)\right) + \mathbb{E}_{\pi^b}[\rho(A, X)\,\hat{\mu}(A, X) \mid X] \mid X]\right]$$

$$= \mathbb{E}_{\pi^b}\left[\rho(A, X)^2 \operatorname{Var}[Y \mid X, A]\right] + \operatorname{Var}_{\pi^b}\left[\mathbb{E}_{\pi^b}[\rho(A, X)\,\mu(A, X) \mid X]\right]$$
$$+ \mathbb{E}_{\pi^b}\left[\operatorname{Var}_{\pi^b}[\rho(A, X)\left(\mu(A, X) - \hat{\mu}(A, X)\right) \mid X]\right]$$

$$\geq \mathbb{E}_{\pi^b}\left[\rho(A, X)^2 \operatorname{Var}[Y \mid X, A]\right] + \operatorname{Var}_{\pi^b}\left[\mathbb{E}_{\pi^b}[\rho(A, X)\,\mu(A, X) \mid X]\right].$$

Using this, we get that

$$n(\operatorname{Var}_{\pi^b}[\hat{\theta}_{\text{DR}}] - \operatorname{Var}_{\pi^b}[\hat{\theta}_{\text{MR}}])$$
$$\geq \mathbb{E}_{\pi^b}\left[\rho(A, X)^2 \operatorname{Var}[Y \mid X, A]\right] + \operatorname{Var}_{\pi^b}\left[\mathbb{E}_{\pi^b}[\rho(A, X)\,\mu(A, X) \mid X]\right] - \operatorname{Var}_{\pi^b}[w(Y)\, Y].$$

Again, using the law of total variance,

$$\operatorname{Var}_{\pi^b}[\rho(A, X)\, Y] = \mathbb{E}_{\pi^b}\left[\operatorname{Var}_{\pi^b}[\rho(A, X)\, Y \mid X, A]\right] + \operatorname{Var}_{\pi^b}\left[\mathbb{E}_{\pi^b}[\rho(A, X)\, Y \mid X, A]\right]$$
$$= \mathbb{E}_{\pi^b}\left[\rho(A, X)^2 \operatorname{Var}[Y \mid X, A]\right] + \operatorname{Var}_{\pi^b}[\rho(A, X)\,\mu(A, X)]$$
$$= \mathbb{E}_{\pi^b}\left[\rho(A, X)^2 \operatorname{Var}[Y \mid X, A]\right] + \operatorname{Var}_{\pi^b}\left[\mathbb{E}_{\pi^b}[\rho(A, X)\,\mu(A, X) \mid X]\right]$$
$$+ \mathbb{E}_{\pi^b}\left[\operatorname{Var}_{\pi^b}[\rho(A, X)\,\mu(A, X) \mid X]\right].$$

Rearranging and substituting back into the expression earlier, we get that

$$n(\operatorname{Var}_{\pi^b}[\hat{\theta}_{\text{DR}}] - \operatorname{Var}_{\pi^b}[\hat{\theta}_{\text{MR}}])$$
$$\geq \operatorname{Var}_{\pi^b}[\rho(A, X)\, Y] - \mathbb{E}_{\pi^b}\left[\operatorname{Var}_{\pi^b}[\rho(A, X)\,\mu(A, X) \mid X]\right] - \operatorname{Var}_{\pi^b}[w(Y)\, Y].$$

Now, from Proposition 3.3 we know that

$$n(\operatorname{Var}_{\pi^b}[\hat{\theta}_{\text{IPW}}] - \operatorname{Var}_{\pi^b}[\hat{\theta}_{\text{MR}}]) = \operatorname{Var}_{\pi^b}[\rho(A, X)\, Y] - \operatorname{Var}_{\pi^b}[w(Y)\, Y] = \mathbb{E}_{\pi^b}\left[\operatorname{Var}_{\pi^b}[\rho(A, X) \mid Y]\, Y^2\right].$$

Therefore,

$$n(\operatorname{Var}_{\pi^b}[\hat{\theta}_{\text{DR}}] - \operatorname{Var}_{\pi^b}[\hat{\theta}_{\text{MR}}])$$
$$\geq \mathbb{E}_{\pi^b}\left[\operatorname{Var}_{\pi^b}[\rho(A, X) \mid Y]\, Y^2\right] - \mathbb{E}_{\pi^b}\left[\operatorname{Var}_{\pi^b}[\rho(A, X)\,\mu(A, X) \mid X]\right]$$
$$= \mathbb{E}_{\pi^b}\left[\operatorname{Var}_{\pi^b}[\rho(A, X)\, Y \mid Y]\right] - \operatorname{Var}_{\pi^b}[\rho(A, X)\,\mu(A, X) \mid X]].$$

$\square$

*Proof of Theorem 3.6.* This result follows straightforwardly from Proposition D.4 in Appendix D.

$\square$

*Proof of Proposition 3.7.*

$$\operatorname{Bias}(\hat{\theta}_{\text{IPW}}) = \mathbb{E}_{\pi^b}[\hat{\rho}(A, X)\, Y] - \mathbb{E}_{\pi^*}[Y]$$
$$= \mathbb{E}_{\pi^b}\left[\mathbb{E}_{\pi^b}[\hat{\rho}(A, X) \mid Y]\, Y\right] - \mathbb{E}_{\pi^*}[Y]$$
$$= \mathbb{E}_{\pi^b}[\hat{w}(Y)\, Y] - \mathbb{E}_{\pi^b}[\epsilon\, Y] - \mathbb{E}_{\pi^*}[Y]$$
$$= \operatorname{Bias}(\hat{\theta}_{\text{MR}}) - \mathbb{E}_{\pi^b}[\epsilon\, Y].$$

Next, to prove the variance result, we first use the law of total variance to obtain

$$
\begin{aligned}
\mathrm{Var}_{\pi^b}[\hat{\theta}_{\mathrm{IPW}}] &= \frac{1}{n} \mathrm{Var}_{\pi^b}[\hat{\rho}(A, X)\, Y] \\
&= \frac{1}{n} \left( \mathrm{Var}_{\pi^b}[\mathbb{E}_{\pi^b}[\hat{\rho}(A, X)\, Y \mid Y]] + \mathbb{E}_{\pi^b}[\mathrm{Var}_{\pi^b}[\hat{\rho}(A, X)\, Y \mid Y]] \right) \\
&= \frac{1}{n} \left( \mathrm{Var}_{\pi^b}[\tilde{w}(Y)\, Y] + \mathbb{E}_{\pi^b}[\mathrm{Var}_{\pi^b}[\hat{\rho}(A, X)\, Y \mid Y]] \right).
\end{aligned}
$$

Moreover, using the fact that $\hat{w}(Y) = \tilde{w}(Y) + \epsilon$ we get that,

$$
\begin{aligned}
\mathrm{Var}_{\pi^b}[\hat{\theta}_{\mathrm{MR}}] &= \frac{1}{n} \mathrm{Var}_{\pi^b}[\hat{w}(Y)\, Y] \\
&= \frac{1}{n} \mathrm{Var}_{\pi^b}[(\tilde{w}(Y) + \epsilon)\, Y] \\
&= \frac{1}{n} \left( \mathrm{Var}_{\pi^b}[\tilde{w}(Y)\, Y] + \mathrm{Var}_{\pi^b}[\epsilon\, Y] + 2\, \mathrm{Cov}(\tilde{w}(Y)\, Y, \epsilon\, Y) \right).
\end{aligned}
$$

Putting together the two variance expressions derived above, we get that

$$
\begin{aligned}
&\mathrm{Var}_{\pi^b}[\hat{\theta}_{\mathrm{IPW}}] - \mathrm{Var}_{\pi^b}[\hat{\theta}_{\mathrm{MR}}] \\
&= \frac{1}{n} \left( \mathbb{E}_{\pi^b}[\mathrm{Var}_{\pi^b}[\hat{\rho}(A, X) \mid Y]\, Y^2] - \mathrm{Var}_{\pi^b}[\epsilon\, Y] - 2\, \mathrm{Cov}(\tilde{w}(Y)\, Y, \epsilon\, Y) \right).
\end{aligned}
$$

$\square$

# B   Comparison with extensions of the doubly robust estimator

In this section, we theoretically investigate the variance of MR against the commonly used extensions of the DR estimator, namely Switch-DR [5] and DR with Optimistic Shrinkage (DRos) [17]. At a high level, these estimators seek to reduce the variance of the vanilla DR estimator by considering modified importance weights, thereby trading off the variance for additional bias. Below, we provide the explicit definitions of these estimators for completeness.

**Switch-DR estimator**   The original DR estimator can still have a high variance when the importance weights are large due to a large policy shift. Switch-DR [5] aims to circumvent this problem by switching to DM when the importance weights are large:

$$
\hat{\theta}_{\mathrm{SwitchDR}} := \frac{1}{n} \sum_{i=1}^{n} \rho(a_i, x_i)\, (y_i - \hat{\mu}(a_i, x_i))\, \mathbb{1}(\rho(a_i, x_i) \leq \tau) + \hat{\eta}(\pi^*),
$$

where $\tau \geq 0$ is a hyperparameter, $\hat{\mu}(a, x) \approx \mathbb{E}[Y \mid X = x, A = a]$ is the outcome model, and

$$
\hat{\eta}(\pi^*) = \frac{1}{n} \sum_{i=1}^{n} \sum_{a' \in \mathcal{A}} \hat{\mu}(a', x_i) \pi^*(a' \mid x_i) \approx \mathbb{E}_{\pi^*}[\hat{\mu}(A, X)]
$$

where $a_i^* \sim \pi^*(\cdot \mid x_i)$.

**Doubly Robust with Optimal Shrinkage (DRos)**   DRos proposed by [17] uses new weights $\hat{\rho}_\lambda(a_i, x_i)$ which directly minimises sharp bounds on the MSE of the resulting estimator,

$$
\hat{\theta}_{\mathrm{DRos}} := \frac{1}{n} \sum_{i=1}^{n} \hat{\rho}_\lambda(a_i, x_i)\, (y_i - \hat{\mu}(a_i, x_i)) + \hat{\eta}(\pi^*),
$$

where $\lambda \geq 0$ is a pre-defined hyperparameter and $\hat{\rho}_\lambda$ is defined as

$$
\hat{\rho}_\lambda(a, x) := \frac{\lambda}{\rho^2(a, x) + \lambda}\, \rho(a, x).
$$

When $\lambda = 0$, $\hat{\rho}_\lambda(a, x) = 0$ leads to DM, whereas as $\lambda \to \infty$, $\hat{\rho}_\lambda(a, x) \to \rho(a, x)$ leading to DR.

More generally, both of these estimators can be written as follows:

$$\hat{\theta}^{\tilde{\rho}}_{\text{DR}} := \frac{1}{n} \sum_{i=1}^{n} \tilde{\rho}(a_i, x_i) \left( y_i - \hat{\mu}(a_i, x_i) \right) + \hat{\eta}(\pi^*).$$

Here, when $\tilde{\rho}(a, x) = \rho(a, x) \mathbb{1}(\rho(a_i, x_i) \leq \tau)$, we recover the Switch-DR estimator and likewise when $\tilde{\rho}(a, x) = \hat{\rho}_\lambda(a, x)$, we recover DRos.

## B.1 Variance comparison with the DR extensions

Next, we provide a theoretical result comparing the variance of the MR estimator with these DR extension methods.

**Proposition B.1.** *When the weights $w(y)$ are known exactly and the outcome model is exact, i.e., $\hat{\mu}(a, x) = \mu(a, x) = \mathbb{E}[Y \mid X = x, A = a]$ in the DR estimator $\hat{\theta}^{\tilde{\rho}}_{\text{DR}}$ defined above,*

$$\text{Var}_{\pi^b}[\hat{\theta}^{\tilde{\rho}}_{\text{DR}}] - \text{Var}_{\pi^b}[\hat{\theta}_{\text{MR}}] \geq \frac{1}{n} \mathbb{E}_{\pi^b} \left[ \text{Var}_{\pi^b} \left[ \rho(A, X) \mid Y \right] Y^2 - \text{Var}_{\pi^b} \left[ \rho(A, X)\mu(A, X) \mid X \right] \right] - \Delta,$$

*where $\Delta := \frac{1}{n} \mathbb{E}_{\pi^b} \left[ (\rho^2(A, X) - \tilde{\rho}^2(A, X)) \text{Var}[Y \mid X, A] \right]$.*

*Proof of Proposition B.1.* Using the fact that $\hat{\mu}(a, x) = \mu(a, x)$ and the law of total variance, we get that

$$
\begin{aligned}
n \, \text{Var}_{\pi^b}[\hat{\theta}^{\tilde{\rho}}_{\text{DR}}] &= \text{Var}_{\pi^b}[\tilde{\rho}(A, X) \left( Y - \hat{\mu}(A, X) \right) + \sum_{a' \in \mathcal{A}} \hat{\mu}(a', X)\pi^*(a' \mid X)] \\
&= \text{Var}_{\pi^b}[\tilde{\rho}(A, X) \left( Y - \hat{\mu}(A, X) \right) + \mathbb{E}_{\pi^*}[\hat{\mu}(A, X) \mid X]] \\
&= \text{Var}_{\pi^b}[\tilde{\rho}(A, X) \left( Y - \mu(A, X) \right) + \mathbb{E}_{\pi^*}[\mu(A, X) \mid X]] \\
&= \text{Var}_{\pi^b}[\mathbb{E}_{\pi^b}[\tilde{\rho}(A, X) \left( Y - \mu(A, X) \right) + \mathbb{E}_{\pi^*}[\mu(A, X) \mid X] \mid X, A]] \\
&\quad + \mathbb{E}_{\pi^b}[\text{Var}_{\pi^b}[\tilde{\rho}(A, X) \left( Y - \mu(A, X) \right) + \mathbb{E}_{\pi^*}[\mu(A, X) \mid X] \mid X, A]] \\
&= \text{Var}_{\pi^b}[\mathbb{E}_{\pi^*}[\mu(A, X) \mid X]] + \mathbb{E}_{\pi^b}[\tilde{\rho}^2(A, X)\text{Var}[Y \mid X, A]] \\
&= \text{Var}_{\pi^b}[\mathbb{E}_{\pi^*}[\mu(A, X) \mid X]] + \mathbb{E}_{\pi^b}[\rho^2(A, X) \, \text{Var}[Y \mid X, A]] \\
&\quad + \underbrace{\mathbb{E}_{\pi^b}[(\tilde{\rho}^2(A, X) - \rho^2(A, X)) \, \text{Var}[Y \mid X, A]]}_{-n\,\Delta} \\
&= \text{Var}_{\pi^b}[\mathbb{E}_{\pi^b}[\rho(A, X)\,\mu(A, X) \mid X]] + \mathbb{E}_{\pi^b}[\rho^2(A, X) \, \text{Var}[Y \mid X, A]] - n\,\Delta.
\end{aligned}
$$

Again, using the law of total variance we can rewrite the second term on the RHS above as,

$$
\begin{aligned}
\mathbb{E}_{\pi^b}&[\rho^2(A, X) \, \text{Var}[Y \mid X, A]] \\
&= \text{Var}_{\pi^b}[\rho(A, X)\,Y] - \text{Var}_{\pi^b}[\rho(A, X)\,\mu(A, X)] \\
&= \text{Var}_{\pi^b}[\mathbb{E}_{\pi^b}[\rho(A, X) \mid Y]\,Y] + \mathbb{E}_{\pi^b}[\text{Var}_{\pi^b}[\rho(A, X) \mid Y]\,Y^2] \\
&\quad - \text{Var}_{\pi^b}[\rho(A, X)\,\mu(A, X)] \\
&= \text{Var}_{\pi^b}[w(Y)\,Y] + \mathbb{E}_{\pi^b}[\text{Var}_{\pi^b}[\rho(A, X) \mid Y]\,Y^2] - \text{Var}_{\pi^b}[\rho(A, X)\,\mu(A, X)] \\
&= n\, \text{Var}_{\pi^b}[\hat{\theta}_{\text{MR}}] + \mathbb{E}_{\pi^b}[\text{Var}_{\pi^b}[\rho(A, X) \mid Y]\,Y^2] - \text{Var}_{\pi^b}[\rho(A, X)\,\mu(A, X)].
\end{aligned}
$$

Putting this together, we get that

$$
\begin{aligned}
n\, \text{Var}_{\pi^b}&[\hat{\theta}^{\tilde{\rho}}_{\text{DR}}] \\
&= n\, \text{Var}_{\pi^b}[\hat{\theta}_{\text{MR}}] + \mathbb{E}_{\pi^b}[\text{Var}_{\pi^b}[\rho(A, X) \mid Y]\,Y^2] - \text{Var}_{\pi^b}[\rho(A, X)\,\mu(A, X)] \\
&\quad + \text{Var}_{\pi^b}[\mathbb{E}_{\pi^b}[\rho(A, X)\,\mu(A, X) \mid X]] - n\,\Delta \\
&= n\, \text{Var}_{\pi^b}[\hat{\theta}_{\text{MR}}] + \mathbb{E}_{\pi^b}[\text{Var}_{\pi^b}[\rho(A, X) \mid Y]\,Y^2] - \mathbb{E}_{\pi^b}[\text{Var}_{\pi^b}[\rho(A, X)\,\mu(A, X) \mid X]] - n\,\Delta,
\end{aligned}
$$

where in the last step above, we again use the law of total variance. Rearranging the above leads us to the result. $\square$

**Intuition** Note that for both of the DR extensions under consideration, the modified ratios $\tilde{\rho}(a,x)$ satisfy $0 \le \tilde{\rho}(a,x) \le \rho(a,x)$ and hence $\Delta \ge 0$ (using the definition of $\Delta$ in Proposition B.1). When the modified ratios $\tilde{\rho}(a,x)$ are 'close' to the true policy ratios $\rho(a,x)$, then using the definition of $\Delta$, we have that $\Delta \approx 0$. In this case, the result above provides a similar intuition to Proposition 3.4 in the main text. Specifically, in this case we have that if $\mathrm{Var}_{\pi^b}\left[\rho(A,X)\,Y \mid Y\right]$ is greater than $\mathrm{Var}_{\pi^b}\left[\rho(A,X)\,\mu(A,X) \mid X\right]$ on average, the variance of the MR estimator will be less than that of the DR extension under consideration. Intuitively, this will occur when the dimension of context space $\mathcal{X}$ is high because in this case the conditional variance over $X$ and $A$, $\mathrm{Var}_{\pi^b}\left[\rho(A,X)\,Y \mid Y\right]$ is likely to be greater than the conditional variance over $A$, $\mathrm{Var}_{\pi^b}\left[\rho(A,X)\,\mu(A,X) \mid X\right]$.

In contrast if the modified ratios $\tilde{\rho}(a,x)$ differ substantially from $\rho(a,x)$, then $\Delta$ will be large and the variance of MR may be higher than that of the resulting DR extension. However, this comes at the cost of significantly higher bias in the DR extension and consequently MSE of the DR extension will be high in this case.

# C   Weight estimation error

In this section, we theoretically investigate the effects of using the estimated importance weights $\hat{w}(y)$ rather than $\hat{\rho}(a,x)$ on the bias and variance of the resulting OPE estimator. Further to our discussion in Section 3.1.2, we focus in this section on the approximation error when using a wide neural network to estimate the weights $\hat{w}(y)$. To this end, we use recent results regarding the generalization of wide neural networks [19] to show that the estimation error of the approximation step (ii) in the Section 3.1.2 declines with increasing number of training data when $\hat{w}(y)$ is estimated using wide neural networks. Before providing the main result, we explicitly lay out the assumptions needed.

## C.1   Using wide neural networks to approximate the weights $\hat{w}(y)$

**Assumption C.1.** Let $\tilde{w}(y) := \mathbb{E}_{\pi^b}[\hat{\rho}(A,X) \mid Y=y]$. Suppose $\tilde{w} \in \mathcal{H}_1$ and $||\tilde{w}||_{\mathcal{H}_1} \le R$ for some constant $R$, where $\mathcal{H}_1$ is the reproducing kernel Hilbert space (RKHS) associated with the Neural Tangent Kernel $K_1$ associated with 2 layer neural network defined on $\mathbb{R}$.

**Assumption C.2.** There exists an $M \in [0,\infty)$ such that $\mathbb{P}_{\pi^b}(|Y| \le M) = 1$.

**Assumption C.3.** $\hat{\rho}(a_i,x_i)$ satisfies

$$\hat{\rho}(a_i,x_i) = \tilde{w}(y_i) + \eta_i,$$

where $\eta_i \overset{\text{iid}}{\sim} \mathcal{N}(0,\sigma^2)$ for some $\sigma > 0$.

**Theorem C.4.** *Suppose that the IPW and MR estimators are defined as,*

$$\tilde{\theta}_{\text{IPW}} := \frac{1}{n}\sum_{i=1}^{n} \hat{\rho}(a_i,x_i)\,y_i, \quad \text{and} \quad \tilde{\theta}_{\text{MR}} := \frac{1}{n}\sum_{i=1}^{n} \hat{w}_m(y_i)\,y_i,$$

*where $\hat{w}_m(y)$ is obtained by regressing to the estimated policy ratios $\hat{\rho}(a,x)$ using $m$ i.i.d. training samples $\mathcal{D}_{\text{tr}} := \{(x_i^{\text{tr}}, a_i^{\text{tr}}, y_i^{\text{tr}})\}_{i=1}^{m}$, i.e., by minimising the loss*

$$\mathcal{L}(\phi) = \mathbb{E}_{(X,A,Y)\sim\mathcal{D}_{\text{tr}}}\left[\left(\hat{\rho}(A,X) - f_\phi(Y)\right)^2\right].$$

*Suppose Assumptions C.1-C.3 hold, then for any given $\delta \in (0,1)$, if $f_\phi$ is a two-layer neural network with width $k$ that is sufficiently large and stops the gradient flow at time $t_* \propto m^{2/3}$, then for sufficiently large $m$, there exists a constant $C_1$ independent of $\delta$ and $m$, such that*

$$|\text{Bias}(\tilde{\theta}_{\text{MR}}) - \text{Bias}(\tilde{\theta}_{\text{IPW}})| \le C_1\, m^{-1/3} \log\frac{6}{\delta}$$

*holds with probability at least $(1-\delta)(1-o_k(1))$. Moreover, there exist constants $C_2, C_3$ independent of $\delta$ and $m$ such that*

$$n(\mathrm{Var}_{\pi^b}[\tilde{\theta}_{\text{IPW}}] - \mathrm{Var}_{\pi^b}[\tilde{\theta}_{\text{MR}}]) \ge \underbrace{\mathbb{E}_{\pi^b}[\mathrm{Var}_{\pi^b}[\hat{\rho}(A,X)\,Y \mid Y]]}_{\ge 0} - C_2\, m^{-2/3} \log^2\frac{6}{\delta} - C_3\, m^{-1/3}\log\frac{6}{\delta}$$

*holds with probability at least $(1-\delta)(1-o_k(1))$. Here, the randomness comes from the joint distribution of training samples and random initialization of parameters in the neural network $f_\phi$.*

*Proof of Theorem C.4.* The proof of this theorem relies on [19, Theorem 4.1]. Recall the definition $\tilde{w}(Y) := \mathbb{E}_{\pi^b}[\hat{\rho}(A, X) \mid Y]$. We can rewrite our setup in the setting of [19, Theorem 4.1], by relabelling $\hat{\rho}(a, x)$ in our setup as $y$ in their setup and relabelling $y$ in our setup as $x$ in their setup. Then, given $\delta \in (0, 1)$, from [19, Theorem 4.1], it follows that under Assumptions C.1-C.3 that there exists a constant $C$ independent of $\delta$ and $m$, such that

$$\mathbb{E}_{\pi^b}[\epsilon^2] \leq C\, m^{-2/3}\, \log^2 \frac{6}{\delta} \tag{5}$$

holds with probability at least $(1-\delta)(1-o_k(1))$, where $\epsilon := \hat{w}_m(Y) - \tilde{w}(Y)$. Recall from Proposition 3.7 that

$$|\text{Bias}(\tilde{\theta}_{\text{MR}}) - \text{Bias}(\tilde{\theta}_{\text{IPW}})| = |\mathbb{E}_{\pi^b}[\epsilon\, Y]|.$$

From this it follows using Cauchy-Schwarz inequality that,

$$|\text{Bias}(\tilde{\theta}_{\text{MR}}) - \text{Bias}(\tilde{\theta}_{\text{IPW}})| = |\mathbb{E}_{\pi^b}[\epsilon\, Y]| \leq \left(\mathbb{E}_{\pi^b}[\epsilon^2]\mathbb{E}_{\pi^b}[Y^2]\right)^{1/2}.$$

Combining the above with Eqn. (5), it follows that,

$$|\text{Bias}(\tilde{\theta}_{\text{MR}}) - \text{Bias}(\tilde{\theta}_{\text{IPW}})| \leq C^{1/2}\, m^{-1/3}\, \log \frac{6}{\delta}(\mathbb{E}_{\pi^b}[Y^2])^{1/2} = C_1\, m^{-1/3}\, \log \frac{6}{\delta}$$

holds with probability at least $(1 - \delta)(1 - o_k(1))$, where $C_1 = C^{1/2}\, (\mathbb{E}_{\pi^b}[Y^2])^{1/2}$.

Next, to prove the variance result, we recall from Proposition 3.7 that

$$n(\text{Var}_{\pi^b}[\tilde{\theta}_{\text{IPW}}] - \text{Var}_{\pi^b}[\tilde{\theta}_{\text{MR}}]) = \mathbb{E}_{\pi^b}[\text{Var}_{\pi^b}[\hat{\rho}(A, X) \mid Y]\, Y^2] - \text{Var}_{\pi^b}[\epsilon\, Y] - 2\, \text{Cov}(\epsilon\, Y, \tilde{w}(Y)\, Y)$$

Now note that, under Assumption C.2,

$$\text{Var}_{\pi^b}[\epsilon\, Y] \leq \mathbb{E}_{\pi^b}[(\epsilon\, Y)^2] \leq M^2 \mathbb{E}_{\pi^b}[\epsilon^2] \leq C\, M^2\, m^{-2/3}\, \log^2 \frac{6}{\delta} = C_2\, m^{-2/3}\, \log^2 \frac{6}{\delta},$$

holds with probability at least $(1 - \delta)(1 - o_k(1))$, where $C_2 = C\, M^2$. Similarly, we have that with probability at least $(1 - \delta)(1 - o_k(1))$,

$$
\begin{aligned}
|\text{Cov}(\epsilon\, Y, \tilde{w}(Y)\, Y)| &= |\mathbb{E}_{\pi^b}[\epsilon\, \tilde{w}(Y)\, Y^2] - \mathbb{E}_{\pi^b}[\epsilon\, Y]\mathbb{E}_{\pi^b}[\tilde{w}(Y)\, Y]| \\
&\leq |\mathbb{E}_{\pi^b}[\epsilon\, \tilde{w}(Y)\, Y^2]| + |\mathbb{E}_{\pi^b}[\epsilon\, Y]\mathbb{E}_{\pi^b}[\tilde{w}(Y)\, Y]| \\
&\leq \left(\mathbb{E}_{\pi^b}[\epsilon^2]\mathbb{E}_{\pi^b}[\tilde{w}(Y)^2\, Y^4]\right)^{1/2} + (\mathbb{E}_{\pi^b}[\epsilon^2]\mathbb{E}_{\pi^b}[Y^2])^{1/2}|\mathbb{E}_{\pi^b}[\tilde{w}(Y)\, Y]| \\
&= (\mathbb{E}_{\pi^b}[\epsilon^2])^{1/2} \left((\mathbb{E}_{\pi^b}[\tilde{w}(Y)^2\, Y^4])^{1/2} + (\mathbb{E}_{\pi^b}[Y^2])^{1/2}\, |\mathbb{E}_{\pi^b}[\tilde{w}(Y)\, Y]|\right) \\
&\leq C_3\, m^{-1/3}\, \log \frac{6}{\delta},
\end{aligned}
$$

where $C_3 = C\, (\mathbb{E}_{\pi^b}[\tilde{w}(Y)^2\, Y^4])^{1/2} + (\mathbb{E}_{\pi^b}[Y^2])^{1/2}\, |\mathbb{E}_{\pi^b}[\tilde{w}(Y)\, Y]|$, and we use Cauchy-Schwarz inequality in the third step above. Putting this together, we obtain the required result. $\qquad\square$

**Intuition** This theorem shows that as the number of training samples $m$ increases, the biases of MR and IPW estimators become roughly equal, whereas the variance of MR estimator falls below that of the IPW estimator. The empirical results shown in Appendix F.2 are consistent with this result. Moreover, in Theorem C.4, the estimated policy ratio $\hat{\rho}(a, x)$ is fixed for increasing $m$, i.e., we do not update $\hat{\rho}(a, x)$ as more training data becomes available. While this may seem as a disadvantage for the IPW estimator, we point out that the result also holds when the policy ratio is exact (i.e., $\hat{\rho}(a, x) = \rho(a, x)$) and hence the IPW estimator is unbiased.

**Relaxing Assumption C.3** [19][Theorem 4.1] suppose that the data has the relationship shown in Assumption C.3. However, the theorem relies on Corollary 4.4 in [43], which requires a strictly weaker assumption (Assumption 1 in [43]). Therefore, we can relax Assumption C.3 to the following assumption.

**Assumption C.5.** There exists positive constants $Q$ and $M$ such that for all $l \geq 2$ with $l \in \mathbb{N}$

$$\mathbb{E}_{\pi^b}[\hat{\rho}(A, X)^l \mid Y] \leq \frac{1}{2}\, l!\, M^{l-2}\, Q^2$$

$p_{\pi^b}$-almost surely.

It is easy to check that Assumption C.5 is strictly weaker than Assumption C.3, and is also satisfied if the policy ratio $\hat{\rho}(A, X)$ is almost surely bounded. For simplicity, we use the stronger assumption in our Proposition C.4.

## D  Generalised formulation of the MIPS estimator [14]

As described in Section 3.1.1, the MIPS estimator proposed by [14] assumes the existence of *action embeddings* $E$ which summarise all relevant information about the action $A$, and achieves a lower variance than the IPW estimator. To achieve this, the MIPS estimator only considers the shift in the distribution of $(X, E)$ as a result of policy shift, instead of considering the shift in $(X, A)$ (as in IPW estimator). In this section, we show that this idea can be generalised to instead consider general representations $R$ of the context-action pair $(X, A)$, which encapsulate all relevant information about the outcome $Y$. The MIPS estimator is a special case of this generalised setting where the representation $R$ is of the form $(X, E)$.

**Generalised MIPS (G-MIPS) estimator**   Suppose that there exists an embedding $R$ of the context-action pair $(X, A)$, with the Bayesian network shown in Figure 3. Here, $R$ may be a lower-dimensional representation of the $(X, A)$ pair which contains all the information necessary to predict the outcome $Y$. This corresponds to the following conditional independence assumption:

**Assumption D.1.** The context-action pair $(X, A)$ has no direct effect on the outcome $Y$ given $R$, i.e., $Y \perp\!\!\!\perp (X, A) \mid R$.

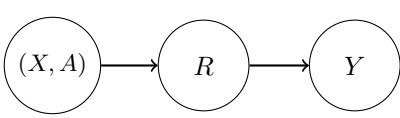

Figure 3: Bayesian network corresponding to Assumption D.1.

As illustrated in Figure 3, Assumption D.1 means that the embedding $R$ fully mediates every possible effect of $(X, A)$ on $Y$. The generalised MIPS estimator $\hat{\theta}_{\text{G-MIPS}}$ of target policy value, $\mathbb{E}_{\pi^*}[Y]$, is defined as

$$\hat{\theta}_{\text{G-MIPS}} := \frac{1}{n} \sum_{i=1}^{n} \frac{p_{\pi^*}(r_i)}{p_{\pi^b}(r_i)} \, y_i,$$

where $p_{\pi^b}(r)$ denote the density of $R$ under the behaviour policy (likewise for $p_{\pi^*}(r)$). Under assumption D.1, $\hat{\theta}_{\text{G-MIPS}}$ provides an unbiased estimator of target policy value. Similar to Lemma 3.1, the density ratio $\frac{p_{\pi^*}(r)}{p_{\pi^b}(r)}$ can be estimated by solving the regression problem

$$\arg \min_f \mathbb{E}_{\pi^b} \left( \frac{\pi^*(A \mid X)}{\pi^b(A \mid X)} - f(R) \right)^2. \tag{6}$$

### D.1  Variance reduction of G-MIPS estimator

By only considering the shift in the embedding $R$, the G-MIPS estimator achieves a lower variance relative to the vanilla IPW estimator. The following result, which is a straightforward extension of [14, Theorem 3.6], formalises this.

**Proposition D.2** (Variance reduction of G-MIPS). *When the ratios $\rho(a, x)$ and $\frac{p_{\pi^*}(r)}{p_{\pi^b}(r)}$ are known exactly then under Assumption D.1, we have that $\mathbb{E}_{\pi^b}[\hat{\theta}_{\text{IPW}}] = \mathbb{E}_{\pi^b}[\hat{\theta}_{\text{G-MIPS}}] = \mathbb{E}_{\pi^*}[Y]$. Moreover,*

$$\text{Var}_{\pi^b}[\hat{\theta}_{\text{IPW}}] - \text{Var}_{\pi^b}[\hat{\theta}_{\text{G-MIPS}}] \geq \frac{1}{n} \mathbb{E}_{\pi^b} \left[ \mathbb{E}[Y^2 \mid R] \text{Var}_{\pi^b}[\rho(A, X) \mid R] \right] \geq 0.$$

*Proof of Proposition D.2.* The following proof, which is included for completeness, is a straightforward extension of [14, Theorem 3.6].

$$n(\text{Var}_{\pi^b}[\hat{\theta}_{\text{IPW}}] - \text{Var}_{\pi^b}[\hat{\theta}_{\text{MIPS}}])$$

$$= \text{Var}_{\pi^b}\left[ \frac{\pi^*(A|X)}{\pi^b(A|X)} Y \right] - \text{Var}_{\pi^b}\left[ \frac{p_{\pi^*}(R)}{p_{\pi^b}(R)} Y \right]$$

$$= \text{Var}_{\pi^b}\left[ \mathbb{E}_{\pi^b}\left[ \frac{\pi^*(A|X)}{\pi^b(A|X)} Y \,\Big|\, R \right] \right] + \mathbb{E}_{\pi^b}\left[ \text{Var}_{\pi^b}\left[ \frac{\pi^*(A|X)}{\pi^b(A|X)} Y \,\Big|\, R \right] \right] - \text{Var}_{\pi^b}\left[ \mathbb{E}_{\pi^b}\left[ \frac{p_{\pi^*}(R)}{p_{\pi^b}(R)} Y \,\Big|\, R \right] \right]$$

$$- \mathbb{E}_{\pi^b}\left[ \text{Var}_{\pi^b}\left[ \frac{p_{\pi^*}(R)}{p_{\pi^b}(R)} Y \,\Big|\, R \right] \right]$$

Now using the conditional independence Assumption D.1, the first term on the RHS above becomes,

$$
\text{Var}_{\pi^b}\left[\mathbb{E}_{\pi^b}\left[\frac{\pi^*(A|X)}{\pi^b(A|X)}Y\middle|R\right]\right] = \text{Var}_{\pi^b}\left[\mathbb{E}_{\pi^b}\left[\frac{\pi^*(A|X)}{\pi^b(A|X)}\middle|R\right]\mathbb{E}_{\pi^b}\left[Y|R\right]\right]
$$

$$
= \text{Var}_{\pi^b}\left[\frac{p_{\pi^*}(R)}{p_{\pi^b}(R)}\mathbb{E}_{\pi^b}\left[Y|R\right]\right],
$$

where in the last step above we use the fact that

$$
\mathbb{E}_{\pi^b}\left[\frac{\pi^*(A|X)}{\pi^b(A|X)}\middle|R\right] = \frac{p_{\pi^*}(R)}{p_{\pi^b}(R)}.
$$

Putting this together, we get that

$$
n(\text{Var}_{\pi^b}[\hat{\theta}_{\text{IPW}}] - \text{Var}_{\pi^b}[\hat{\theta}_{\text{MIPS}}])
$$

$$
= \mathbb{E}_{\pi^b}\left[\text{Var}_{\pi^b}\left[\frac{\pi^*(A|X)}{\pi^b(A|X)}Y\middle|R\right]\right] - \mathbb{E}_{\pi^b}\left[\text{Var}_{\pi^b}\left[\frac{p_{\pi^*}(R)}{p_{\pi^b}(R)}Y\middle|R\right]\right]. \tag{7}
$$

Since we have that

$$
\mathbb{E}_{\pi^b}\left[\frac{\pi^*(A|X)}{\pi^b(A|X)}Y\middle|R\right] = \mathbb{E}_{\pi^b}\left[\frac{\pi^*(A|X)}{\pi^b(A|X)}\middle|R\right]\mathbb{E}_{\pi^b}\left[Y|R\right] = \frac{p_{\pi^*}(R)}{p_{\pi^b}(R)}\mathbb{E}_{\pi^b}\left[Y|R\right],
$$

Eq. (7) becomes,

$$
\mathbb{E}_{\pi^b}\left[\text{Var}_{\pi^b}\left[\frac{\pi^*(A|X)}{\pi^b(A|X)}Y\middle|R\right]\right] - \mathbb{E}_{\pi^b}\left[\text{Var}_{\pi^b}\left[\frac{p_{\pi^*}(R)}{p_{\pi^b}(R)}Y\middle|R\right]\right]
$$

$$
= \mathbb{E}_{\pi^b}\left[\mathbb{E}_{\pi^b}\left[\left(\frac{\pi^*(A|X)}{\pi^b(A|X)}Y\right)^2\middle|R\right] - \mathbb{E}_{\pi^b}\left[\left(\frac{p_{\pi^*}(R)}{p_{\pi^b}(R)}Y\right)^2\middle|R\right]\right]
$$

$$
= \mathbb{E}_{\pi^b}\left[\mathbb{E}_{\pi^b}\left[\left(\frac{\pi^*(A|X)}{\pi^b(A|X)}\right)^2\middle|R\right]\mathbb{E}_{\pi^b}\left[Y^2|R\right] - \left(\frac{p_{\pi^*}(R)}{p_{\pi^b}(R)}\right)^2\mathbb{E}_{\pi^b}\left[Y^2|R\right]\right]
$$

$$
= \mathbb{E}_{\pi^b}\left[\mathbb{E}_{\pi^b}\left[Y^2|R\right]\left(\mathbb{E}_{\pi^b}\left[\left(\frac{\pi^*(A|X)}{\pi^b(A|X)}\right)^2\middle|R\right] - \left(\mathbb{E}_{\pi^b}\left[\frac{\pi^*(A|X)}{\pi^b(A|X)}\middle|R\right]\right)^2\right)\right]
$$

$$
= \mathbb{E}_{\pi^b}\left[\mathbb{E}_{\pi^b}\left[Y^2|R\right]\text{Var}_{\pi^b}\left[\frac{\pi^*(A|X)}{\pi^b(A|X)}\middle|R\right]\right].
$$

$\square$

**Intuition** Here, $R$ contains all relevant information regarding the outcome $Y$. Moreover, intuitively $R$ can be thought of as the state obtained by 'filtering out' relevant information about $Y$ from $(X, A)$. Therefore, $R$ contains less 'redundant' information regarding the outcome $Y$ as compared to the covariate-action pair $(X, A)$. As a result, the G-MIPS estimator which only considers the shift in the marginal distribution of $R$ due to the policy shift is more efficient than the IPW estimator, which considers the shift in the joint distribution of $(X, A)$ instead. In fact, as the amount of 'redundant' information regarding $Y$ decreases in the embedding $R$, the G-MIPS estimator becomes increasingly efficient with decreasing variance. We formalise this as follows:

**Assumption D.3.** Assume there exist embeddings $R^{(1)}, R^{(2)}$ of the covariate-action pair $(X, A)$, with Bayesian network shown in Figure 4. This corresponds to the following conditional independence assumptions:

$$
R^{(2)} \perp\!\!\!\perp (X, A) \mid R^{(1)}, \qquad \text{and} \qquad Y \perp\!\!\!\perp (R^{(1)}, X, A) \mid R^{(2)}.
$$

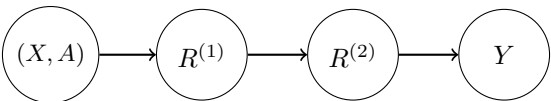

Figure 4: Bayesian network corresponding to Assumption D.3.

We can define G-MIPS estimators for these embeddings to obtain unbiased OPE estimators under Assumption D.3 as follows:

$$\hat{\theta}_{\text{G-MIPS}}^{(j)} := \frac{1}{n} \sum_{i=1}^{n} \frac{p_{\pi^*}(r_i^{(j)})}{p_{\pi^b}(r_i^{(j)})} y_i,$$

for $j \in \{1, 2\}$. Here, $\frac{p_{\pi^*}(r^{(j)})}{p_{\pi^b}(r^{(j)})}$ is the ratio of marginal densities of $R^{(j)}$ under target and behaviour policies. We next show that the variance of $\hat{\theta}_{\text{G-MIPS}}^{(j)}$ decreases with increasing $j$.

**Proposition D.4.** *When the ratios $\rho(a, x)$, $w(y)$ and $\frac{p_{\pi^*}(r^{(j)})}{p_{\pi^b}(r^{(j)})}$ are known exactly for $j \in \{1, 2\}$, then under Assumption D.3 we get that*

$$\mathbb{E}_{\pi^b}[\hat{\theta}_{\text{IPW}}] = \mathbb{E}_{\pi^b}[\hat{\theta}_{\text{G-MIPS}}^{(1)}] = \mathbb{E}_{\pi^b}[\hat{\theta}_{\text{G-MIPS}}^{(2)}] = \mathbb{E}_{\pi^b}[\hat{\theta}_{\text{MR}}] = \mathbb{E}_{\pi^*}[Y].$$

*Moreover,*

$$\text{Var}_{\pi^b}[\hat{\theta}_{\text{IPW}}] \geq \text{Var}_{\pi^b}[\hat{\theta}_{\text{G-MIPS}}^{(1)}] \geq \text{Var}_{\pi^b}[\hat{\theta}_{\text{G-MIPS}}^{(2)}] \geq \text{Var}_{\pi^b}[\hat{\theta}_{\text{MR}}].$$

*Proof of Proposition D.4.* First, we prove that the G-MIPS estimators are unbiased using induction on $j$. We define $R^{(0)} := (X, A)$ and $\hat{\theta}_{\text{G-MIPS}}^{(0)}$ defined as

$$\hat{\theta}_{\text{G-MIPS}}^{(0)} := \frac{1}{n} \sum_{i=1}^{n} \frac{p_{\pi^*}(r_i^{(0)})}{p_{\pi^b}(r_i^{(0)})} y_i,$$

recovers the IPW estimator $\hat{\theta}_{\text{IPW}}$. When $j = 0$, we know that $\hat{\theta}_{\text{G-MIPS}}^{(0)} = \hat{\theta}_{\text{IPW}}$ is unbiased. Now, assume that $\mathbb{E}_{\pi^b}[\hat{\theta}_{\text{G-MIPS}}^{(j)}] = \mathbb{E}_{\pi^*}[Y]$.

Conditional on $R^{(j)}$, $R^{(j+1)}$ does not depend on the policy. Therefore,

$$\frac{p_{\pi^*}(r^{(j)})}{p_{\pi^b}(r^{(j)})} = \frac{p_{\pi^*}(r^{(j)}) \, p(r^{(j+1)} \mid r^{(j)})}{p_{\pi^b}(r^{(j)}) \, p(r^{(j+1)} \mid r^{(j)})} = \frac{p_{\pi^*}(r^{(j)}, r^{(j+1)})}{p_{\pi^b}(r^{(j)}, r^{(j+1)})}.$$

And therefore,

$$\frac{p_{\pi^*}(r^{(j+1)})}{p_{\pi^b}(r^{(j+1)})} = \int_{r^{(j)}} \frac{p_{\pi^*}(r^{(j)}, r^{(j+1)})}{p_{\pi^b}(r^{(j)}, r^{(j+1)})} p_{\pi^b}(r^{(j)} \mid r^{(j+1)}) \, \mathrm{d}r^{(j)}$$

$$= \int_{r^{(j)}} \frac{p_{\pi^*}(r^{(j)})}{p_{\pi^b}(r^{(j)})} p_{\pi^b}(r^{(j)} \mid r^{(j+1)}) \, \mathrm{d}r^{(j)}$$

$$= \mathbb{E}_{\pi^b} \left[ \frac{p_{\pi^*}(R^{(j)})}{p_{\pi^b}(R^{(j)})} \, \middle| \, R^{(j+1)} = r^{(j+1)} \right].$$

Using this and the fact that $R^{(j)} \perp\!\!\!\perp Y \mid R^{(j+1)}$, we get that

$$\mathbb{E}_{\pi^b}\left[\hat{\theta}_{\text{G-MIPS}}^{(j+1)}\right] = \mathbb{E}_{\pi^b}\left[\frac{p_{\pi^*}(R^{(j+1)})}{p_{\pi^b}(R^{(j+1)})}Y\right]$$

$$= \mathbb{E}_{\pi^b}\left[\frac{p_{\pi^*}(R^{(j+1)})}{p_{\pi^b}(R^{(j+1)})}\mathbb{E}_{\pi^b}[Y|R^{(j+1)}]\right]$$

$$= \mathbb{E}_{\pi^b}\left[\mathbb{E}_{\pi^b}\left[\frac{p_{\pi^*}(R^{(j)})}{p_{\pi^b}(R^{(j)})}\middle|R^{(j+1)}\right]\mathbb{E}_{\pi^b}[Y|R^{(j+1)}]\right]$$

$$= \mathbb{E}_{\pi^b}\left[\mathbb{E}_{\pi^b}\left[\frac{p_{\pi^*}(R^{(j)})}{p_{\pi^b}(R^{(j)})}Y\middle|R^{(j+1)}\right]\right]$$

$$= \mathbb{E}_{\pi^b}\left[\frac{p_{\pi^*}(R^{(j)})}{p_{\pi^b}(R^{(j)})}Y\right]$$

$$= \mathbb{E}_{\pi^b}\left[\hat{\theta}_{\text{G-MIPS}}^{(j)}\right] = \mathbb{E}_{\pi^*}[Y].$$

Next, to prove the variance result we consider the difference

$$\text{Var}_{\pi^b}[\hat{\theta}_{\text{G-MIPS}}^{(j)}] - \text{Var}_{\pi^b}[\hat{\theta}_{\text{G-MIPS}}^{(j+1)}]$$

$$= \frac{1}{n}\left(\text{Var}_{\pi^b}\left[\frac{p_{\pi^*}(R^{(j)})}{p_{\pi^b}(R^{(j)})}Y\right] - \text{Var}_{\pi^b}\left[\frac{p_{\pi^*}(R^{(j+1)})}{p_{\pi^b}(R^{(j+1)})}Y\right]\right)$$

$$= \frac{1}{n}\left(\text{Var}_{\pi^b}\left[\mathbb{E}_{\pi^b}\left[\frac{p_{\pi^*}(R^{(j)})}{p_{\pi^b}(R^{(j)})}Y\middle|R^{(j+1)}\right]\right] + \mathbb{E}_{\pi^b}\left[\text{Var}_{\pi^b}\left[\frac{p_{\pi^*}(R^{(j)})}{p_{\pi^b}(R^{(j)})}Y\middle|R^{(j+1)}\right]\right]\right.$$

$$\left. - \text{Var}_{\pi^b}\left[\frac{p_{\pi^*}(R^{(j+1)})}{p_{\pi^b}(R^{(j+1)})}\mathbb{E}_{\pi^b}[Y \mid R^{(j+1)}]\right] - \mathbb{E}_{\pi^b}\left[\left(\frac{p_{\pi^*}(R^{(j+1)})}{p_{\pi^b}(R^{(j+1)})}\right)^2\text{Var}_{\pi^b}[Y \mid R^{(j+1)}]\right]\right)$$

where in the last step we use the law of total variance. Now, using the fact that $R^{(j)} \perp\!\!\!\perp Y \mid R^{(j+1)}$, we can rewrite the expression above as

$$= \frac{1}{n}\left(\text{Var}_{\pi^b}\left[\mathbb{E}_{\pi^b}\left[\frac{p_{\pi^*}(R^{(j)})}{p_{\pi^b}(R^{(j)})}\middle|R^{(j+1)}\right]\mathbb{E}_{\pi^b}[Y|R^{(j+1)}]\right] + \mathbb{E}_{\pi^b}\left[\text{Var}_{\pi^b}\left[\frac{p_{\pi^*}(R^{(j)})}{p_{\pi^b}(R^{(j)})}Y\middle|R^{(j+1)}\right]\right]\right.$$

$$\left. - \text{Var}_{\pi^b}\left[\frac{p_{\pi^*}(R^{(j+1)})}{p_{\pi^b}(R^{(j+1)})}\mathbb{E}_{\pi^b}[Y \mid R^{(j+1)}]\right] - \mathbb{E}_{\pi^b}\left[\left(\frac{p_{\pi^*}(R^{(j+1)})}{p_{\pi^b}(R^{(j+1)})}\right)^2\text{Var}_{\pi^b}[Y \mid R^{(j+1)}]\right]\right)$$

$$= \frac{1}{n}\left(\mathbb{E}_{\pi^b}\left[\text{Var}_{\pi^b}\left[\frac{p_{\pi^*}(R^{(j)})}{p_{\pi^b}(R^{(j)})}Y\middle|R^{(j+1)}\right]\right] - \mathbb{E}_{\pi^b}\left[\left(\frac{p_{\pi^*}(R^{(j+1)})}{p_{\pi^b}(R^{(j+1)})}\right)^2\text{Var}_{\pi^b}[Y \mid R^{(j+1)}]\right]\right).$$

Moreover, again using the conditional independence $R^{(j)} \perp\!\!\!\perp Y \mid R^{(j+1)}$, we can expand the first term in the expression above as follows:

$$\mathbb{E}_{\pi^b}\left[\text{Var}_{\pi^b}\left[\frac{p_{\pi^*}(R^{(j)})}{p_{\pi^b}(R^{(j)})}Y\middle|R^{(j+1)}\right]\right] = \mathbb{E}_{\pi^b}\left[\mathbb{E}_{\pi^b}\left[\frac{p_{\pi^*}^2(R^{(j)})}{p_{\pi^b}^2(R^{(j)})}\middle|R^{(j+1)}\right]\mathbb{E}_{\pi^b}[Y^2|R^{(j+1)}]\right.$$

$$\left. - \left(\mathbb{E}_{\pi^b}\left[\frac{p_{\pi^*}(R^{(j)})}{p_{\pi^b}(R^{(j)})}\middle|R^{(j+1)}\right]\mathbb{E}_{\pi^b}[Y|R^{(j+1)}]\right)^2\right]$$

$$\geq \mathbb{E}_{\pi^b}\left[\left(\mathbb{E}_{\pi^b}\left[\frac{p_{\pi^*}(R^{(j)})}{p_{\pi^b}(R^{(j)})}\middle|R^{(j+1)}\right]\right)^2\mathbb{E}_{\pi^b}[Y^2|R^{(j+1)}]\right.$$

$$\left. - \left(\frac{p_{\pi^*}(R^{(j+1)})}{p_{\pi^b}(R^{(j+1)})}\mathbb{E}_{\pi^b}[Y|R^{(j+1)}]\right)^2\right]$$

$$= \mathbb{E}_{\pi^b}\left[\left(\frac{p_{\pi^*}(R^{(j+1)})}{p_{\pi^b}(R^{(j+1)})}\right)^2\text{Var}_{\pi^b}[Y \mid R^{(j+1)}]\right].$$

Here, to get the inequality above, we use the fact that $\mathbb{E}[X^2] \geq (\mathbb{E}[X])^2$. Putting this together, we get that $\mathrm{Var}_{\pi^b}[\hat{\theta}_{\text{G-MIPS}}^{(j)}] - \mathrm{Var}_{\pi^b}[\hat{\theta}_{\text{G-MIPS}}^{(j+1)}] \geq 0$.

Moreover, the result $\mathrm{Var}_{\pi^b}[\hat{\theta}_{\text{G-MIPS}}^{(2)}] \geq \mathrm{Var}_{\pi^b}[\hat{\theta}_{\text{MR}}]$ follows straightforwardly from above by defining $R^{(3)} := Y$. Then, the embeddings satisfy the causal structure

$$R^{(0)} \to R^{(1)} \to R^{(2)} \to R^{(3)} \to Y.$$

Using the result above, we know that $\mathrm{Var}_{\pi^b}[\hat{\theta}_{\text{G-MIPS}}^{(2)}] \geq \mathrm{Var}_{\pi^b}[\hat{\theta}_{\text{G-MIPS}}^{(3)}]$. But now it is straightforward to see that $\hat{\theta}_{\text{G-MIPS}}^{(3)} = \hat{\theta}_{\text{MR}}$, and the result follows. $\qquad\square$

**Intuition** Here, $R^{(j+1)}$ can be thought of as the embedding obtained by 'filtering out' relevant information about $Y$ from $R^{(j)}$. As such, the amount of 'redundant' information regarding the outcome $Y$ decreases successively along the sequence $R^{(0)}(:= (X, A)), R^{(1)}, R^{(2)}$. As a result, the G-MIPS estimators which only consider the shift in the marginal distributions of $R^{(j)}$ due to policy shift become increasingly efficient with decreasing variance as $j$ increases. Define the representation $R^{(3)} := Y$, then the corresponding G-MIPS estimator reduces to the MR estimator, i.e., $\hat{\theta}_{\text{G-MIPS}}^{(3)} = \hat{\theta}_{\text{MR}}$. Moreover, this estimator has minimum variance among all the G-MIPS estimators $\{\hat{\theta}_{\text{G-MIPS}}^{(j)}\}_{0 \leq j \leq k}$, as the representation $R^{(3)}$ contains precisely the least amount of information necessary to obtain the outcome $Y$. In other words, $Y$ itself serves as the 'best embedding' of covariate-action pair $R^{(0)}$ which contains all relevant information regarding $Y$. We verify this empirically in Appendix F.2 by reproducing the experimental setup in [14] along with the MR baseline. Additionally, the MR estimator does not rely on assumptions like D.1 for unbiasedness.

In addition to this, solving the regression problem in Eq. (6) will typically be more difficult when $R$ is higher dimensional (as is likely to be the case for many choices of embeddings $R$), leading to high bias. In contrast, for MR the embedding $R = Y$ is one dimensional and therefore the regression problem is significantly easier to solve and yields lower bias. Our empirical results in Appendix F confirm this.

### D.2 Doubly robust G-MIPS estimators

Consider the setup for the G-MIPS estimator shown in Figure 3. In this case, we can derive a doubly robust extension of the G-MIPS estimator, denoted as GM-DR, which uses an estimate of the conditional mean $\tilde{\mu}(r) \approx \mathbb{E}[Y \mid R = r]$ as a control variate to decrease the variance of G-MIPS estimator. This can be explicitly written as follows:

$$\tilde{\theta}_{\text{DM-DR}} := \frac{1}{n} \sum_{i=1}^{n} \frac{p_{\pi^*}(r_i)}{p_{\pi^b}(r_i)} \left(y_i - \tilde{\mu}(r_i)\right) + \tilde{\eta}(\pi^*). \tag{8}$$

where $\tilde{\eta}(\pi^*) = \frac{1}{n} \sum_{i=1}^{n} \sum_{r' \in \mathcal{R}} \tilde{\mu}(r') \, p_{\pi^*}(r' \mid x_i)$ is the analogue of the direct method. Here, $\mathcal{R}$ denotes the space of the possible of the representations $R^2$. Moreover, given the density $p(r \mid x, a)$, we can compute $p_{\pi^*}(r \mid x)$ using

$$p_{\pi^*}(r \mid x) = \sum_{a' \in \mathcal{A}} p(r \mid x, a') \, \pi^*(a' \mid x).$$

It is straightforward to extend ideas from [13] to show that estimator $\tilde{\theta}_{\text{DM-DR}}$ is doubly robust in that it will yield accurate value estimates if either the importance weights $\frac{p_{\pi^*}(r)}{p_{\pi^b}(r)}$ or the outcome model $\tilde{\mu}(r)$ is well estimated.

**There is no analogous DR extension of the MR estimator** A consequence of considering the embedding $R = Y$ (as in MR) is that in this case we do not have an analogous doubly robust extension as above. To see why this is the case, note that when $R = Y$, we get that $\tilde{\mu}(r) = \mathbb{E}[Y \mid R = r] = \mathbb{E}[Y \mid Y = y] = y$. If we substitute this $\tilde{\mu}(r)$ in (8), we are simply left with $\tilde{\eta}(\pi^*)$ on the right hand side (as the first term cancels out). This means that the resulting estimator does not retain the doubly robust nature as we no longer obtain an accurate estimate if either the outcome model or the importance ratios are well estimated.

---

[2] the $\sum_{r' \in \mathcal{R}}$ can be replaced with $\int_{r' \in \mathcal{R}} \mathrm{d}r'$ when $\mathcal{R}$ is continuous

# E   Application to causal inference

In this section, we investigate the application of the MR estimator for the estimation of average treatment effect (ATE). In this setting, we suppose that $\mathcal{A} = \{0, 1\}$, and the goal is to estimate ATE defined as follows:

$$\text{ATE} \coloneqq \mathbb{E}[Y(1) - Y(0)]$$

Here, we use the potential outcomes notation [44] to denote the outcome under a deterministic policy $\pi^*(a' \mid x) = \mathbb{1}(a' = a)$ as $Y(a)$.

Specifically, the IPW estimator applied to ATE estimation yields:

$$\widehat{\text{ATE}}_{\text{IPW}} = \frac{1}{n} \sum_{i=1}^{n} \rho_{\text{ATE}}(a_i, x_i) \times y_i,$$

where

$$\rho_{\text{ATE}}(a, x) \coloneqq \frac{\mathbb{1}(a = 1) - \mathbb{1}(a = 0)}{\pi^b(a|x)}.$$

Similarly, the MR estimator can be written as

$$\widehat{\text{ATE}}_{\text{MR}} = \frac{1}{n} \sum_{i=1}^{n} w_{\text{ATE}}(y_i) \times y_i,$$

where

$$w_{\text{ATE}}(y) = \frac{p_{\pi^{(1)}}(y) - p_{\pi^{(0)}}(y)}{p_{\pi^b}(y)},$$

and $\pi^{(a)}(a' \mid x) \coloneqq \mathbb{1}(a' = a)$ for $a \in \{0, 1\}$.

Again, using the fact that $w_{\text{ATE}}(Y) \overset{\text{a.s.}}{=} \mathbb{E}[\rho_{\text{ATE}}(A, X) \mid Y]$, we can obtain $w_{\text{ATE}}$ by minimising a simple mean-squared loss:

$$w_{\text{ATE}} = \arg \min_f \mathbb{E}_{\pi^b} \left[ \frac{\mathbb{1}(A = 1) - \mathbb{1}(A = 0)}{\pi^b(A|X)} - f(Y) \right]^2.$$

**Proposition E.1** (Variance comparison with IPW ATE estimator). *When the weights $\rho_{\text{ATE}}(a, x)$ and $w_{\text{ATE}}(y)$ are known exactly, we have that* $\text{Var}[\widehat{\text{ATE}}_{\text{MR}}] \leq \text{Var}[\widehat{\text{ATE}}_{\text{IPW}}]$. *Specifically,*

$$\text{Var}[\widehat{\text{ATE}}_{\text{IPW}}] - \text{Var}[\widehat{\text{ATE}}_{\text{MR}}] = \frac{1}{n} \mathbb{E} \left[ \text{Var} \left[ \rho_{\text{ATE}}(A, X)|Y \right] Y^2 \right] \geq 0.$$

*Proof of Proposition E.1.* We have

$$\text{Var}[\widehat{\text{ATE}}_{\text{IPW}}] - \text{Var}[\widehat{\text{ATE}}_{\text{MR}}] = \frac{1}{n} \left( \text{Var}[\rho_{\text{ATE}}(A, X) Y] - \text{Var}[w_{\text{ATE}}(Y) Y] \right). \tag{9}$$

Using the tower law of variance, we get that

$$\begin{aligned}
\text{Var}[\rho_{\text{ATE}}(A, X) Y] &= \text{Var}[\mathbb{E}[\rho_{\text{ATE}}(A, X) Y \mid Y]] + \mathbb{E}[\text{Var}[\rho_{\text{ATE}}(A, X) Y \mid Y]] \\
&= \text{Var}[\mathbb{E}[\rho_{\text{ATE}}(A, X) \mid Y] Y] + \mathbb{E}[\text{Var}[\rho_{\text{ATE}}(A, X) \mid Y] Y^2] \\
&= \text{Var}[w_{\text{ATE}}(Y) Y] + \mathbb{E}[\text{Var}[\rho_{\text{ATE}}(A, X) \mid Y] Y^2].
\end{aligned}$$

Putting this together with (9) we obtain,

$$\text{Var}[\widehat{\text{ATE}}_{\text{IPW}}] - \text{Var}[\widehat{\text{ATE}}_{\text{MR}}] = \frac{1}{n} \mathbb{E}[\text{Var}[\rho_{\text{ATE}}(A, X) \mid Y] Y^2],$$

which straightforwardly leads to the result.   $\square$

Given the above definitions, the IPW estimator for $\mathbb{E}[Y(a)]$ would only consider datapoints with $A = a$, as it weights the samples using the policy ratios $\mathbb{1}(A = a)/\pi^b(A|X)$ which are only non-zero when $A = a$. This is however not the case with the MR estimator, as it uses the weights $p_{\pi^*}(Y)/p_{\pi^b}(Y)$ which are not necessarily zero for $A \neq a$. Therefore, MR uses all evaluation

datapoints $\mathcal{D}$ when estimating $\mathbb{E}[Y(a)]$. The MR estimator therefore leads to a more efficient use of evaluation data in this example.

Likewise, the doubly robust (DR) estimator applied to ATE estimation yields,

$$\widehat{\text{ATE}}_{\text{DR}} := \frac{1}{n}\sum_{i=1}^{n} \rho_{\text{ATE}}(a_i, x_i)\,(y_i - \hat{\mu}(a_i, x_i)) + \frac{1}{n}\sum_{i=1}^{n}(\hat{\mu}(1, x_i) - \hat{\mu}(0, x_i)),$$

where $\hat{\mu}(a, x) \approx \mathbb{E}[Y \mid X = x, A = a]$. Like in classical off-policy evaluation, DR yields an accurate estimator of ATE when either the weights $\rho_{\text{ATE}}(a, x)$ or the outcome model i.e., $\hat{\mu}(a, x) = \mathbb{E}[Y \mid X = x, A = a]$, are well estimated. However, despite this doubly robust nature of the estimator, we can show that the variance of the DR estimator may be higher than that of the MR estimator in many cases. The following result formalises this variance comparison between the DR and MR estimators, and is analogous to the result in Proposition 3.4 derived for classical off-policy evaluation.

**Proposition E.2** (Variance comparison with DR ATE estimator)**.** *When the weights $\rho_{\text{ATE}}(a, x)$ and $w_{\text{ATE}}(y)$ are known exactly,*

$$\text{Var}[\widehat{\text{ATE}}_{\text{DR}}] - \text{Var}[\widehat{\text{ATE}}_{\text{MR}}] \geq \frac{1}{n}\mathbb{E}\left[\text{Var}\left[\rho_{\text{ATE}}(A, X)\,Y \mid Y\right] - \text{Var}\left[\rho_{\text{ATE}}(A, X)\mu(A, X) \mid X\right]\right],$$

*where $\mu(A, X) := \mathbb{E}[Y \mid X, A]$.*

*Proof of Proposition E.2.* Using the law of total variance, we get that

$$
\begin{aligned}
n\,\text{Var}[\widehat{\text{ATE}}_{\text{DR}}] &= \text{Var}[\rho_{\text{ATE}}(A, X)\,(Y - \hat{\mu}(A, X)) + (\hat{\mu}(1, X) - \hat{\mu}(0, X))] \\
&= \text{Var}[\mathbb{E}[\rho_{\text{ATE}}(A, X)\,(Y - \hat{\mu}(A, X)) + (\hat{\mu}(1, X) - \hat{\mu}(0, X)) \mid X, A]] \\
&\quad + \mathbb{E}[\text{Var}[\rho_{\text{ATE}}(A, X)\,(Y - \hat{\mu}(A, X)) + (\hat{\mu}(1, X) - \hat{\mu}(0, X)) \mid X, A]] \\
&= \text{Var}[\rho_{\text{ATE}}(A, X)\,(\mu(A, X) - \hat{\mu}(A, X)) + (\hat{\mu}(1, X) - \hat{\mu}(0, X))] \\
&\quad + \mathbb{E}[\rho_{\text{ATE}}^2(A, X)\text{Var}[Y \mid X, A]].
\end{aligned}
$$

Again, using the law of total variance we can rewrite the first term on the RHS above as,

$$
\begin{aligned}
&\text{Var}[\rho_{\text{ATE}}(A, X)\,(\mu(A, X) - \hat{\mu}(A, X)) + (\hat{\mu}(1, X) - \hat{\mu}(0, X))] \\
&= \text{Var}[\mathbb{E}[\rho_{\text{ATE}}(A, X)\,(\mu(A, X) - \hat{\mu}(A, X)) + (\hat{\mu}(1, X) - \hat{\mu}(0, X)) \mid X]] \\
&\quad + \mathbb{E}[\text{Var}[\rho_{\text{ATE}}(A, X)\,(\mu(A, X) - \hat{\mu}(A, X)) + (\hat{\mu}(1, X) - \hat{\mu}(0, X)) \mid X]] \\
&\geq \text{Var}[\mathbb{E}[\rho_{\text{ATE}}(A, X)\,(\mu(A, X) - \hat{\mu}(A, X)) + (\hat{\mu}(1, X) - \hat{\mu}(0, X)) \mid X]] \\
&= \text{Var}[\mathbb{E}[\rho_{\text{ATE}}(A, X)\,(\mu(A, X) - \hat{\mu}(A, X)) + \rho_{\text{ATE}}(A, X)\,\hat{\mu}(A, X) \mid X]] \\
&= \text{Var}[\mathbb{E}[\rho_{\text{ATE}}(A, X)\,\mu(A, X) \mid X]],
\end{aligned}
$$

where, in the second last step above we use the fact that

$$\mathbb{E}[\rho_{\text{ATE}}(A, X)\,\hat{\mu}(A, X) \mid X] = \hat{\mu}(1, X) - \hat{\mu}(0, X).$$

Putting this together, we get that

$$n\,\text{Var}[\widehat{\text{ATE}}_{\text{DR}}] \geq \text{Var}[\mathbb{E}[\rho_{\text{ATE}}(A, X)\,\mu(A, X) \mid X]] + \mathbb{E}[\rho_{\text{ATE}}^2(A, X)\text{Var}[Y \mid X, A]].$$

Therefore,

$$n \left( \mathrm{Var}[\widehat{\mathrm{ATE}}_{\mathrm{DR}}] - \mathrm{Var}[\widehat{\mathrm{ATE}}_{\mathrm{MR}}] \right)$$

$$\geq \mathrm{Var}[\mathbb{E}[\rho_{\mathrm{ATE}}(A,X)\,\mu(A,X) \mid X]] + \mathbb{E}[\rho_{\mathrm{ATE}}^2(A,X)\mathrm{Var}[Y \mid X,A]] - \mathrm{Var}[w_{\mathrm{ATE}}(Y)\,Y]$$

$$= \mathrm{Var}[\mathbb{E}[\rho_{\mathrm{ATE}}(A,X)\,\mu(A,X) \mid X]] + \mathbb{E}[\mathrm{Var}[\rho_{\mathrm{ATE}}(A,X)\,Y \mid X,A]] - \mathrm{Var}[w_{\mathrm{ATE}}(Y)\,Y]$$

$$= \mathrm{Var}[\mathbb{E}[\rho_{\mathrm{ATE}}(A,X)\,\mu(A,X) \mid X]] + \mathrm{Var}[\rho_{\mathrm{ATE}}(A,X)\,Y] - \mathrm{Var}[\mathbb{E}[\rho_{\mathrm{ATE}}(A,X)\,Y \mid X,A]]$$
$$\quad - \mathrm{Var}[w_{\mathrm{ATE}}(Y)\,Y]$$

$$= \mathrm{Var}[\mathbb{E}[\rho_{\mathrm{ATE}}(A,X)\,\mu(A,X) \mid X]] + \mathrm{Var}[\mathbb{E}[\rho_{\mathrm{ATE}}(A,X) \mid Y]\,Y] + \mathbb{E}[\mathrm{Var}[\rho_{\mathrm{ATE}}(A,X) \mid Y]\,Y^2]$$
$$\quad - \mathrm{Var}[\mathbb{E}[\rho_{\mathrm{ATE}}(A,X)\,Y \mid X,A]] - \mathrm{Var}[w_{\mathrm{ATE}}(Y)\,Y]$$

$$= \mathrm{Var}[\mathbb{E}[\rho_{\mathrm{ATE}}(A,X)\,\mu(A,X) \mid X]] + \mathrm{Var}[w_{\mathrm{ATE}}(Y)\,Y] + \mathbb{E}[\mathrm{Var}[\rho_{\mathrm{ATE}}(A,X) \mid Y]\,Y^2]$$
$$\quad - \mathrm{Var}[\mathbb{E}[\rho_{\mathrm{ATE}}(A,X)\,Y \mid X,A]] - \mathrm{Var}[w_{\mathrm{ATE}}(Y)\,Y]$$

$$= \mathrm{Var}[\mathbb{E}[\rho_{\mathrm{ATE}}(A,X)\,\mu(A,X) \mid X]] - \mathrm{Var}[\mathbb{E}[\rho_{\mathrm{ATE}}(A,X)\,Y \mid X,A]] + \mathbb{E}[\mathrm{Var}[\rho_{\mathrm{ATE}}(A,X) \mid Y]\,Y^2]$$

$$= \mathrm{Var}[\rho_{\mathrm{ATE}}(A,X)\,\mu(A,X)] - \mathbb{E}[\mathrm{Var}[\rho_{\mathrm{ATE}}(A,X)\,\mu(A,X) \mid X]] - \mathrm{Var}[\rho_{\mathrm{ATE}}(A,X)\,\mu(A,X)]$$
$$\quad + \mathbb{E}[\mathrm{Var}[\rho_{\mathrm{ATE}}(A,X) \mid Y]\,Y^2]$$

$$= \mathbb{E}\left[ \mathrm{Var}\left[\rho_{\mathrm{ATE}}(A,X) \mid Y\right] Y^2 - \mathrm{Var}\left[\rho_{\mathrm{ATE}}(A,X)\mu(A,X) \mid X\right] \right].$$

$\square$

Proposition E.2 shows that if $\mathrm{Var}\left[Y\,\rho_{\mathrm{ATE}}(A,X) \mid Y\right]$ is greater than $\mathrm{Var}\left[\rho_{\mathrm{ATE}}(A,X)\mu(A,X) \mid X\right]$ on average, the variance of the MR estimator will be less than that of the DR estimator. Intuitively, this is likely to happen when the dimension of context space $\mathcal{X}$ is high because in this case, the conditional variance over $X$ and $A$, $\mathrm{Var}\left[Y\,\rho_{\mathrm{ATE}}(A,X) \mid Y\right]$ is likely to be greater than the conditional variance over $A$, $\mathrm{Var}\left[\rho_{\mathrm{ATE}}(A,X)\mu(A,X) \mid X\right]$.

## F  Experimental Results

In this section, we provide additional experimental details for the results presented in the main text. We also include extensive experimental results to provide further empirical evidence in favour of the MR estimator.

**Computational details**  We ran our experiments on Intel(R) Xeon(R) CPU E5-2690 v4 @ 2.60GHz with 8GB RAM per core. We were able to use 150 CPUs in parallel to iterate over different configurations and seeds. However, we would like to note that for each run our algorithms only requires 1 CPU and at most 30 minutes to run as our neural networks are relatively small. Throughout our experiments, whenever the outcome $Y$ was continuous, we used a fully connected neural network with three hidden layers with 512, 256 and 32 nodes respectively (and ReLU activation function) to estimate the weights $\hat{w}(y)$. On the other hand, when the outcome is discrete we can directly estimate $\hat{w}(y) \approx \mathbb{E}[\hat{\rho}(A,X) \mid Y = y]$ by calculating the sample mean of $\hat{\rho}(A,X)$ on samples with $Y = y$. Additionally, for each configuration of parameters in our experiments, we ran experiments for 10 different seeds (in $\{0, 1, \ldots, 9\}$).

### F.1  Alternative methodology of estimating MR

In addition to the OPE baselines like IPW, DM and DR estimators considered in the main text, we also include empirically investigate an alternative methodology of estimating MR. Below we describe this methodology, denoted as 'MR (alt)', in greater detail:

#### F.1.1  MR (alt)

Recall our definition of MR estimator:

$$\hat{\theta}_{\mathrm{MR}} := \frac{1}{n} \sum_{i=1}^{n} w(y_i)\,y_i.$$

In the main text, we propose estimating the weights $w(y)$ first and using this to estimate $\hat{\theta}_{\text{MR}}$ using the above expression. Alternatively, we can estimate $h(y) := y\,w(y)$ using

$$h = \arg\min_f \mathbb{E}_{\pi^b}\left[\left(Y\frac{\pi^*(A|X)}{\pi^b(A|X)} - f(Y)\right)^2\right].$$

Subsequently, the MR estimator can be written as:

$$\hat{\theta}_{\text{MR}} = \frac{1}{n}\sum_{i=1}^n h(y_i).$$

We refer to this alternative methodology as 'MR-alt' and compare it empirically against the original methodology (which we simply refer to as 'MR'). In general, it is difficult to say which of the two methods will perform better. Intuitively speaking, in cases where the behaviour of the quantity $Y\frac{\pi^*(A|X)}{\pi^b(A|X)}$ with varying $Y$ is 'smoother' than that of $\frac{\pi^*(A|X)}{\pi^b(A|X)}$, the alternative method is expected to perform better. Our empirical results in the next sections show that the relative performance of the two methods varies for different data generating mechanisms.

### F.2 Synthetic data experiments

Here, we include additional experimental details for the synthetic data experiments presented in Section 5.1 for completeness. For this experiment, we use the same setup as the synthetic data experiment in [14], reproduced by reusing their code with minor modifications.

**Setup** Here, we sample the $d$-dimensional context vectors $x$ from a standard normal distribution. The setup used also includes 3-dimensional categorical action embeddings $E \in \mathcal{E}$, which are sampled from the following conditional distribution given action $A = a$,

$$p(e \mid a) = \prod_{k=1}^3 \frac{\exp\left(\alpha_{a,e_k}\right)}{\sum_{e'\in\mathcal{E}_k}\exp\left(\alpha_{a,e'}\right)},$$

which is independent of the context $X$. $\{\alpha_{a,e_k}\}$ is a set of parameters sampled independently from the standard normal distribution. Each dimension of $\mathcal{E}$ has a cardinality of 10, i.e., $\mathcal{E}_k = \{1, 2, \ldots, 10\}$.

**Reward function** The expected reward is then defined as:

$$q(x, e) = \sum_{k=1}^3 \eta_k \cdot (x^T M\, x_{e_k} + \theta_x^T x + \theta_e^T x_{e_k}),$$

where $M$, $\theta_x$ and $\theta_e$ are parameter matrices or vectors sampled from a uniform distribution with range $[-1, 1]$. $x_{e_k}$ is a context vector corresponding to the $k$-th dimension of the action embedding, which is unobserved to the estimators. $\eta_k$ specifies the importance of the $k$-th dimension of the action embedding, sampled from Dirichlet distribution so that $\sum_{k=1}^3 \eta_k = 1$.

**Behaviour and target policies** The behaviour policy $\pi^b$ is defined by applying the softmax function to $q(x, a) = \mathbb{E}[q(X, E) \mid A = a, X = x]$ as

$$\pi^b(a \mid x) = \frac{\exp\left(-q(x, a)\right)}{\sum_{a'\in\mathcal{A}}\exp\left(-q(x, a')\right)}.$$

For the target policy, we define the class of parametric policies,

$$\pi^{\alpha^*}(a|x) = \alpha^* \mathbb{1}(a = \arg\max_{a'\in\mathcal{A}} q(x, a')) + \frac{1 - \alpha^*}{|\mathcal{A}|},$$

where $\alpha^* \in [0, 1]$ controls the shift between the behaviour and target policies. As shown in the main text, as $\alpha^* \to 1$, the shift between behaviour and target policies increases.

**Baselines**   In the main text, we compare MR with DM, IPW, DR and MIPS estimators. In addition to these baselines, here we also consider Switch-DR [5] and DR with Optimistic Shrinkage (DRos) [17]. Following [14], we use the random forest [45] along with 2-fold cross-fitting [46] to obtain $\hat{q}(x, e)$ for DR and DM methods. To estimate $p_{\pi^b}(a \mid x, e)$ for MIPS estimator, we use logistic regression. We also include the results for MR estimated using the alternative methodology described in Section F.1.1. We refer to this as 'MR (alt)'.

**Estimation of behaviour policy $\widehat{\pi}^b$ and marginal ratio $\hat{w}(y)$**   We do not assume that the true behaviour policy $\pi^b$ is known, and therefore estimate $\widehat{\pi}^b$ using the available training data. For the MR estimator, we estimate the behaviour policy using a random forest classifier trained on 50% of the training data and use the rest of the training data to estimate the marginal ratios $\hat{w}(y)$ using multi-layer perceptrons (MLP). Moreover, for a fair comparison we use a different behaviour policy estimate $\widehat{\pi}^b$ for all other baselines which is trained on the entire training data.

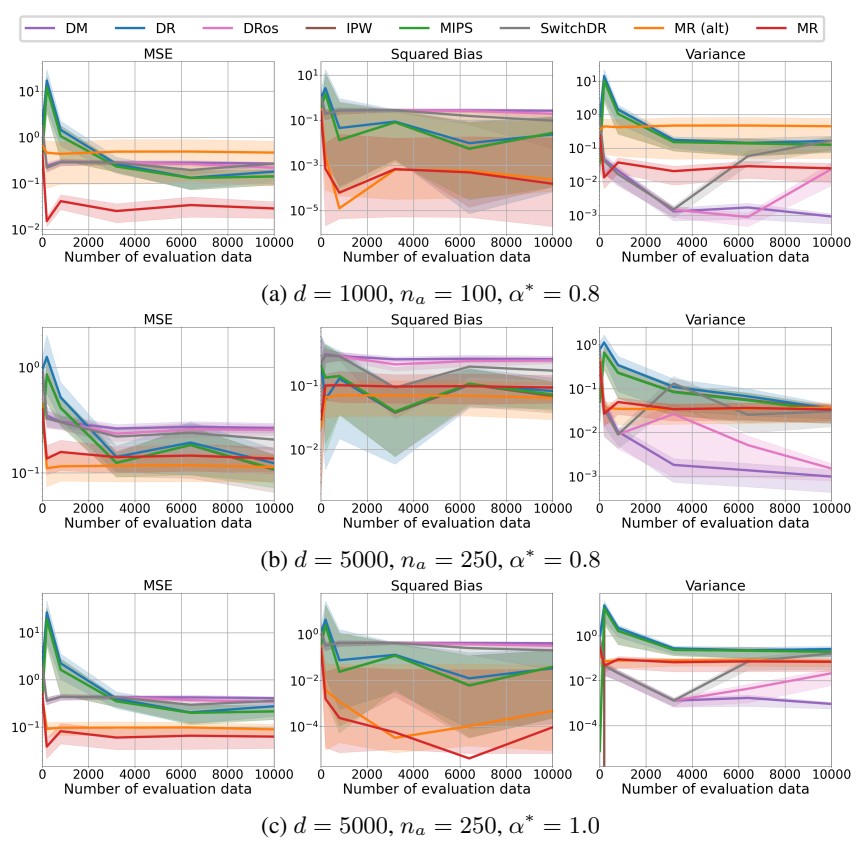

(a) $d = 1000, n_a = 100, \alpha^* = 0.8$

(b) $d = 5000, n_a = 250, \alpha^* = 0.8$

(c) $d = 5000, n_a = 250, \alpha^* = 1.0$

Figure 5: MSE with varying size of evaluation dataset $n$ for different choices of parameters.

### F.2.1   Results

For this experiment, the results are computed over 10 different sets of logged data replicated with different seeds, and in Figures 5 - 8 we use a total of $m = 5000$ training data.

**Varying size of evaluation data $n$**   Figure 5 shows that MR outperforms the other baselines, in terms of MSE and squared bias, when the number of evaluation data $n \leq 1000$. Additionally, we observe that in this experiment, MR estimated using our original methods ('MR'), yields better results than the alternative method of estimating MR ('MR (alt)'). Moreover, while the variance of DM is lower than that of MR, the DM method has a high bias and consequently a high MSE. We note that while the difference between MSE and variance of MIPS and MR estimators decreases with increasing evaluation data size, MR still outperforms MIPS in terms of both MSE and variance.

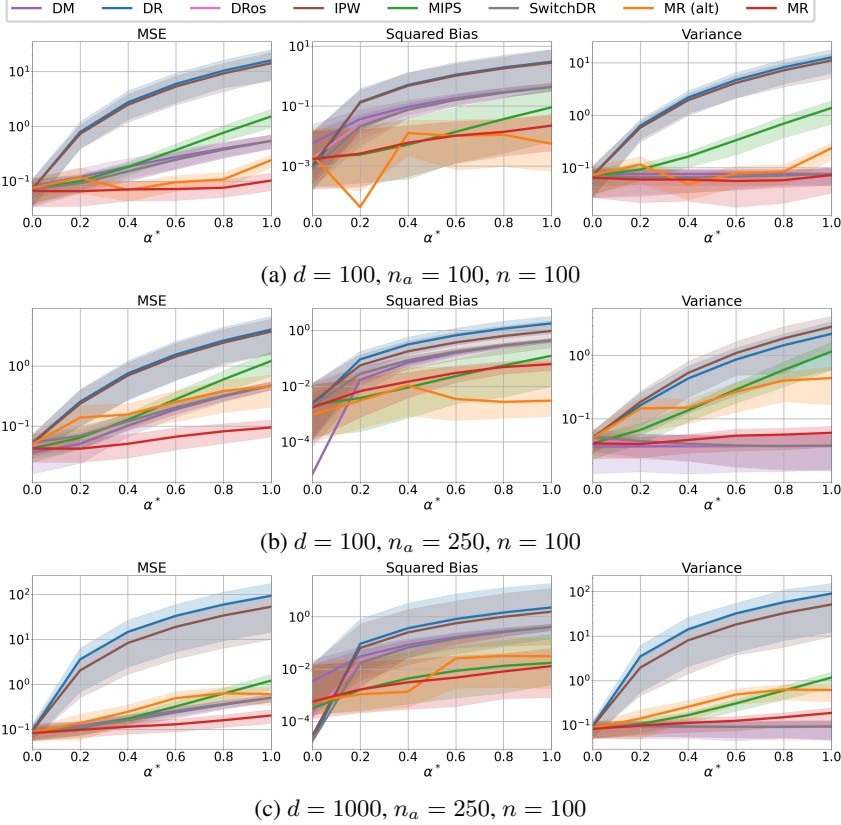

(a) $d = 100$, $n_a = 100$, $n = 100$

(b) $d = 100$, $n_a = 250$, $n = 100$

(c) $d = 1000$, $n_a = 250$, $n = 100$

Figure 6: MSE with varying $\alpha^*$ for different choices of parameters.

**Varying $\alpha^*$** Figure 6 shows the results with increasing policy shift. It can be seen that overall MR methods achieve the smallest MSE with increasing policy shift. Moreover, the difference between MSE and variance of MR and IPW/DR methods increases with increasing policy shift, showing that MR performs especially better than these baselines when the difference between behaviour and target policies is large. Similarly, we observe in Figure 6 that as the shift between the behaviour and target policy increases with increasing $\alpha^*$, so does the difference between the MSE and variance of MR and the MIPS estimators. This shows that generally MR outperforms MIPS estimator in terms of variance and MSE, and that MR performs especially better than MIPS as the difference between behaviour and target policies increases.

**Varying $d$ and $n_a$** Figures 7 and 8 show that MR outperforms the other baselines as the context dimensions and/or number of actions increase. In fact, these figures show that MR is significantly robust to increasing dimensions of action and context spaces, whereas baselines like IPW and DR perform poorly in large action spaces.

**Varying $m$** Figure 10 shows the results with increasing number of training data $m$. We again observe that the MR methods 'MR' and 'MR (alt)' outperforms the other baselines in terms of the MSE and squared bias even when the number of training data is low. Moreover, the variance of both the MR estimators continues to improve with increasing number of training data.

In this experiment, we observe that overall 'MR (alt)' performs worse than the original MR estimator ('MR' in the figures). However, as we observe in Appendix F.5, this does not happen consistently across all experiments, which suggests that the comparative performance of the two MR methods depends on the data generating mechanism.

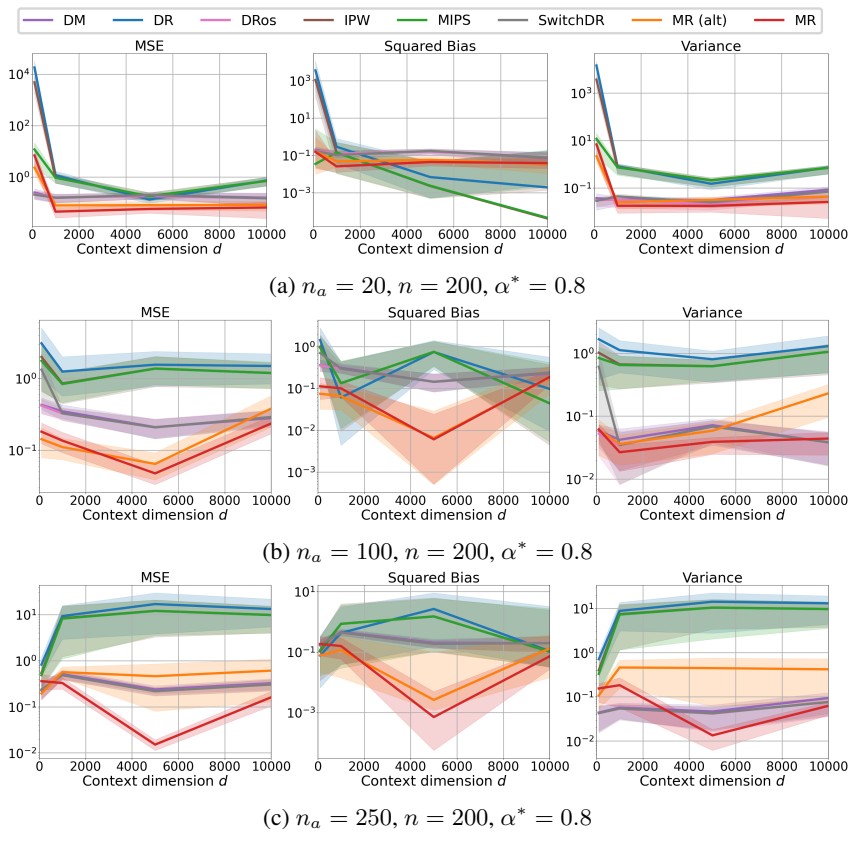

(a) $n_a = 20$, $n = 200$, $\alpha^* = 0.8$

(b) $n_a = 100$, $n = 200$, $\alpha^* = 0.8$

(c) $n_a = 250$, $n = 200$, $\alpha^* = 0.8$

Figure 7: MSE with varying context dimensions $d$ for different choices of parameters.

Table 3: Mean-squared error results with 2 standard errors for synthetic data setup considered in Section 5.1 with $d = 5000$, $n_a = 50$, $\alpha^* = 0.8$. We use a fixed budget of datapoints (denoted by $N$) for each baseline and in the case of MR we use $m = 2000$ of the available datapoints to estimate $\hat{w}(y)$ and the rest of data to evaluate the MR estimator (i.e. $n = N - 2000$ for MR). In contrast, for IPW and MIPS since the importance ratios are already known, we use all of the $N$ datapoints for evaluation of the off-policy value (i.e. $n = N$ for IPW and MIPS).

| | $N$ | 2800 | 3200 | 6400 | 10000 | 12000 |
|---|---|---|---|---|---|---|
| **GT weights $\rho(a, x)$ and estimated reward model $\hat{\mu}(a, x)$** ($m = 2000$ used for training $\hat{\mu}(a, x)$ and $n = N - 2000$ used for evaluation) | DM | 0.137±0.028 | 0.099±0.012 | 0.103±0.012 | 0.093±0.010 | 0.089±0.010 |
| | DR | 0.227±0.065 | 0.068±0.035 | 0.068±0.022 | **0.024±0.011** | 0.045±0.015 |
| | DRos | 0.128±0.027 | 0.072±0.011 | 0.049±0.014 | 0.063±0.014 | 0.051±0.016 |
| | SwitchDR | 0.128±0.027 | 0.059±0.014 | 0.052±0.013 | 0.061±0.015 | 0.056±0.016 |
| **GT weights** (all of $N$ datapoints are used for evaluation) | IPW | 0.237±0.062 | 0.066±0.036 | 0.067±0.021 | 0.025±0.011 | **0.044±0.014** |
| | MIPS | 0.236±0.062 | 0.065±0.035 | 0.067±0.021 | 0.025±0.011 | **0.044±0.014** |
| **Estimated weights $\hat{w}(y)$** ($m = 2000$ used for training and $n = N - 2000$ used for evaluation) | MR (Ours) | **0.045±0.015** | **0.042±0.014** | **0.048±0.020** | 0.049±0.020 | 0.047±0.016 |

### F.2.2 Known policy ratios $\rho(a, x)$

Our previous setting of unknown importance policy ratios $\rho(a, x)$ captures a wide variety of real-world applications, ranging from health care to autonomous driving. In addition, to demonstrate the utility of MR in settings with known $\rho(a, x), p(e \mid a, x)$ and unknown $w(y)$ (for our proposed method, MR), we have conducted additional experiments. Here, we use a fixed budget of datapoints (denoted by $N$) for each baseline and for MR we allocate $m = 2000$ of the available datapoints to estimate $\hat{w}(y)$ and use the remaining for evaluating the MR estimator (i.e., $n = N - 2000$ for MR). In contrast, for IPW and MIPS (since the importance ratios are already known), we use all of the $N$ datapoints to evaluate the off-policy value (i.e. $n = N$ for IPW and MIPS).

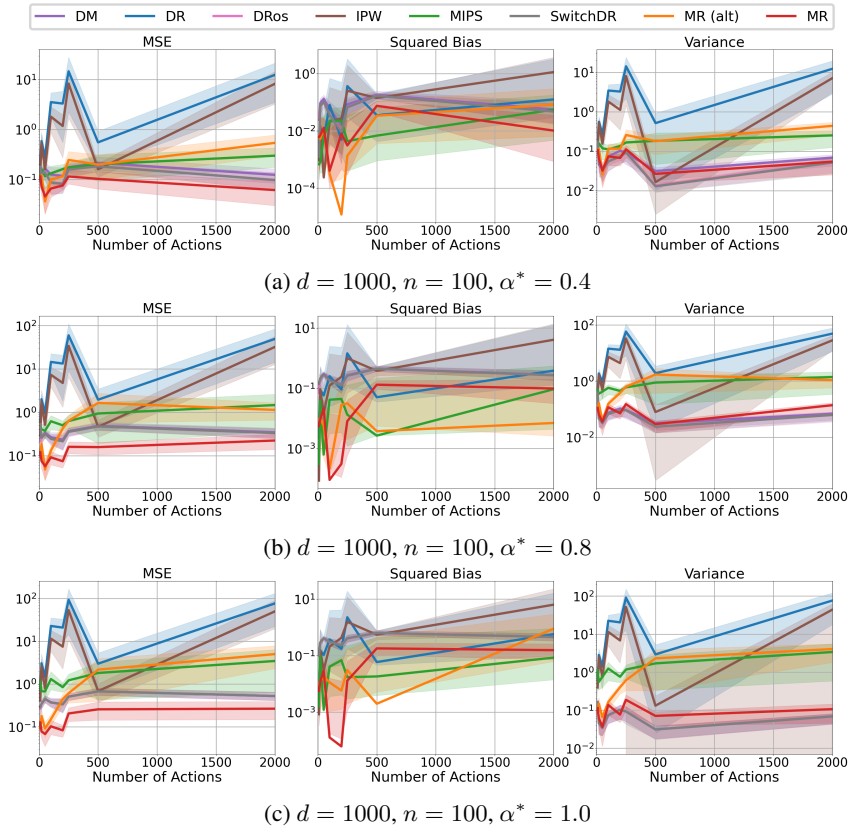

(a) $d = 1000$, $n = 100$, $\alpha^* = 0.4$

(b) $d = 1000$, $n = 100$, $\alpha^* = 0.8$

(c) $d = 1000$, $n = 100$, $\alpha^* = 1.0$

Figure 8: MSE with varying number of actions $n_a$ for different choices of parameters.

The results included in Table 3 show that MR achieves the smallest MSE among the baselines for $N \leq 6400$. However, we observe that the MSE of IPW, DR and MIPS (with true importance weights) falls below that of MR (with estimated weights $\hat{w}$) when the data size $N$ is large enough (i.e., $N \geq 10,000$). This is to be expected since IPW, DR and MIPS are unbiased (i.e., use ground truth importance ratios $\rho(a, x)$) whereas MR uses estimated weights $\hat{w}(y)$ (and hence may be biased). MR still performs the best when $N \leq 6400$.

### F.3 Experiments on classification datasets

Here, we conduct experiments on four classification datasets, OptDigits, PenDigits, SatImage and Letter datasets from the UCI repository [37], the Digits dataset from scikit-learn library, as well as the Mnist [38] and CIFAR-100 datasets [39].

**Setup**  Following previous works [13, 22, 36, 5], the classification datasets are transformed to contextual bandit feedback data. The classification dataset comprises $\{x_i, a_i^{\text{gt}}\}_{i=1}^{n_0}$, where $x_i \in \mathcal{X}$ are feature vectors and $a_i^{\text{gt}} \in \mathcal{A}$ are the ground-truth labels. In the contextual bandits setup, the feature vectors $x_i$ are considered to be the contexts, whereas the actions correspond to the possible class of labels. We split the dataset into training and testing datasets of sizes $m$ and $n$ respectively. We present the results for a range of different values of $m$ and $n$.

**Reward function**  Let $X$ be a context with ground truth label $A^{\text{gt}}$, we define the reward for action $A$ as:

$$Y := \mathbb{1}(A = A^{\text{gt}}).$$

**Behaviour and target policies**  Using the $m$ training datapoints, we first train a classifier $f : \mathcal{X} \to \mathbb{R}^{|\mathcal{A}|}$ which takes as input the feature vectors $x_i$ and outputs a vector of softmax probabilities over

labels, i.e. the $a$-th component of the vector $f(x)$, denoted as $(f(x))_a$ corresponds to the estimated probability $\mathbb{P}(A^{\text{gt}} = a \mid X = x)$.

Next, we use $f$ to define the ground truth behaviour policy,

$$\pi^b(a \mid x) = (f(x))_a.$$

For the target policies, we use $f$ to define a parametric class of target policies using a trained classifier $f : \mathcal{X} \to \mathbb{R}^{|\mathcal{A}|}$.

$$\pi^{\alpha^*}(a \mid x) = \alpha^* \cdot \mathbb{1}(a = \arg\max_{a' \in \mathcal{A}}(f(x))_{a'}) + \frac{1 - \alpha^*}{|\mathcal{A}|},$$

where $\alpha^* \in [0, 1]$. A value of $\alpha^*$ close to 1 leads to a near-deterministic and well-performing policy. As $\alpha^*$ decreases, the policy gets increasingly worse and 'noisy'. In this experiment, we consider target policies $\pi^* = \pi^{\alpha^*}$ for $\alpha^* \in \{0.0, 0.2, 0.4, \ldots, 1.0\}$.

Using the behaviour policy defined above, we generate the contextual bandits data described with training and evaluation datasets of sizes $m$ and $n$ respectively.

**Estimation of behaviour policy $\widehat{\pi}^b$ and marginal ratio $\hat{w}(y)$**   We do not assume that the behaviour policy $\pi^b$ is known, and therefore estimate it using training data. To estimate the behaviour policy $\widehat{\pi}^b$, we train a random forest classifier using the training data. This estimate of behaviour policy is used for all the baselines in our experiment. Since the reward is binary, we can estimate the marginal ratios $\hat{w}(y) = \mathbb{E}_{\pi^b}[\hat{\rho}(A, X) \mid Y = y]$ by directly estimating the sample mean of $\hat{\rho}(A, X)$ for datapoints with $Y = y$. We re-use the $m$ training datapoints to estimate this sample mean.

**Baselines**   We compare our estimator with Direct Method (DM), IPW and DR estimators. In addition, we also consider Switch-DR [5] and DR with Optimistic Shrinkage (DRos) [17]. To estimate $\hat{q}(x, a)$ for DM and DR estimators, we use random forest classifiers (since reward $Y$ is binary). Moreover, because of the binary nature of $Y$, the alternative method of estimating MR yields the same estimator as the original method, therefore we do not consider the two separately here. Additionally, in this experiment, we do not include MIPS (or G-MIPS) baseline, as there is no natural informative embedding $E$ of the action $A$.

### F.3.1   Results

For this experiment, we compute the results over 10 different sets of logged data replicated with different seeds. Figures 11 - 17 show the results corresponding to each baseline for the different datasets. It can be seen that across all datasets, the MR achieves the smallest MSE with increasing evaluation data size $n$. Moreover, across all datasets, MR attains the minimum MSE with relatively small number of evaluation data ($n \leq 100$).

Unlike the experiments in Section 5.1, we observe that the KL-divergence between target and behaviour policy decreases as $\alpha^*$ increases (see Figure 9). Therefore, as $\alpha^*$ increases the shift between target and behaviour policies decreases. Figures 11 - 16 show that as $\alpha^*$ increases, the difference between the MSE, squared bias and variance of MR and the other baselines decreases. This confirms our findings from earlier experiments that MR performs especially better than the other baselines when the difference between behaviour and target policies is large.

Moreover, the figures also include results with increasing number of training data $m$. It can be seen that MR out-performs the baselines even when the number of training data $m$ is small ($m = 100$). Moreover, the relative advantage of MR improves with increasing $m$.

### F.4   Application to Average Treatment Effect (ATE) estimation

In this subsection, we provide additional details for our experiment applying MR to the problem of ATE estimation presented in the main text. We begin by describing the dataset being used in this experiment.

**Twins dataset**   We use the Twins dataset as studied by [40], which comprises data from twin births in the USA between 1989-1991. The treatment $a = 1$ corresponds to being born the heavier twin and the outcome $Y$ corresponds to the mortality of each of the twins in their first year of life. Since

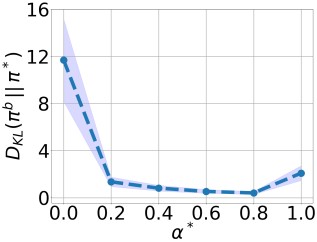

Figure 9: KL divergence $D_{\mathrm{KL}}(\pi^b \| \pi^*)$ with increasing $\alpha^*$ for the classification data experiments. Here, we only include the results for a specific choice of parameters for the Letter dataset. We observe similar results for other datasets and parameter choices.

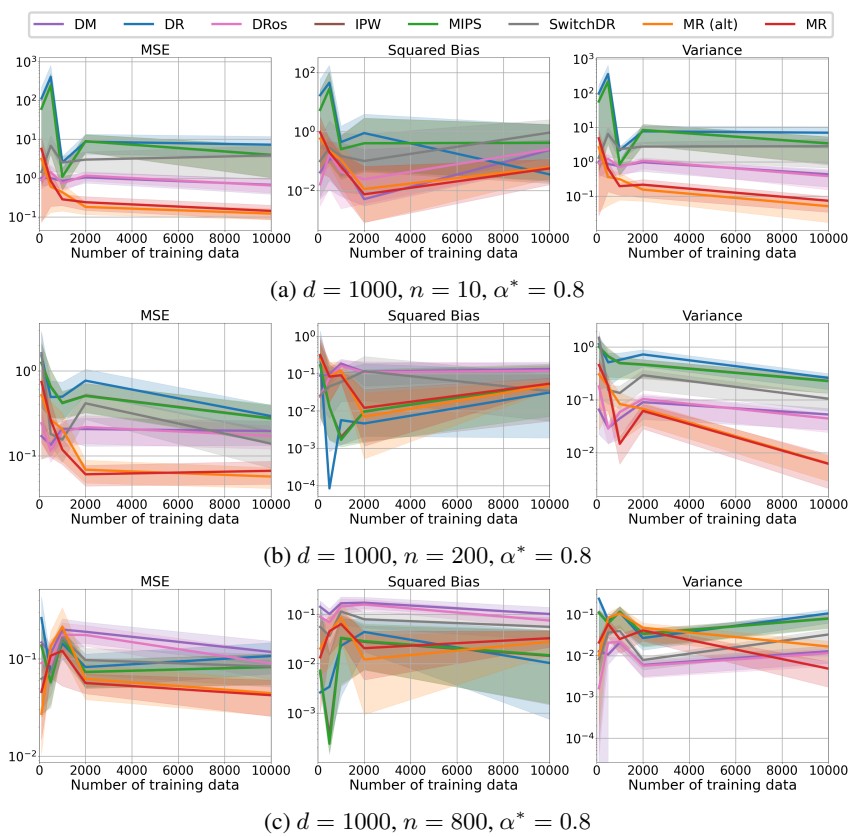

Figure 10: MSE with varying number of training data $m$ for different choices of parameters.

the data includes records for both twins, their outcomes would be considered as the two potential outcomes. Specifically, $Y(1)$ corresponds to the mortality of the heavier twin (and likewise for $Y(0)$). Closely following the methodology of [40], we only chose twins which are the same sex and weigh less than 2kgs. This provides us with a dataset of 11984 pairs of twins.

The mortality rate for the lighter twin is 18.9% and for the heavier twin is 16.4%, leading to the ATE value being $\theta_{\mathrm{ATE}} = -2.5\%$. For each twin-pair we obtained 46 covariates relating to the parents, the pregnancy and birth.

**Treatment assignment**   To simulate an observational study, we selectively hide one of the two twins by defining the treatment variable $A$ which depends on the feature *GESTAT10*. This feature, which takes integer values from 0 to 9, is obtained by grouping the number of gestation weeks prior to birth into 10 groups. Then we sample actions $A$ as follows,

$$A \mid X \sim \mathrm{Bern}(Z/10),$$

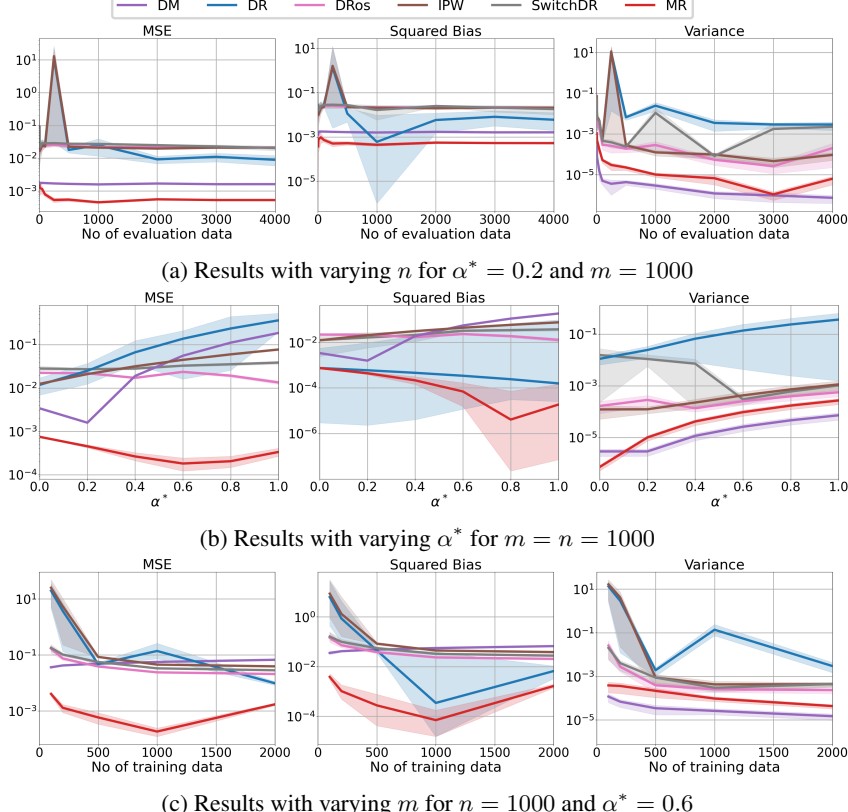

(a) Results with varying $n$ for $\alpha^* = 0.2$ and $m = 1000$

(b) Results with varying $\alpha^*$ for $m = n = 1000$

(c) Results with varying $m$ for $n = 1000$ and $\alpha^* = 0.6$

Figure 11: Results for OptDigits dataset

where $Z$ is *GESTAT10*, and $X$ are all the 46 features corresponding to a twin pair (including *GESTAT10*).

Using the treatment assignments defined above, we generate the observational data by selectively hiding one of the two twins from each pair. Next, we randomly split this dataset into training and evaluation datasets of sizes $m$ and $n$ respectively. In this experiment, we consider $m = 5000$ training datapoints.

**Baselines** Recall that ATE estimation can be formulated as the difference between off-policy values of deterministic policies $\pi^{(1)} := \mathbb{1}(A = 1)$ and $\pi^{(0)} := \mathbb{1}(A = 0)$. Therefore, any OPE estimator can be applied to ATE estimation. In this experiment, we compare our estimator against the baselines considered in our OPE experiments in Section F.3. This includes the Direct Method (DM), IPW and DR estimators as well as Switch-DR [5] and DR with Optimistic Shrinkage (DRos) [17]. To estimate $\hat{q}(x, a)$ for DM and DR estimators, we use multi-layer perceptrons (MLP) trained on the $m$ training datapoints. Additionally, we estimate the behaviour policy $\widehat{\pi}^b$ using random forest classifier trained on the full training dataset.

Since the outcome in this experiment is binary, we estimate the weights $w(y) = \mathbb{E}_{\pi^b}[\hat{\rho}(A, X) \mid Y = y]$ directly by estimating the sample mean of $\hat{\rho}(A, X)$ for datapoints with $Y = y$. This means that the alternative method of estimating MR yields the same value as the default method. We therefore do not consider these estimators separately. Additionally, since there is no natural embedding $R$ of the covariate-action space which satisfies the conditional dependence Assumption D.1, we do not consider the G-MIPS (or MIPS) estimator either.

**Performance metric** For our evaluation, we consider the absolute error in ATE estimation, $\epsilon_{\text{ATE}}$, defined as:

$$\epsilon_{\text{ATE}} := |\hat{\theta}_{\text{ATE}}^{(n)} - \theta_{\text{ATE}}|.$$

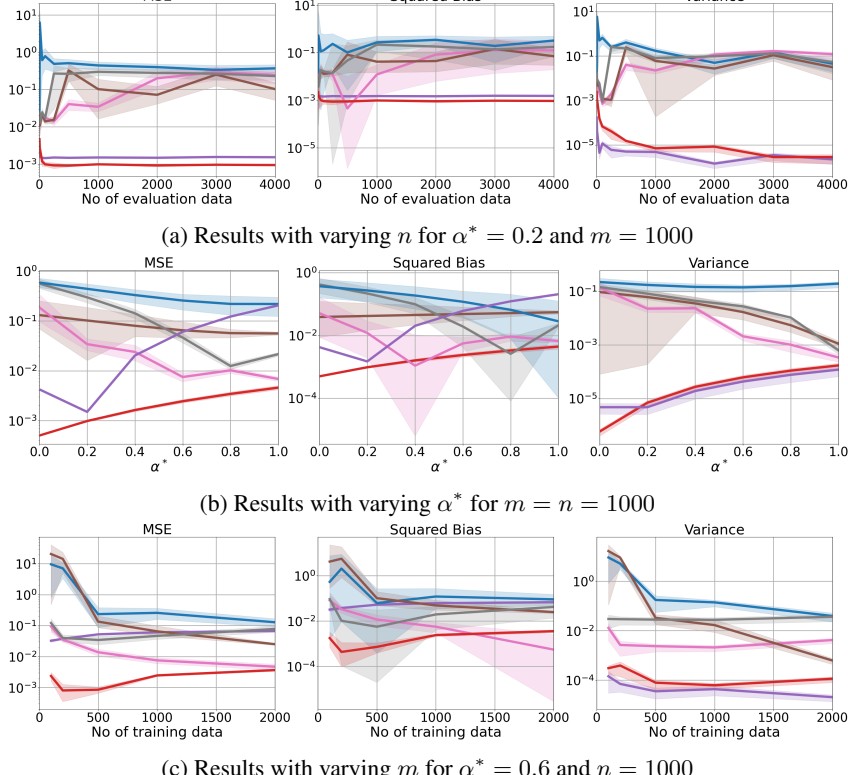

(a) Results with varying $n$ for $\alpha^* = 0.2$ and $m = 1000$

(b) Results with varying $\alpha^*$ for $m = n = 1000$

(c) Results with varying $m$ for $\alpha^* = 0.6$ and $n = 1000$

Figure 12: Results for PenDigits dataset

Here, $\hat{\theta}_{\text{ATE}}^{(n)}$ denotes the value of the ATE estimated using $n$ evaluation datapoints. For example, for the IPW estimator, the $\hat{\theta}_{\text{ATE}}^{(n)}$ can be written as:

$$\hat{\theta}_{\text{ATE}}^{(n)} = \widehat{\text{ATE}}_{\text{IPW}} = \frac{1}{n} \sum_{i=1}^{n} \left( \frac{\mathbb{1}(a_i = 1) - \mathbb{1}(a_i = 0)}{\hat{\pi}^b(a_i \mid x_i)} \right) y_i.$$

All results for this experiment are provided in the main text.

### F.5 Additional synthetic data experiments

In addition to the synthetic data experiments provided in Section 5.1, we also consider an additional synthetic data setup to obtain further empirical evidence in favour of the MR estimator, and also compare it against the generalised version of the MIPS estimator (described as G-MIPS in Appendix D). Here, we use a similar setup to [14] (albeit without action embeddings $E$) where the $d$-dimensional context vectors $x$ are sampled from a standard normal distribution. Likewise, the action space is finite and comprises of $n_a$ actions, i.e. $\mathcal{A} = \{0, \ldots, n_a - 1\}$, with $n_a$ taking a range of different values. The reward function is defined as follows:

**Reward function** The expected reward $q(x, a) := \mathbb{E}[Y \mid x, a]$ for these experiments is defined as follows:

$$q(x, a) = \sin\left(a \cdot ||x||_2\right).$$

The reward $Y$ is obtained by adding a normal noise random variable to $q(x, a)$

$$Y = q(X, A) + \epsilon,$$

where $\epsilon \sim \mathcal{N}(0, 0.01)$. Here, it can be seen that conditional on $R = (||X||_2, A)$, the reward $Y$ does not depend on $(X, A)$, i.e., the embedding $R$ satisfies the conditional independence assumption $Y \perp\!\!\!\perp (X, A) \mid R$.

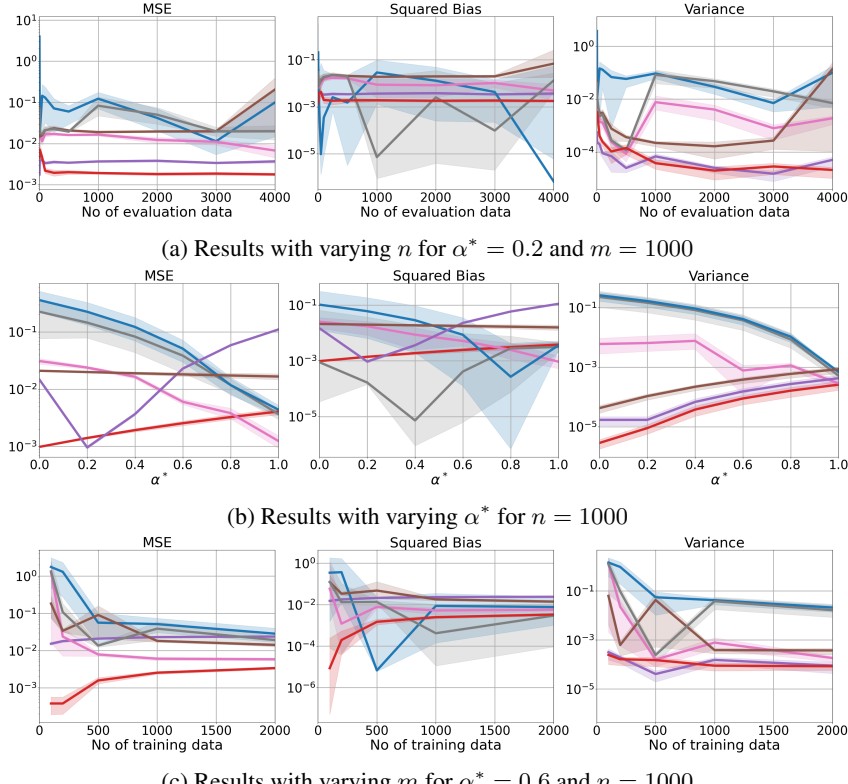

(a) Results with varying $n$ for $\alpha^* = 0.2$ and $m = 1000$

(b) Results with varying $\alpha^*$ for $n = 1000$

(c) Results with varying $m$ for $\alpha^* = 0.6$ and $n = 1000$

Figure 13: Results for SatImage dataset

**Behaviour and target policies** We first define a behaviour policy by applying softmax function to $q(x, a)$ as

$$\pi^b(a \mid x) = \frac{\exp\left(q(x,a)\right)}{\sum_{a' \in \mathcal{A}} \exp\left(q(x,a')\right)}.$$

Just like in Section 5.1, to investigate the effect of increasing policy shift, we define a class of policies,

$$\pi^{\alpha^*}(a|x) = \alpha^* \, \mathbb{1}(a = \arg\max_{a' \in \mathcal{A}} q(x,a')) + \frac{1 - \alpha^*}{|\mathcal{A}|} \quad \text{where} \quad q(x,a) := \mathbb{E}[Y \mid X = x, A = a],$$

where $\alpha^* \in [0, 1]$ allows us to control the shift between $\pi^b$ and $\pi^*$. Again, the shift between $\pi^b$ and $\pi^*$ increases as $\alpha^* \to 1$. Using the ground truth behaviour policy $\pi^b$, we generate a dataset which is split into training and evaluation datasets of sizes $m$ and $n$ respectively.

In Figures 18 - 21, we present the results for this experimental setup for different choices of paramater configurations.

**Estimation of behaviour policy $\widehat{\pi}^b$ and marginal ratio $\hat{w}(y)$** For the MR estimator, we estimate the behaviour policy using a random forest classifier trained on 50% of the training data and use the rest of the training data to estimate the marginal ratios $\hat{w}(y)$ using multi-layer perceptrons (MLP). Moreover, for a fair comparison we use a different behaviour policy estimate $\widehat{\pi}^b$ for all other baselines which is trained on the entire training data.

**Additional Baselines** In addition to the baselines considered in the main text (Section 5.1), we also consider Switch-DR [5] and DR with Optimistic Shrinkage (DRos) [17]. In addition, we also include the results for MR estimated using the alternative method ('MR (alt)') outlined in Section F.1.1. For

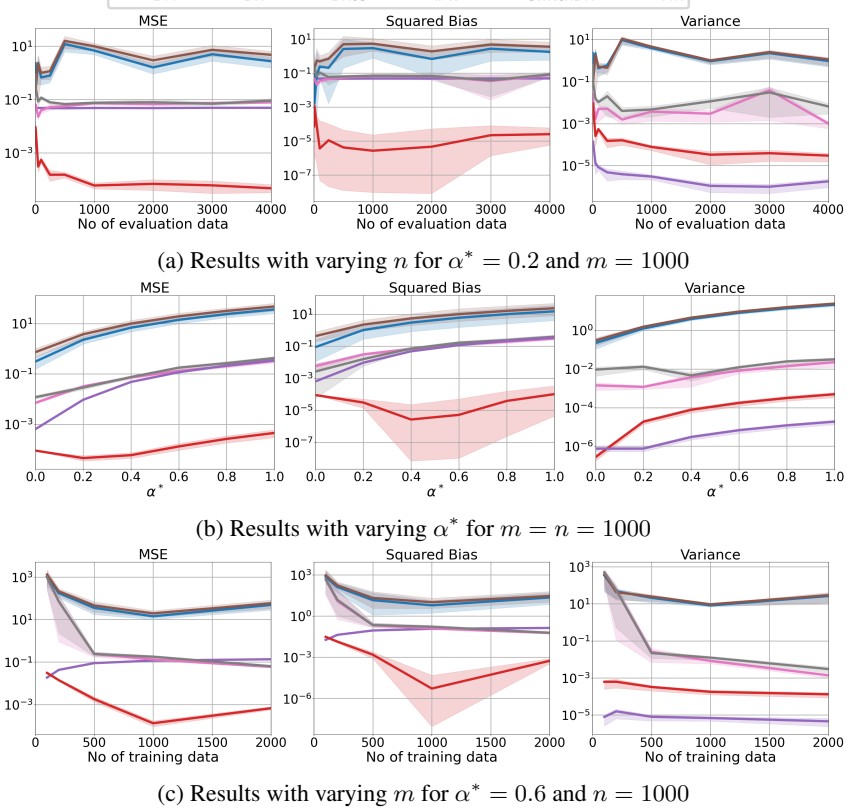

(a) Results with varying $n$ for $\alpha^* = 0.2$ and $m = 1000$

(b) Results with varying $\alpha^*$ for $m = n = 1000$

(c) Results with varying $m$ for $\alpha^* = 0.6$ and $n = 1000$

Figure 14: Results for Letter dataset

the G-MIPS estimator (defined in Appendix D) considered here, we use $R = (a, ||x||_2)^3$. To estimate $\hat{q}(x, a)$ for DM and DR estimators, we use multi-layer perceptrons (MLPs).

### F.5.1 Results

For this experiment, the results are computed over 10 different sets of logged data replicated with different seeds, and in Figures 18 - 21 we use a total of $m = 5000$ training data.

**Varying $n$** Figure 18 shows that MR outperforms the other baselines, in terms of MSE and squared bias, when the number of evaluation data $n \leq 1000$. Additionally, we observe that in this experiment, MR esitmated using alternative methods, MR (alt), yields better results than the original method of estimating MR. Moreover, while the variance of DM is lower than that of MR, the DM method has a high bias and consequently a high MSE.

**Varying $\alpha^*$** Figure 19 shows the results with increasing policy shift. It can be seen that overall MR methods achieve the smallest MSE with increasing policy shift. Moreover, the difference between MSE and variance of MR and IPW/DR methods increases with increasing policy shift, showing that MR performs especially better than these baselines when the difference between behaviour and target policies is large.

**Varying $d$ and $n_a$** Figures 20 and 21 show that MR outperforms the other baselines as the context dimensions and/or number of actions increase. In fact, Figure 21 shows that MR is significantly robust to increasing action space, whereas baselines like IPW and DR perform poorly in large action spaces.

---

[3]It is easy to see that in our setup, the embedding $R = (a, ||x||_2)$ satisfies the conditional independence assumption $Y \perp\!\!\!\perp (X, A) \mid R$ needed for G-MIPS estimator to be unbiased

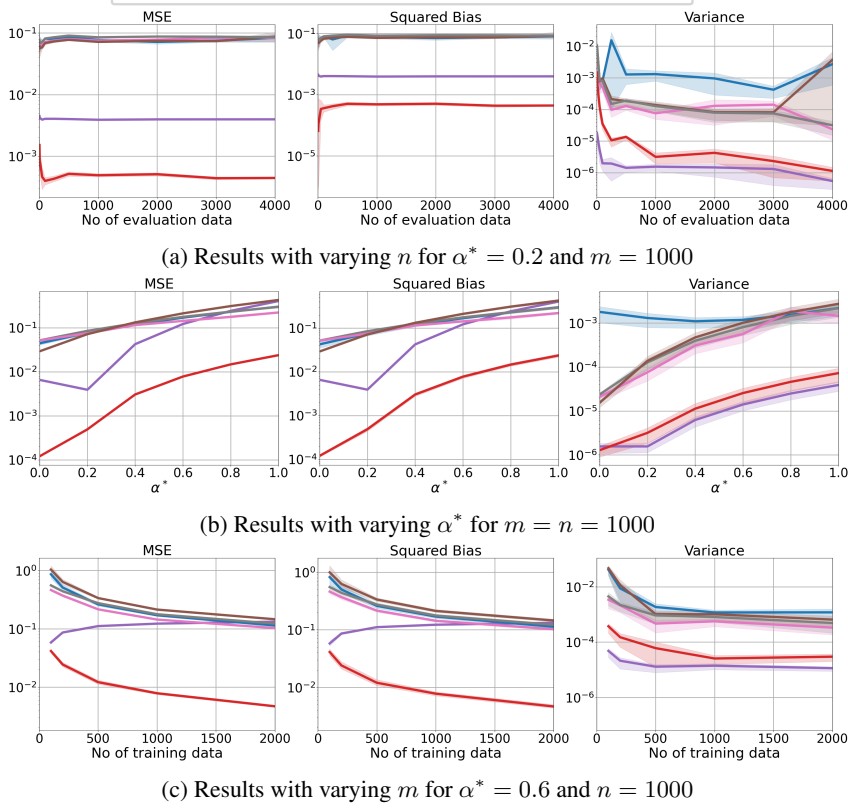

(a) Results with varying $n$ for $\alpha^* = 0.2$ and $m = 1000$

(b) Results with varying $\alpha^*$ for $m = n = 1000$

(c) Results with varying $m$ for $\alpha^* = 0.6$ and $n = 1000$

Figure 15: Results for Mnist dataset

**Varying $m$**  Figure 22 shows the results with increasing number of training data $m$. We again observe that the MR methods 'MR' and 'MR (alt)' outperforms the other baselines in terms of the MSE and squared bias even when the number of training data is low. Moreover, the variance of both the MR estimators continues to improve with increasing number of training data.

Unlike our experimental results in Section F.2, 'MR (alt)' performs better than the original MR estimator overall. This shows that one of these two methods is not better than the other consistently in all cases, and their relative performance depends on the dataset under consideration.

### F.6   Self-normalised MR estimator

Self-normalization trick has been used in practice to reduce the variance in off-policy estimators [30]. This technique is also applicable to the MR estimator, and leads to the self-normalized MR estimator (denoted as $\theta_{\text{SNMR}}$) defined as follows:

$$\theta_{\text{SNMR}} := \sum_{i=1}^{n} \frac{w(Y_i)}{\sum_{j=1}^{n} w(Y_j)}\, Y_i.$$

We conducted experiments to investigate the effect of self-normalisation on the performance of the IPW, DR and MR estimators. Figure 23 shows results for three different choices of parameter configurations. Overall, we observe that in all settings, the MR and self-normalised MR (SNMR) estimator outperform all other baselines including the self-normalised IPW and DR estimators (denoted as SNIPW and SNDR respectively). Moreover, in some settings, where the importance ratios achieve very high values, self-normalisation can reduce the variance and MSE of the corresponding estimator (for example, Figure 23b). However, we also observe cases in which self-normalization does not significantly change the results (Figure 23a), or may even slightly worsen the MSE of the estimators (Figure 23c).

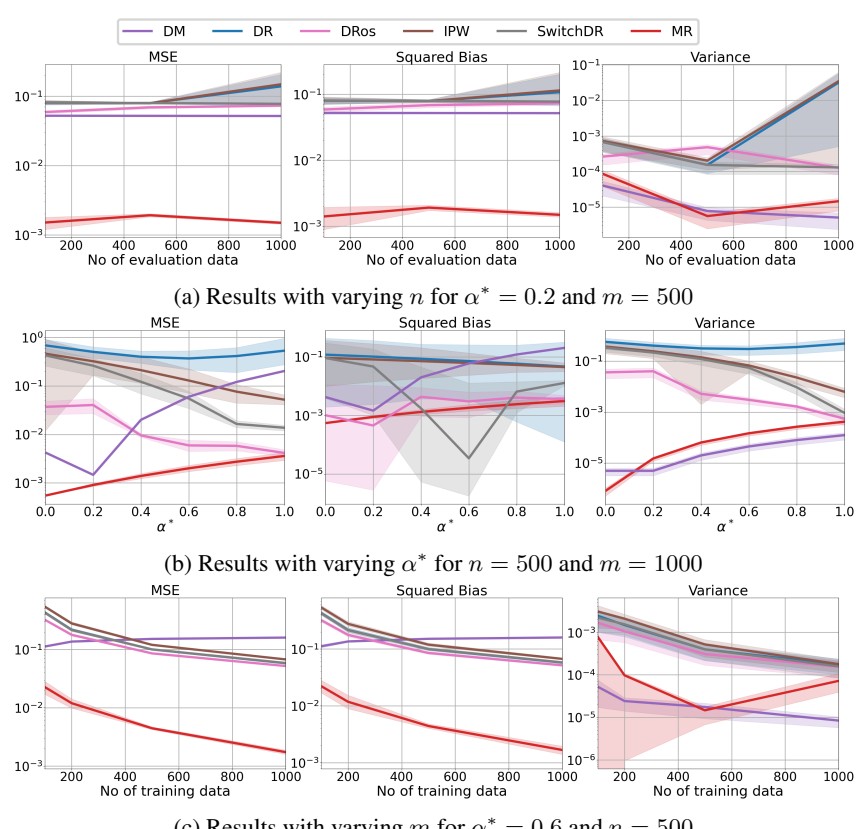

(a) Results with varying $n$ for $\alpha^* = 0.2$ and $m = 500$

(b) Results with varying $\alpha^*$ for $n = 500$ and $m = 1000$

(c) Results with varying $m$ for $\alpha^* = 0.6$ and $n = 500$

Figure 16: Results for Digits dataset. Note that compared to other datasets we consider smaller maximum dataset sizes $m, n$ here as the total number of available datapoints was 1797.

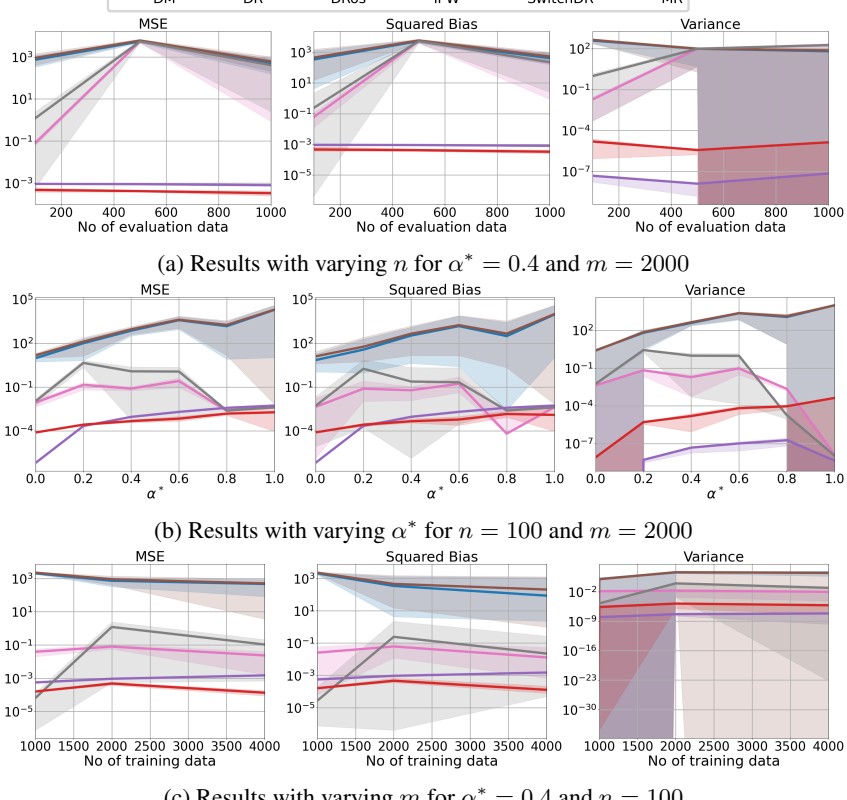

(a) Results with varying $n$ for $\alpha^* = 0.4$ and $m = 2000$

(b) Results with varying $\alpha^*$ for $n = 100$ and $m = 2000$

(c) Results with varying $m$ for $\alpha^* = 0.4$ and $n = 100$

Figure 17: Results for CIFAR-100 dataset.

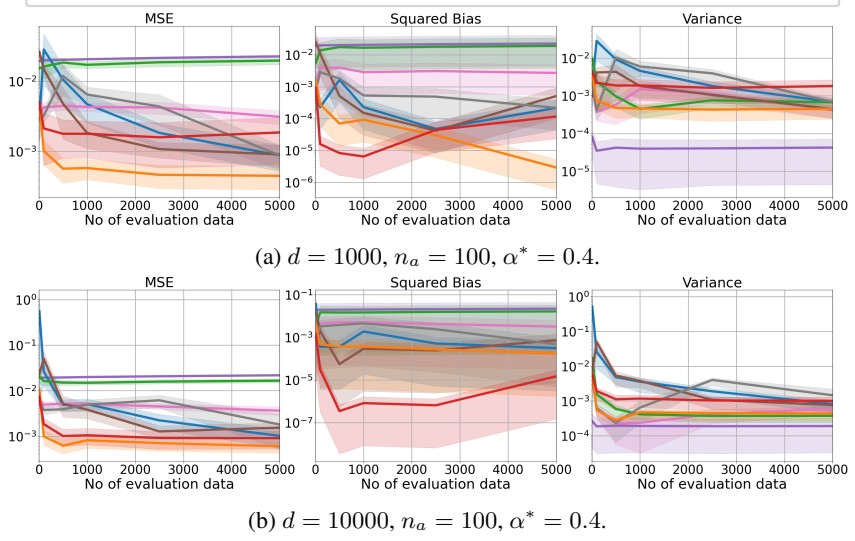

(a) $d = 1000$, $n_a = 100$, $\alpha^* = 0.4$.

(b) $d = 10000$, $n_a = 100$, $\alpha^* = 0.4$.

Figure 18: Results with varying size of evaluation dataset $n$.

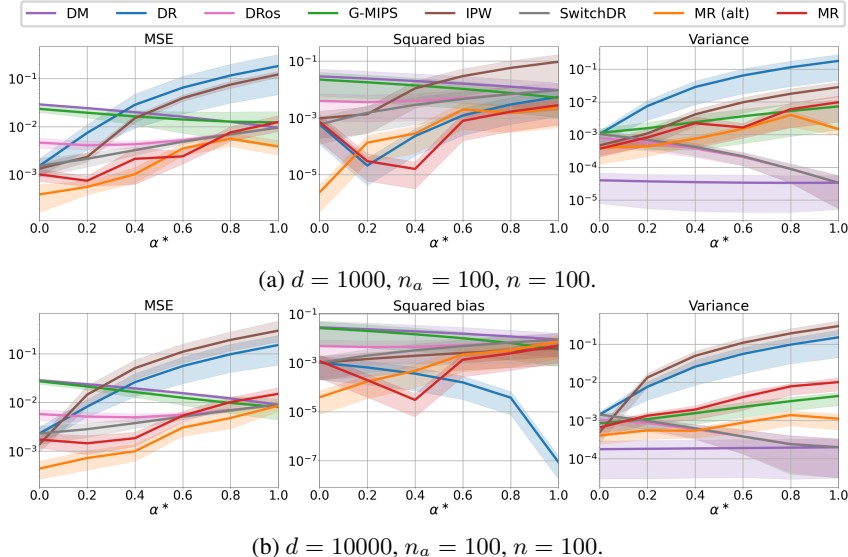

(a) $d = 1000$, $n_a = 100$, $n = 100$.

(b) $d = 10000$, $n_a = 100$, $n = 100$.

Figure 19: Results with varying $\alpha^*$.

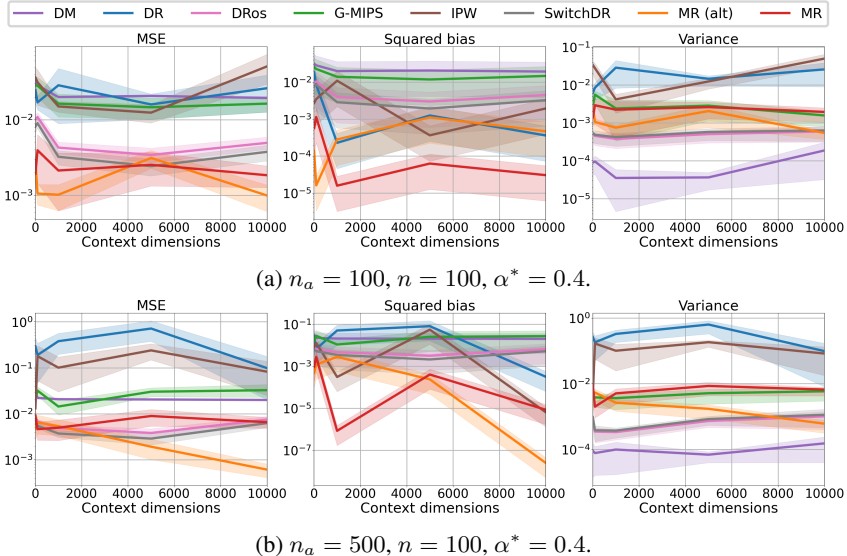

(a) $n_a = 100$, $n = 100$, $\alpha^* = 0.4$.

(b) $n_a = 500$, $n = 100$, $\alpha^* = 0.4$.

Figure 20: Results with varying context dimensions $d$.

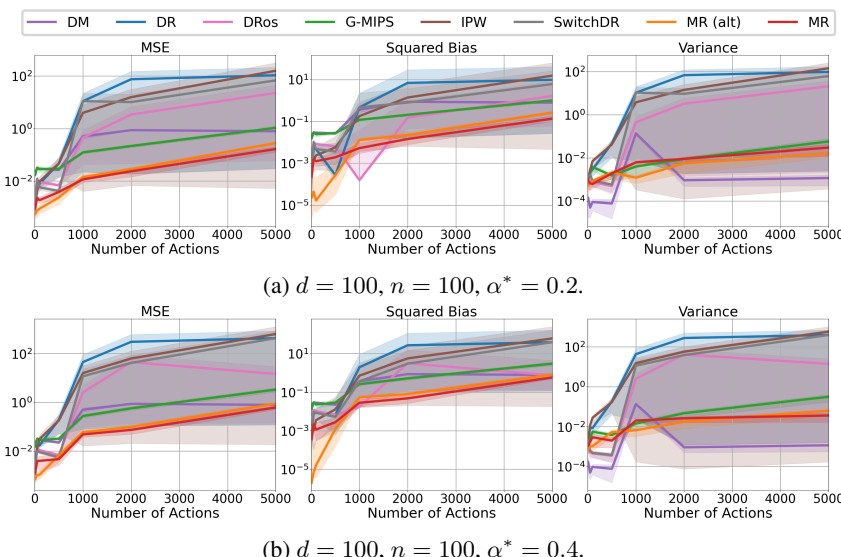

(a) $d = 100$, $n = 100$, $\alpha^* = 0.2$.

(b) $d = 100$, $n = 100$, $\alpha^* = 0.4$.

Figure 21: Results with varying number of actions $n_a$.

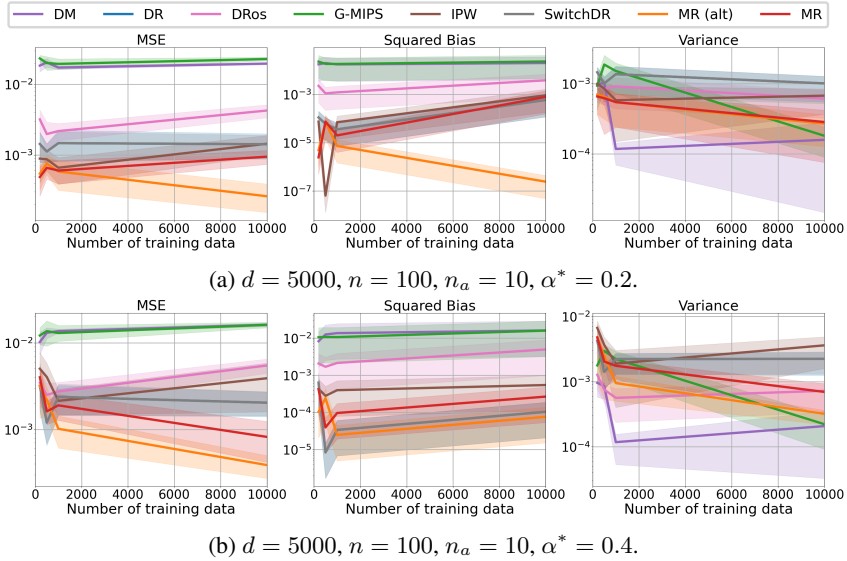

(a) $d = 5000$, $n = 100$, $n_a = 10$, $\alpha^* = 0.2$.

(b) $d = 5000$, $n = 100$, $n_a = 10$, $\alpha^* = 0.4$.

Figure 22: Results with varying number of training data $m$.

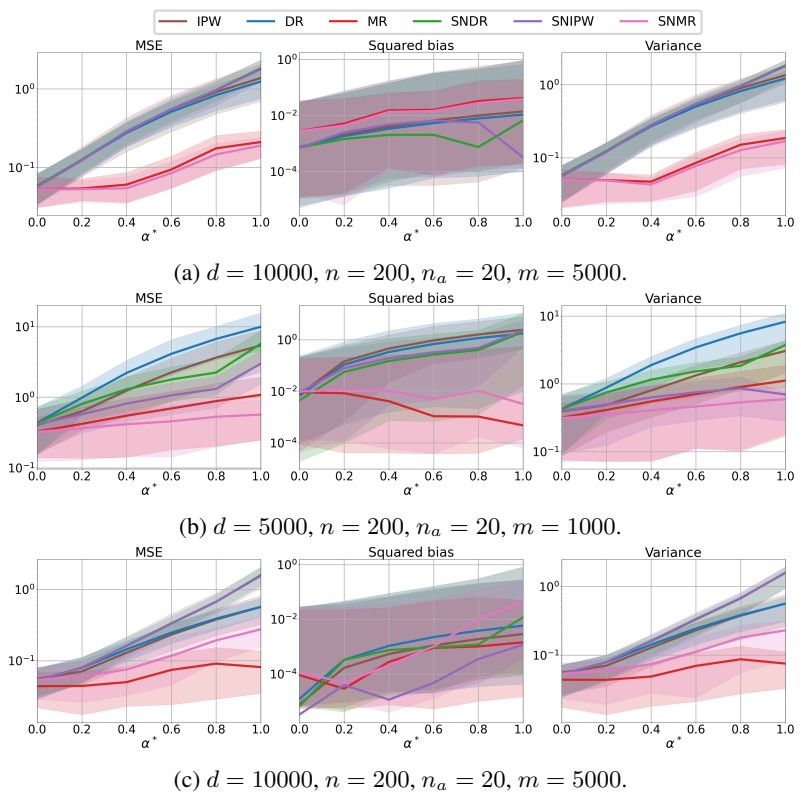

(a) $d = 10000$, $n = 200$, $n_a = 20$, $m = 5000$.

(b) $d = 5000$, $n = 200$, $n_a = 20$, $m = 1000$.

(c) $d = 10000$, $n = 200$, $n_a = 20$, $m = 5000$.

Figure 23: Results for self-normalised estimators with varying target policy shift $\alpha^*$ for synthetic data setup considered in Section 5.1. Here, "SN" denotes self-normalised estimators.

