# OpenReview forum: "Marginal Density Ratio for Off-Policy Evaluation in Contextual Bandits"
_NeurIPS.cc/2023/Conference — NeurIPS 2023 poster_

### Official Review · Reviewer_jTTf · 2023-06-29

**Soundness:** 3 good
**Presentation:** 3 good
**Contribution:** 3 good
**Rating:** 5
**Confidence:** 4

**Summary:**

This paper addresses Off-Policy Evaluation (OPE) in contextual bandits, a significant problem in fields such as healthcare and personalized recommendation systems. OPE involves evaluating the performance of new policies using only existing data generated by a current policy, which can pose a challenge due to high variance in estimators, particularly in situations with large action or context spaces. The study proposes a new OPE estimator called the Marginal Ratio (MR) estimator, which mitigates this issue by considering the shift in the marginal distribution of the outcome resulting from the policy shift, rather than the policy shift itself. This approach makes the MR estimator more robust to increasing sizes of action and context spaces than existing methods like Inverse Probability Weighting (IPW) or Doubly Robust (DR).

**Strengths:**

- The paper tackles the practically relevant problem of off-policy evaluation


- The paper is well-written and it was easy to follow the main arguments. The intuition statements right after each theorem were also useful


- The paper proposes a new estimator called the MR estimator, which defines the weight in terms of the shift in the conditional reward distribution between the logging and target policies and is expected to reduce the variance compared to IPW, DR, and MIPS. In an ideal case, MR can also be unbiased.


- Advantages of the proposed method over a range of conventional methods such as IPW, DR, and MIPS are demonstrated in a comprehensive way


**Weaknesses:**

- Eq.(3) may be unstable in cases where the target and behavior policies greatly differ, and there are many actions where the variance of $\rho(x,a)$ becomes very large.


- It is intuitive that MR reduces the variance from IPS and MIPS, but it needs to estimate the weight in terms of the marginal reward distributions, which are always unknown. If we can estimate the reward distributions well, then we could simply rely on DM. Therefore, it would be necessary to formalize the advantages of MR over DM in the case of an unknown $w(y)$.


- The experiment design could be improved. As I understand it, in the experiments, both $\rho(x,a)$, $p(e|x,a)$, and $w(y)$ are estimated from the logged data. While it's true that $w(y)$ is always unknown, in many industry applications where we can control $\pi_b$, $\rho(x,a)$ of IPW&DR and $p(e|x,a)$ of MIPS are known. Therefore, it would be necessary to compare MR with estimated $\rho(x,a)$ and IPS, DR, and MIPS with their respective true weights.


- This is not critical, but there are a few concurrent work that aims to further improve MIPS so would be great to discuss these papers in a revised version:

    - Jie Peng, Hao Zou, Jiashuo Liu, Shaoming Li, Yibao Jiang, Jian Pei, Peng Cui. Offline Policy Evaluation in Large Action Spaces via Outcome-Oriented Action Grouping. WWW2023. https://dl.acm.org/doi/abs/10.1145/3543507.3583448

    - Yuta Saito, Qingyang Ren, Thorsten Joachims. Off-Policy Evaluation for Large Action Spaces via Conjunct Effect Modeling. https://arxiv.org/abs/2305.08062

**Questions:**

- Could you provide some additional experiment results comparing MR with an estimated weight $\hat{w}(y)$ and IPS, DR, and MIPS with the true respective weights?


- How does MR perform when $\rho(x,a)$ takes some extreme values, making solving Eq.(3) hard? Specifically, I would love to see experiment results, for example with $n =5,000, n_a = 5000, \alpha = 1.0$, which should produce the situation I am interested in. Figure 8 in the appendix is relevant, but $n$ is too small to observe large empirical $\rho(x,a)$.


- What would be the theoretical comparison between DM and MR? Why does MR perform much better than DM, even though both perform OPE by estimating some aspect of the reward distribution?


- Can we extend MR to a doubly-robust version? How would it look, and how would its variance compare to that of DR with $\rho(x,a)$?


**Limitations:**

The limitations of the work are stated in the final section and they look reasonable.

---

> ### Author Rebuttal · Authors · 2023-08-09
>
> We would like to thank the reviewer for their useful comments and acknowledging the practical relevance of our proposed method.
> We clarify some of the misunderstandings below.
>
> > Eq.(3) may be unstable in cases where the target and behavior policies greatly differ, and there are many actions where the variance of $\rho(a, x)$ becomes very large.
>
> We agree with the reviewer that the variance of $\rho(A, X)$ might increase for large policy shifts. However, since Eq. (3) is a simple scalar to scalar regression it is easy to optimise in practice. Our experimental results which include settings where variance of $\rho$ is large, show that even with 1000 to 2000 training datapoints we are able to estimate the weights $\hat{w}$ well. For example, Figures 10 and 20 in the App.
>
> Additionally, we emphasise that while an increase in the variance of $\rho(a, x)$ could lead to an increase in the variance of MR, it can also lead to a **comparatively larger** increase in the variance of IPW and DR (see Prop. 3.3 and 3.4). Recall the Eq. 4 in Prop. 3.7 of our paper (after minor modifications):
>
> $$
> Var_{\pi^b}[\tilde\theta_{IPW}] - Var_{\pi^b}[\tilde\theta_{MR}]= \frac{1}{n}E_{\pi^b}[Var_{\pi^b}[\hat{\rho}(A, X)\mid Y]Y^2] + \Delta.
> $$
>
> Intuitively $\Delta$ is an error term arising from the approximation error of weights $\hat{w}(Y)$ and $\Delta \rightarrow 0$ as $E_{\pi^b}[(\hat{w}(Y) - E_{\pi^b}[\hat{\rho}(A, X)\mid Y])^2] \rightarrow 0$.
> Hence as $Var_{\pi^b}[\hat{\rho}(A, X)\mid Y]$ increases, so will the difference $Var_{\pi^b}[\tilde\theta_{IPW}] - Var_{\pi^b}[\tilde{\theta}_{MR}]$. In other words, the variance of the IPW estimator is likely to increase relative to that of MR as the variance of $\hat{\rho}(A, X)$ increases. Our results confirm this claim. In all our experiments, using both synthetic and real-world data, we observed that the variance of the MR estimator increases with the policy shift. However, the increase is significantly smaller compared to other baselines, particularly IPW and DR methods. See Figures 2b, 6, and 18.
>
> > If we can estimate the reward distributions well, then we could simply rely on DM. [...] Why does MR perform much better than DM, even though both perform OPE by estimating some aspect of the reward distribution?
>
> We emphasise that MR does not rely on estimating the target reward distribution directly (i.e. $p_{\pi^\ast}(y)$ or $p(y\mid x, a)$).  Instead, the MR estimator involves estimating the ratio $w(y) = p_{\pi^\ast}(y)/p_{\pi^b}(y)$ which, as shown in Lemma 3.1 can be estimated by solving a simple scalar to scalar regression:
> $$
> w = \arg\min_{f} E_{\pi^b}[(\hat{\rho}(A, X) - f(Y))^2].
> $$
> In contrast, DM involves estimating the expected conditional reward $\mu(a, x)\coloneqq E[Y\mid X=x, A=a]$, which is obtained by regressing from $(X, A)$ to $Y$. Intuitively, this can be a significantly more challenging regression in practice when the $(X, A)$ space is high-dimensional, as compared to regressing from scalars to scalars (as in MR). Therefore DM might suffer from higher bias.
>
> When the action space is discrete, estimating $\mu(a, x)$ requires using only datapoints with $A=a$. This is limiting, especially with many actions and few observed instances of some actions. Unlike this, MR doesn't need such data partitioning, making better use of training data. Section 5 and Appendix F show MR outperforms DM in MSE and bias.
>
> > The experiment design could be improved [...] Could you provide some additional experiment results comparing MR with an estimated weight $\hat{w}(y)$ and IPS, DR, and MIPS with the true respective weights?
>
> Our setting of unknown importance policy ratios $\rho(a, x)$ captures a wide variety of real-world applications, ranging form health-care to autonomous driving.
> In addition, to demonstrate the utility of MR in settings with **known** $\rho(a, x)$ and $p(e\mid x, a)$ and **unknown** $w(y)$ (for our proposed method, MR), we have conducted additional experiments. Here, we use a fixed budget of datapoints (denoted by $N$) for each baseline and for MR we allocate $m=2000$ of the available datapoints to estimate $\hat{w}(y)$ and used the remaining for evaluating the MR estimator (i.e., $n=N - 2000$ for MR). In contrast, for IPW and MIPS (since the importance ratios are already known), we use all of the $N$ datapoints to evaluate the off-policy value (i.e. $n=N$ for IPW and MIPS).
>
> The results included in Table 1 in our rebuttal show that MR achieves the smallest MSE among the baselines for $N\leq 6400$. However, we observe that the MSE of IPW, DR and MIPS (with true importance weights) falls below that of MR (with estimated weights $\hat{w}$) when the data size $N$ is large enough (i.e., $N\geq 10000$). This is to be expected since IPW, DR and MIPS are unbiased (i.e., use ground truth importance ratios $\rho(a, x)$) whereas MR uses estimated weights $\hat{w}(y)$ (and hence may be biased). MR still performs the best when $N\leq 6400$.
>
> > I would love to see experiment results, for example with $n=5,000, n_a=5000, \alpha=1.0$ [...]
>
> We have included the requested results in Figure 1 of our rebuttal document. The figure shows that MR achieves the smallest MSE and bias among all the baselines considered even when $n_a = 5000$.
>
> > Can we extend MR to a doubly-robust version?
>
> We thank the reviewer for this suggestion. We in fact discuss this in the Appendix D.2 (line 764), where we point out that the natural idea used to derive DR extensions of IPW does not work in the case of MR. However, this does not completely rule out the existence of **some** doubly-robust extension of MR. This is an interesting direction, which we leave for future work.
>
> We hope that the above has addressed the questions raised by the reviewer adequately, and that the reviewer will consider raising their score.

---

> > ### Comment · Reviewer_jTTf · 2023-08-11
> >
> > I appreciate the authors' clarifications. Most of my main concerns were addressed nicely. I still have minor points as below, but I can increase my score to 5 to indicate that at least I am no longer on the negative side.
> >
> > > However, since Eq. (3) is a simple scalar to scalar regression it is easy to optimise in practice.
> >
> > It is hard to believe that every scalar to scalar regression is easy. I do consider the empirical analysis of the paper that demonstrates the effectiveness of MR with the estimated weights in a range of situations to be convincing, but since the estimation of $w(y)$ is such a crucial step of MR, it might be possible to further improve the paper by, for example, showing the accuracy of this regression problem, including the case where $\rho(a,x)$ has high variance, in the empirical analysis.
> >
> > > In contrast, DM involves estimating the expected conditional reward $\mu(a,x)$, which is obtained by regressing from $(a,x)$ to $y$.
> >
> > This is true if there are only action index, but in practice, there are often many useful action features and it is typical to use them when performing OPL or OPE. This should only increase the accuracy of DM or DR via improving the reward regression accuracy, and thus I am not still very optimistic that the superior empirical performance of MR can be generalized to this practically typical setup without further analysis.
> >
> > > Our setting of unknown importance policy ratios captures a wide variety of real-world applications, ranging form health-care to autonomous driving.
> >
> > This is totally true, but my argument was that there are also many situations especially in the industry that we know the true importance weights for IPS and DR. Thus, the comparison of MR with the estimated weights and IPS/DR/etc. with the true weights should also be the content in the main text, but it was nice to see the authors' additional efforts to perform empirical analysis about this situation.

---

> > > ### Author Response · Authors · 2023-08-14
> > > **Thank you!**
> > >
> > > Firstly, we are deeply thankful to the reviewer for increasing their score.
> > >
> > > >it might be possible to further improve the paper by, for example, showing the accuracy of this regression problem [...]
> > >
> > > We take the reviewer's suggestion onboard and will include additional investigation into the accuracy of the regression problem with increasing variance of policy ratios $\rho(a, x)$.
> > >
> > > >There are often many useful action features and it is typical to use them when performing OPL or OPE. This should only increase the accuracy of DM or DR via improving the reward regression accuracy [...]
> > >
> > > We agree that in settings where the action features are highly predictive of the outcome $Y$, the reward model $\mu(a, x)$ may be easy to learn and DM may perform well. However, we would like to note that in many settings, such as healthcare (where the outcome $Y$ is often noisy), this may not be the case leading to a high bias in DM due to model misspecification [Saito et al., 2020; Farajtabar et al., 2018]. We will include a more detailed discussion on this in our paper.
> > >
> > > Lastly, the suggestion to incorporate a comparison of MR with estimated weights and IPS, DR and MIPS with true weights in the main text is duly noted, and we will include the empirical results for this setting in the revised version of our paper.
> > >
> > > Once again, we appreciate reviewer's thoughtful feedback and are grateful for their reconsideration of our paper's score.
> > >
> > > Yuta Saito, Aihara Shunsuke, Matsutani Megumi, and Narita Yusuke. Open bandit dataset and pipeline: Towards realistic and reproducible off-policy evaluation. arXiv preprint 474 arXiv:2008.07146, 2020.
> > >
> > > Mehrdad Farajtabar, Yinlam Chow, and Mohammad Ghavamzadeh. More robust doubly robust off-policy evaluation. In International Conference on Machine Learning, volume 80, pages 1447–1456. PMLR, 2018.

---

### Official Review · Reviewer_KWAf · 2023-07-02

**Soundness:** 3 good
**Presentation:** 3 good
**Contribution:** 3 good
**Rating:** 7
**Confidence:** 4

**Summary:**

This paper proposes a marginal OPE estimator that directly corrects the distribution shift wrt rewards instead of correcting the distribution shift wrt actions. The proposed estimator generalizes the idea of marginal IPS (MIPS) and achieves the minimum variance among them when the marginal importance weight is accurately estimated. Experiments on both synthetic and classification-to-bandit datasets demonstrate that the proposed estimator enables more accurate OPE than baseline estimators.

**Strengths:**

1. The idea of applying importance sampling on rewards rather than actions is reasonable from the context. The paper is easy to follow, and the contribution is clearly stated.

2. The paper also provides a data-driven method to estimate the marginal importance weight, which is a simple and straightforward approach.

3. The connection between MIPS, which estimate the expected outcome by applying marginal importance weight wrt action embeddings is discussed, and the theory suggests that the proposed MR achieve minimum variance among marginal estimators.

**Weaknesses:**

1. As author(s) discuss in the limitation, one potential concern lies in the accuracy of the marginal importance weight. In my understanding, the proposed method seems to work well when the reward is discrete, as an adequate amount of data is observed for each reward. However, when the reward is continuous, I guess sometimes only one importance weight is observed for each reward, which leads to the variance in the estimation of the marginal importance weight. This may lead to a similar difficulty as [Kallus&Zhou, 18]. I have several questions regarding this concern, which I list in the following “Questions” section.

Nathan Kallus and Angela Zhou. “Policy Evaluation and Optimization with Continuous Treatments” (AISTATS’18)

2. The marginal importance weight on reward has not been discussed in the standard OPE in contextual bandits, but some similar ideas have been discussed in OPE of RL [Rowland et al., 20] and another OPE framework which estimates the cumulative distribution function of a policy [Xu et al., 22] [Wu et al., 23]. As discussions with these methods are important to clarify the novelty of this work, I suggest discussing connections between these papers in related work.

Mark Rowland, Anna Harutyunyan, Hado van Hasselt, Diana Borsa, Tom Schaul, Rémi Munos, and Will Dabney. “Conditional Importance Sampling for Off-Policy Learning” (ICML’20)

Yang Xu, Chengchun Shi, Shikai Luo, Lan Wang, and Rui Song. “Quantile Off-Policy Evaluation via Deep Conditional Generative Learning” (‘22)

Runzhe Wu, Masatoshi Uehara, and Wen Sun. “Distributional Offline Policy Evaluation with Predictive Error Guarantee” (ICML’23)

**Questions:**

The following questions are related to my concern discussed in Weakness 1.

1. In which condition the marginal importance weight is accurately estimated?

2. Could you provide an error analysis on the estimation of the marginal importance weight (especially when the reward is continuous)?

3. How the marginal importance weight is parametrized and estimated? Are there any assumptions or regularization? How difficult the hyperparameter tuning is?

**Limitations:**

As discussed in Weakness 1, it would be valuable to discuss the potential difficulty in estimating the marginal importance weight when the reward is continuous.

---

> ### Author Rebuttal · Authors · 2023-08-09
>
> We thank the reviewer for their time reviewing our paper and for appreciating the clarity of our paper. Below we clarify some of the questions raised:
>
> > As author(s) discuss in the limitation, one potential concern lies in the accuracy of the marginal importance weight. In my understanding, the proposed method seems to work well when the reward is discrete, as an adequate amount of data is observed for each reward. However, when the reward is continuous, I guess sometimes only one importance weight is observed for each reward, which leads to the variance in the estimation of the marginal importance weight. This may lead to a similar difficulty as [Kallus\&Zhou, 18]...
> In which condition the marginal importance weight is accurately estimated?
>
> When the reward $Y$ is continuous, the estimation of our marginal ratios $\hat{w}(y)$ can be obtained by solving a simple 1d regression problem given a policy ratio estimate $\hat{\rho}$, as shown in the Eqn 3 in our paper:
> $$
> w = \arg\min_f E_{\pi^b}[(\hat{\rho}(A, X) - f(Y))^2].
> $$
> This is a straightforward regression problem since both, the inputs and outputs of $f$ are scalars. Moreover, this also means that given enough training data and an expressive enough model class (e.g. neural networks), we will be able to accurately estimate the marginal ratios $\hat{w}$.
> In our experiments (Section 5 in main paper and Appendix F), we consider two synthetic data setups where the reward is continuous. Figures 10 and 21 in the Appendix show that in both setups, even a moderately sized training dataset (roughly 1000 datapoints) is enough to estimate the marginal ratios well enough that the MR estimator outperforms all the baselines considered.
>
> In comparison, the setup considered in [Kallus \& Zhou, 18] is different as they consider continuous valued actions (in their setting, the reward may or may not be continuous). Consequently, the challenges posed in their setting are orthogonal to our setting because their methodology involves estimating the density of the continuous valued actions under observational data, which may be a difficult task in general.
> Our methodology in contrast only involves a simple scalar to scalar regression which is much easier to estimate in general.
>
> > Could you provide an error analysis on the estimation of the marginal importance weight (especially when the reward is continuous)?
>
> Section 3.1.2 and Appendix C in our paper provides theoretical analyses regarding weight estimation errors which hold when the reward is continuous. Specifically, let $\epsilon \coloneqq \hat{w}(Y) - E_{\pi^b}[\hat{\rho}(A, X)\mid Y]$ denote the approximation error of weights $\hat{w}(y)$.  In Appendix C, we use some recent results regarding the generalization of 2-layer wide neural networks [Lai et al., 2023] to show that, when using wide neural networks to estimate the weights $\hat{w}(y)$, we have that
> $$E_{\pi^b}[\epsilon^2] \leq \mathcal{O}(m^{-2/3})$$
> holds with a high probability, where $m$ is the number of training data.
>
> In addition, Proposition 3.7 shows how the variance and bias of the MR estimator depends on the weight approximation error $\epsilon$. Specifically, the result shows that as $E_{\pi^b}[\epsilon^2]\rightarrow 0$, MR achieves a lower variance than IPW, while incurring little extra bias. This result is general as it is not specific to any model class and does not require any strong assumptions.
>
> > How the marginal importance weight is parametrized and estimated? Are there any assumptions or regularization? How difficult the hyperparameter tuning is?
>
> Throughout our experiments, whenever the outcome $Y$ was continuous, we used a fully connected neural network with three hidden layers with 512, 256 and 32 nodes respectively (and ReLU activation function) to estimate the weights $\hat{w}(y)$. Moreover, we used sklearn library with the default learning rate of 0.001, and $l_2$ regularization coefficient of 0.01 to perform the regression. We found that the same architecture and hyperparameters worked well across all of our experimental settings with continuous rewards, and therefore did not need to perform hyperparameter tuning separately for each setting or dataset. While we mention the model architecture details in Appendix F, we will also add the hyperparameter details in the updated version of our paper.
>
> > The marginal importance weight on reward has not been discussed in the standard OPE in contextual bandits, but some similar ideas have been discussed in OPE of RL [Rowland et al., 20] and another OPE framework which estimates the cumulative distribution function of a policy [Xu et al., 22] [Wu et al., 23]. As discussions with these methods are important to clarify the novelty of this work, I suggest discussing connections between these papers in related work.
>
> We thank the reviewer for the relevant literature. We will add a detailed discussion regarding these works in the updated version of our paper.
>
> We hope the above addressed the concerns raised by the reviewer and that the reviewer will consider increasing their score.
>
> Jianfa Lai, Manyun Xu, Rui Chen, and Qian Lin. Generalization ability of wide neural networks on $\mathbb{R}$, 2023. URL https://arxiv.org/abs/2302.05933.

---

> > ### Comment · Reviewer_KWAf · 2023-08-14
> >
> > Thank you for the detailed answers to the questions, and the clarifications on the experimental settings and results resolved some of my concerns.
> >
> > However, let me clarify my intention in the comment below.
> >
> > > **Q.** when the reward is continuous, I guess sometimes only one importance weight is observed for each reward, which leads to the variance in the estimation of the marginal importance weight. This may lead to a similar difficulty as [Kallus&Zhou, 18].
> >
> > > **A.** In comparison, the setup considered in [Kallus & Zhou, 18] is different as they consider continuous valued actions (in their setting, the reward may or may not be continuous). Consequently, the challenges posed in their setting are orthogonal to our setting because their methodology involves estimating the density of the continuous valued actions under observational data, which may be a difficult task in general. Our methodology in contrast only involves a simple scalar to scalar regression which is much easier to estimate in general.
> >
> > I do understand that [Kallus & Zhou, 18] consider a different setting where the action space is continuous. What I meant here is that,  [Kallus & Zhou, 18] needed to incorporate the nearby actions to deal with the variance issue of IPS caused by sparsely observed actions in the action range of $(-\infty, \infty)$ even when the action space is 1-dimensional. In your setting, a similar situation can happen, because it also considers importance sampling on the 1-dimensional reward space in the range of $(-\infty, \infty)$. For instance, how about the case where the reward function is given as E[r|x, a] = a and Var(r) = 0? In this case, the true reward propensity is equivalent to the action propensity, and thus similar issues to [Kallus & Zhou, 18] can happen with the proposed method.

---

> > > ### Author Response · Authors · 2023-08-16
> > > **Thank you for your question!**
> > >
> > > We thank the reviewer for their insightful questions and continued engagement. Below we address the reviewer's comments by elucidating the similarities and differences between our setup and that of [Kallus & Zhou, 18].
> > >
> > > Both, [Kallus & Zhou, 18] and our work address the problem of off-policy evaluation using importance sampling methodologies. The importance ratios in [Kallus & Zhou, 18] are the policy ratios $\rho(a, x) = \pi^*(a\mid x)/\pi^b(a\mid x)$ whereas the importance ratios in our setting is the ratio of marginal distributions  $w(y) = p_{\pi^*}(y)/p_{\pi^b}(y)$.
> > >
> > > [Kallus & Zhou, 18] consider a setting with continuous action space (e.g. Medicine doses) and a target policy $\pi^*(a\mid x)$ which is Dirac delta at a specific action value $t(x)$ (i.e. $\pi^*(a\mid x) = \delta_{t(x)}(a)$). Since the action space is continuous, we have that $p_{\pi^b}(A = t(X)) = 0$ and we can not use the traditional IPW estimator shown below (as it will almost surely be 0):
> > > $$
> > > \hat\theta_{IPW}= 1/n\sum_{i=1}^n \frac{\mathbb{1}(a_i = t(x_i))}{\pi^b(a_i \mid x_i)} Y_i.
> > > $$
> > > Instead, the authors propose relaxing the unit mass of the Dirac delta function using a kernel function $K(u)$. To summarise, the main challenge being addressed in this work is the estimation of $\rho(a, x)$ in the case when action space is continuous and target policy is deterministic (conditioned on $X$). The authors address this problem by replacing the dirac delta target policy $\pi^*$ with the smoothed out kernel density estimation.
> > >
> > > In contrast, for MR estimator, given a policy ratio $\rho(a, x)$, we can estimate the marginal ratios $w(y)$ using the regression shown in Eq 3 when the rewards are continuous. In other words, compared to IPW, the MR method only involves the additional step of estimating the marginal ratios $w(y)$, which can be estimated directly using a simple regression and does not involve estimating the marginal distributions $p_{\pi^*}(y)$ or $p_{\pi^b}(y)$ separately.  In practice, when we use neural networks (for example) to perform this regression, this imposes an implicit restriction of smoothness on the computed ratios $\hat{w}(y)$. Therefore, if a specific value of reward $y'$ is unobserved in observational data, our trained regression model will infer the value of the ratio $\hat{w}(y')$ by interpolation.
> > >
> > > >how about the case where the reward function is given as E[r|x, a] = a and Var(r) = 0?
> > >
> > > We assume that the reviewer means that $Var_{\pi^*}(r) = 0$, (i.e. the target reward distribution is Dirac delta). Given the two conditions above, it can be shown that $Y = A = a'$ almost surely under the target policy $\pi^*$ for some fixed value $a'$, and therefore the target policy is also Dirac delta $\pi^*(a\mid x) = \delta_{a'}(a)$.
> > >
> > > Hence, in the example given above, the setting is the same as the one considered in [Kallus & Zhou, 18], i.e. the action space is continuous and the target policy is Dirac delta. This means that estimating the policy ratios $\rho(a, x)$ will involve the same challenge as considered in [Kallus & Zhou, 18]. However, we emphasise that this problem does not arise because of continuous reward space, but is more generally present whenever we are dealing with Dirac delta target policies and continuous action spaces, and will also be present if the reward space was discrete in this example. Moreover, this challenge is not specific to the MR estimator, and will also be present in any methodology which uses the policy ratios $\rho(a, x)$ (such as IPW, DR, MIPS, etc).
> > >
> > > In comparison, specifically for the MR estimator, if we were **given an estimate of the behaviour policy $\rho(a, x)$** (obtained using the methodology in [Kallus & Zhou, 18], for example) then we can simply estimate $\hat{w}(y)$ using a regression, and the rest of our methodology remains the same. In other words, MR does not impose any significant additional complexity compared to IPW as it only involves an additional scalar to scalar regression. We do not need to explicitly incorporate any smoothing methodologies when estimating $\hat{w}$, as the regression implicitly performs this smoothing in practice.
> > >
> > > We hope that the above addresses the reviewer's question adequately.

---

> > > > ### Comment · Reviewer_KWAf · 2023-08-18
> > > >
> > > > I appreciate the authors' additional clarification. Now I have resolved my questions and concerns, and given that the authors have promised to enhance the related work, I am happy to raise my score to 7. The idea of applying importance sampling directly to the reward would be worth sharing with the community.

---

### Official Review · Reviewer_LKfa · 2023-07-03

**Soundness:** 3 good
**Presentation:** 3 good
**Contribution:** 3 good
**Rating:** 7
**Confidence:** 4

**Summary:**

This paper introduced a new off-policy evaluation (OPE) method called Marginal Ratio (MR) to contextual bandits problems. Conventional methods such as Inverse Probability Weighting (IPW) and Doubly Robust (DR) rely on estimating policy ratios, which could incur large variances when there is low overlap between the target policy and the behavior policy.

Variance and bias analyses of the MR estimator are provided. Specifically, the MR estimator achieves smaller variance compared to both IPW and DR when the important weights $\rho$ and marginal ratio $\omega$ are known. When both of them are unknown, the authors proposed a method that can approximate the weights and showed that with high probability the MR estimator is more favorable than the IPW counterpart.

Extensive numerical experiments are provided, where the MR estimator outperformed baselines in terms of the mean square losses while maintained small variances.

**Strengths:**

The paper is overall well-written and easy to follow. The theorems, propositions and comparisons are well-organized. The authors provided a good motivation to study the MR estimator, which addressed the high variance issues when using IPW and DR in practice. The idea is novel and interesting and the simulation results showed great potentials of the proposed estimator.

**Weaknesses:**

The assumptions of proposition 3.3 and 3.4 are quite strong. Though the MR estimator achieves lower variances comparing to IPW and DR estimators, I think the assumption of knowing the ratio $\omega$ is much stronger than knowing the importance weight $\rho$. Thus, it is not a very fair comparison.

When the behavior policy and the marginal ratio $\omega$ are known, the authors showed that as the training data size $m$ increases, the biases of MR and IPW estimators are almost the same with high probability and the variance of MR estimator could be smaller than the IPW counterpart. I don't think this is a strong result as the MR estimator depends on the unknown importance ratio $\rho$ and the order term $\mathcal{O}(m^{-1/3})$ in the variance analysis could be very large in practice.

Overall, I think the theoretical results the MR estimator are sound yet incremental compared to existing methods.

**Questions:**

1. Can the order in the term $\mathcal{O}(m^{-1/3})$ be improved, e.g.,  $\mathcal{O}(m^{-1/2})$?

2. I am a bit confused by the connections to MIPS and the application to causal inference. As you mentioned, MIPS is trying to solve large action space problem while in causal inference the action space is usually binary. What is the benefits of using the MR estimator? Is it more applicable to large action space problems or small action space problems? Also, the connection and comparison to MIPS is not very straightforward to me as the final analyses do not explicitly depend on the action space size. Can you provide more explanations here?

**Limitations:**

N.A.

---

> ### Author Rebuttal · Authors · 2023-08-09
>
> We thank the reviewer for acknowledging the novelty of our idea and that our ``simulation results show great potential'' of our method.  Below we clarify some of the questions raised by the reviewer:
>
> > [...] I think the assumption of knowing the ratio $w(y)$ is much stronger than knowing the importance weight $\rho(a, x)$. Thus, it is not a very fair comparison.
>
> While our results in Propostions 3.3 and 3.4 assume that the importance weights $w(y)$ are known, we also include theoretical results in the case where the weights $w(y)$ are not known in Section 3.1.2 and Appendix C. Specifically, Proposition 3.7 intuitively shows that if our weights $\hat{w}$ are estimated `well enough', then the variance of the MR estimator will be smaller than that of the IPW estimator while incurring little extra bias. We note that this result also holds for the case where $\rho(a, x)$ is known exactly, whereas $w(y)$ is not. Our empirical results also corroborate this finding, as in all of our experiments the marginal ratio $\hat{w}(y)$ are estimated (and not exact), and we observe that overall MR consistently achieves a lower variance than most of the baselines considered.
>
> Furthermore, we conducted additional experiments with **known** policy ratios $\rho(a, x)$ and $p(e\mid x, a)$ (for IPW and MIPS estimators), and **estimated** marginal ratios $\hat{w}(y)$ (for MR estimator). For a fair comparison, we use a fixed budget of datapoints (denoted by $N$) for each baseline and in the case of MR we allocate $m=2000$ of the available datapoints to estimate $\hat{w}(y)$ and the rest of data to evaluate the MR estimator (i.e. $n = N-2000$ for MR). In contrast, for IPW and MIPS since the importance ratios are already known, we use all of the $N$ datapoints for evaluation of the off-policy value (i.e. $n=N$ for IPW and MIPS).
>
> The results in Table 1 of our rebuttal document show that MR achieves the smallest MSE among the baselines for $N\leq 6400$. However, we observe that the MSE of IPW, DR and MIPS (with true importance weights) falls below that of MR (with estimated weights $\hat{w}$) when the data size $N$ is large enough (i.e., $N\geq 10000$). This is to be expected since in this setting, the IPW, DR and MIPS estimators are unbiased (i.e., use ground truth importance ratios) whereas MR uses estimated weights (and hence may not be unbiased). MR still performs the best when $N\leq 6400$.
>
> > When the behavior policy and the marginal ratio are known, [...] as the training data size increases, the biases of MR and IPW estimators are almost the same [...]. I don't think this is a strong result [...].
>
> We thank the reviewer for their comment. We assume that the reviewer meant ``When the behavior policy and the marginal ratio
>  are **unknown**, [...]''.
>  We would like to note that the main purpose of Theorem C.1 (which provides the $\mathcal{O}(m^{-1/3})$ order term), is to show that when estimating the marginal ratios $\hat{w}(y)$, there exists one model class (i.e. neural networks) for which at least asymptotically in the number of training data, the MR estimator will achieve a lower variance than that of the IPW estimator, while incurring no additional bias. As we mention in Section 3.1.2, this theorem uses recent results regarding the generalization of 2-layer wide neural networks [Lai et al., 2023], and therefore may not capture the convergence rate for general models.
>  In contrast, Proposition 3.7 which expresses the variance of MR in terms of the weight approximation error $\epsilon$ is a general result which is not specific to a model class, and does not rely on any regularity assumptions (unlike Theorem C.1).
>  We will further clarify this in the updated version of the paper.
>
> > Can the order in the term $\mathcal{O}(m^{-1/3})$ be improved, e.g., $\mathcal{O}(m^{-1/2})$?
>
> Our result uses Theorem 4.1 from [Lai et al., 2023] which theoretically investigates the generalization of wide neural networks with 1d inputs and outputs, as we mention in Section 3.1.2. This theorem provides the convergence rate of $\mathcal{O}(m^{-1/3})$.
>  Improving this convergence rate is not straightforward as it will require improving upon the results provided in [Lai et al., 2023]. This is nonetheless an interesting direction to explore for future work.
>
> > [...] What is the benefits of using the MR estimator? Is it more applicable to large action space problems or small action space problems? Also, [...] the final analyses do not explicitly depend on the action space size. Can you provide more explanations here?
>
> We would like to emphasise that the performance of MR is not directly dependent on the size of the action space, but rather on the variance of policy ratios $\rho(a, x)$.
>  Specifically, our theoretical results in Propositions 3.3, 3.4 and 3.7 show that MR performs especially well relative to the IPW and DR estimators in cases where the policy ratios $\rho(a, x)$ have high variance. This high variance in $\rho(a, x)$ can stem from many reasons such as large action spaces [Saito \& Joachims, 2022] as well as large policy shifts (which can often occur in the setting of causal inference when estimating ATE). This is also supported by our experimental results where we observe that when either the policy shift increases (e.g. Figure 2b) or the size of action space increases (e.g. Figure 8), the variance and MSE of all baselines increases but this increase is smallest for the MR estimator (among all the baselines considered).
>
> We hope that we were able to address all the reviewer's questions in the above and hope that the reviewer would consider increasing their score.
>
>  Jianfa Lai, Manyun Xu, Rui Chen, and Qian Lin. Generalization ability of wide neural networks on $\mathbb{R}$, 2023. URL https://arxiv.org/abs/2302.05933.
>
>  Yuta Saito and Thorsten Joachims. Off-policy evaluation for large action spaces via embeddings. In Proceedings of the 39th International Conference on Machine Learning, pages 19089–19122. PMLR, 2022.

---

> > ### Comment · Reviewer_LKfa · 2023-08-18
> >
> > Thanks for the response. The authors addressed my concerns and I raised the score.

---

### Official Review · Reviewer_ceqD · 2023-07-05

**Soundness:** 3 good
**Presentation:** 4 excellent
**Contribution:** 3 good
**Rating:** 7
**Confidence:** 3

**Summary:**

The paper looks at the problem of off-policy evaluation (OPE) in contextual bandits.
The problem consists of estimating the expected reward obtained from a policy different from the one that collected the data.
The authors first analyze classic OPE estimators used for this problem, such as Inverse Probability Weighting (IPW) and Doubly Robust (DR). The authors notice how such estimators have high variance (especially IPW), and this is due to their focus on policy shift. The authors propose a novel estimator, called Marginal Ratio (MR) estimator, which focuses on the shift of the marginal distribution of the rewards instead. The authors also theoretically show that, whenever the marginal ratio is known exactly, MR has lower or equal variance compared to IPW. The authors also derive a relationship between the variances of MR and DR. In this case, however, they show that it is not true that the variance of MR is always lower or equal to the one of DR. Furthermore, the authors show that MR has lower variance than Marginalized IPS (MIPS, a recently proposed estimator), under the same assumption used by MIPS.

**Strengths:**

- Simple and interesting idea with a novel point of view
- Clear presentation
- Careful experimental analysis (also considering the appendix) and strong empirical performance


**Weaknesses:**

- Since the problem with IPW and DR is high variance, a really simple way to reduce the variance for such estimators is the Self-Normalization trick [1]. I would have preferred to see the empirical comparison also against the Self-Normalized variant of IPW and DR.
- There is no investigation on whether the Self-Normalization trick is applicable to the proposed MR. This could further improve the performance of the proposed estimator (if applicable)
- The experimental setting seems to be such that IPW has high variance, which is fine since it is a known characteristic of IPW. However, there are also settings where IPW performs well (see [2] for an in-depth review of the performance of state-of-the-art OPE estimators). It could have been useful to see a comparison between the baseline estimators and MR in such settings, even only in the appendix.


[1] A. Swaminathan and T. Joachims, The self-normalized estimator for counterfactual learning, NeurIPS 2015.

[2] Saito et al., Evaluating the Robustness of Off-Policy Evaluation, RecSys 2021.

**Questions:**

Taken from the above section:

- Do you think that the Self-Normalization trick is applicable to MR?

**Limitations:**

The authors have addressed the limitations of their work.

---

> ### Author Rebuttal · Authors · 2023-08-09
>
> We thank the reviewer for appreciating the novel point of view presented in our paper and our careful experimental analysis. Below we address some of the questions raised:
>
> > Since the problem with IPW and DR is high variance, a really simple way to reduce the variance for such estimators is the Self-Normalization trick [1]. I would have preferred to see the empirical comparison also against the Self-Normalized variant of IPW and DR.
>
> This is an interesting suggestion and indeed, self-normatlization trick is applicable to the MR estimator. Specifically, the self-normalized MR estimator (denoted as $\hat{\theta}_{\text{SNMR}}$) can be written as follows:
>
> $$
>  \hat{\theta}\_{SNMR} \coloneqq\sum_{i=1}^n \frac{\hat{w}(Y_i)}{\sum_{j=1}^n \hat{w}(Y_j)}Y_i.
> $$
>
> We conducted experiments to investigate the effect of self-normalisation on the performance of the IPW, DR and MR estimators as suggested. Figure 2 in our rebuttal show results for three different choices of parameter configurations. Overall, we observe that in all settings, the MR and self-normalised MR (SNMR) estimator outperform all other baselines including the self-normalised IPW and DR estimators (denoted as SNIPW and SNDR respectively).
> Moreover, in some settings, where the importance ratios $\rho(a, x)$ achieve very high values, self-normalisation can reduce the variance and MSE of the corresponding estimator (for example, Figure 2b). However, we also observe cases in which self-normalization does not significantly change the results (Figure 2a), or may even slightly worsen the MSE of the estimators (Figure 2c). We thank the reviewer again for their insightful suggestion and will add these experiments in the updated version of our paper. It is worth noting that our investigation of self-normalization only considers the synthetic data setup, and we will include a more thorough investigation in the updated version of the paper.
>
> > The experimental setting seems to be such that IPW has high variance, which is fine since it is a known characteristic of IPW. However, there are also settings where IPW performs well (see [2] for an in-depth review of the performance of state-of-the-art OPE estimators). It could have been useful to see a comparison between the baseline estimators and MR in such settings, even only in the appendix.
>
> We thank the reviewer for their suggestion and in the updated version we will include the settings considered in [Saito et al., 2021] where IPW performs relatively well. It is worth mentioning here that we conducted additional experiments with **known** policy ratios $\rho(a, x)$ and $p(e\mid x, a)$ (for IPW and MIPS estimators), and **estimated** marginal ratios $\hat{w}(y)$ (for MR estimator).
> For a fair comparison, we use a fixed budget of datapoints (denoted by $N$) for each baseline and in the case of MR we allocate $m=2000$ of the available datapoints to estimate $\hat{w}(y)$ and the rest of data to evaluate the MR estimator (i.e. $n = N-2000$ for MR). In contrast, for IPW and MIPS since the importance ratios are already known, we use all of the $N$ datapoints for evaluation of the off-policy value (i.e. $n=N$ for IPW and MIPS).
>
> The results in Table 1 of our rebuttal document show that MR achieves the smallest MSE among the baselines for $N\leq 6400$. However, we observe that the MSE of IPW, DR and MIPS (with true importance weights) falls below that of MR (with estimated weights $\hat{w}$) when the data size $N$ is large enough (i.e., $N\geq 10000$). This is to be expected since in this setting, the IPW, DR and MIPS estimators are unbiased (i.e., use ground truth importance ratios) whereas MR uses estimated weights (and hence may not be unbiased). MR, despite being at a disadvantage in this setting, still performs the best when $N\leq 6400$.
>
> We hope the above addressed the questions raised sufficiently.

---

> > ### Comment · Reviewer_ceqD · 2023-08-18
> >
> > Thank you for your detailed responses and clarifications. I believe that the additional experiments mentioned in your comment will greatly benefit the paper. I remain convinced that this paper is deserving of acceptance, and thus, I will maintain my score of 7.

---

### Official Review · Reviewer_2CGe · 2023-07-06

**Soundness:** 4 excellent
**Presentation:** 3 good
**Contribution:** 3 good
**Rating:** 7
**Confidence:** 5

**Summary:**

This paper addresses the problem of off-policy evaluation (OPE) using a variant of the inverse propensity score (IPS) estimator on logged bandit data. The authors specifically aim to reduce the variance of IPS by focusing on the shift in the marginal distribution of rewards instead of the policies themselves, resulting in a new estimator called the Marginal Ratio (MR) estimator. The proposed estimator is extensively analyzed, including comparisons with other related works such as IPS, DR, and MIPS. Furthermore, the authors demonstrate the utility of the MR estimator in causal inference for estimating average treatment effects (ATE). The effectiveness of the MR estimator in OPE is evaluated through a standard experimental setup, involving both synthetic and semi-synthetic data.

# POST-REBUTTAL:
I have read the author’s rebuttal and the authors addressed all my concerns.

**Strengths:**

The clarity and quality of writing in this paper are solid. The authors not only address the standard contextual bandit setting but also extend their work to causal inference for estimating average treatment effects (ATE). The overall idea of considering the shift in the marginal distribution of outcomes instead of the policies themselves is simple and very interesting. The theoretical concepts are effectively explained, and the proposed estimator is comprehensively compared to standard estimators such as IPS, DR, and MIPS. To the best of my knowledge, the derived guarantees are original contributions. The authors employ a standard experimental setup to validate their claims and showcase the favorable performance of their estimator through an extensive set of experiments.

**Weaknesses:**

**Computational Efficiency:**

The proposed MR estimator introduces an additional step of estimating $w(y)$, which may pose computational challenges compared to IPS, where the importance weights $\pi/\pi_0$ are readily accessible. While this might be manageable in the context of off-policy evaluation, where a policy $\pi$ is given and the goal is to estimate its value using $\hat{\theta}$ through IPS, DR, MIPS, or MR, it may become problematic in off-policy learning. In off-policy learning, we optimize $\hat{\theta} = \hat{\theta}(\pi)$ with respect to $\pi$ to find the policy that maximizes the reward. Then, performing a regression to estimate $w(y)$ in each optimization step with respect to $\pi$ can be computationally demanding. This presents a notable challenge.

**Experiments:**

The experimental setup presented in the paper is solid. However, it would be beneficial to incorporate more complex computer vision datasets such as Fashion-MNIST and CIFAR-100, as seen in recent studies (e.g., [1, 2]). These datasets feature larger action sets and contexts when converted to contextual bandit instances.

**Related Work:**

While not a weakness, it would be valuable to include citations for two recent papers [1, 2] that propose new corrections for the importance weights to reduce variance. In [1], the authors suggest clipping only the propensity scores $\pi_0$ (as $\rm{max}(\pi_0, 0)$) instead of the importance weight $\pi/\pi_0$. Additionally, [2] introduces a smooth regularization of the importance weights by incorporating $\pi/\pi_0^\alpha$, where $\alpha \in [0, 1]$. While it is not necessary to directly compare the proposed approach with these papers, acknowledging their existence and contributions would be appropriate.

[1] Ben London and Ted Sandler. Bayesian counterfactual risk minimization. In International Conference on Machine Learning, pp. 4125–4133. PMLR, 2019. Preprint version arXiv:1806.11500.

[2] Imad Aouali, Victor-Emmanuel Brunel, David Rohde, and Anna Korba. Exponential Smoothing for Off-Policy Learning. arXiv preprint arXiv:2305.15877, 2023.

**Questions:**

- In terms of computational efficiency, how does the efficiency of the MR estimator compare to IPS?
- Is there an efficient method to optimize the MR estimator with respect to the policy $\pi$ (i.e., in the context of off-policy learning) considering that estimating $w(y)$ is required in each iteration during the optimization process?

**Limitations:**

The authors adequately addressed limitations in Section 6.

---

> ### Author Rebuttal · Authors · 2023-08-09
>
> We thank the reviewer for their thoughtful comments on our contributions and soundness of our approach. Below we respond to the questions raised:
>
> > In terms of computational efficiency, how does the efficiency of the MR estimator compare to IPS?
>
> In the setting of off-policy evaluation, computing the MR estimator involves an additional step of estimating the weights $\hat{w}(y)$ (compared to the IPW estimator). When the outcome $Y$ is continuous, the estimation of weights $\hat{w}(y)$ can be achieved by solving a simple regression over functions $f:\mathbb{R}\rightarrow \mathbb{R}$ as we show in Lemma 3.1:
> $$
> w = \arg\min_{f} E_{\pi^b}[(\hat{\rho}(A, X) - f(Y))^2].
> $$
> This is a simple scalar to scalar regression, which does not impose any significant computational overhead, and in practice $\hat{w}(Y)$ is well estimated given a moderately sized training data (around 1000 to 2000 datapoints). See Figure 10 in the appendix, for example. In the case when $Y$ is discrete, this estimation becomes even more straightforward as we can directly use the formulation $w(y) = E_{\pi^b} [\rho(A, X)\mid Y=y]$ and estimate $w(y)$ by computing the empirical mean of $\hat{\rho}(A, X)$ over datapoints where $Y=y$.
>
> > The proposed MR estimator introduces an additional step of estimating $w(y)$, which may pose computational challenges compared to IPS, where the importance weights $\pi/\pi_0$ are readily accessible. While this might be manageable in the context of off-policy evaluation, where a policy $\pi$ is given and the goal is to estimate its value using $\hat{\theta}$ through IPS, DR, MIPS, or MR, it may become problematic in off-policy learning. In off-policy learning, we optimize $\hat{\theta} = \hat{\theta}(\pi)$ with respect to $\pi$ to find the policy that maximizes the reward. Then, performing a regression to estimate
> $w(y)$ in each optimization step with respect to $\pi$ can be computationally demanding. This presents a notable challenge.
> Is there an efficient method to optimize the MR estimator with respect to the policy $\pi$ (i.e., in the context of off-policy learning) considering that estimating $w(y)$ is required in each iteration during the optimization process?
>
> We emphasise that this paper mainly addresses the problem of off-policy evaluation. We agree that directly applying our methodology to off-policy learning will indeed incur additional computational overhead since we need to run additional optimisation to update our ratios $\hat{w}(y)$ after every policy update. However, one possible way to avoid this computational overhead is to use ideas from transfer learning literature [Lu et al., 2021]. Specifically, Section 5.1 of [Lu et al., 2021] proposes a methodology which can be used to combine the two optimisation problems into one, and hence could be used to avoid running regression separately after each optimisation step. We reiterate, however, that our paper is specific to off-policy evaluation and extending this to off-policy learning is an interesting avenue for future research.
>
> > The experimental setup presented in the paper is solid. However, it would be beneficial to incorporate more complex computer vision datasets such as Fashion-MNIST and CIFAR-100, as seen in recent studies (e.g., [1, 2]). These datasets feature larger action sets and contexts when converted to contextual bandit instances.
>
> We thank the reviewer for their suggestion. However, like the fashion-MNIST dataset, MNIST dataset included in our experiments also comprises 10 labels and $28 \times 28$ images, and therefore the size of the action and context spaces is the same for both datasets. Additionally, our synthetic data setup also includes experiments with significantly larger actions spaces (of sizes up to 5000). We will consider adding additional experiments on the image data like CIFAR-100 as suggested in the updated version of our paper.
>
> > While not a weakness, it would be valuable to include citations for two recent papers [1, 2] that propose new corrections for the importance weights to reduce variance.
>
> Thank you for this suggestion, we will cite these works in the updated version of our paper.
>
> We hope our clarifications have addressed the reviewer's concerns, and kindly ask them to consider increasing their score.
>
> Nan Lu, Tianyi Zhang, Tongtong Fang, Takeshi Teshima, Masashi Sugiyama. Rethinking Importance Weighting for Transfer Learning, 2021.

---

> > ### Comment · Reviewer_2CGe · 2023-08-11
> > **Thank you!**
> >
> > I appreciate your comprehensive response. My concerns have been addressed. I recommend incorporating a detailed discussion that explores the challenge of applying MR in off-policy learning as it is an important task related to off-policy evaluation, alongside a comparison with the missing related works. Additionally, the inclusion of the suggested experiments and the new CIFAR-100 trials would significantly enrich the paper. __Given your responses and assuming that the paper will be revised accordingly, I am pleased to raise my score to 7.__

---

> > > ### Author Response · Authors · 2023-08-14
> > > **Thank you for the feedback!**
> > >
> > > We are deeply grateful to the reviewer for providing positive feedback and for increasing their score. We also appreciate the useful suggestions which will greatly benefit our work and will make the suggested changes to our paper.

---

> > > > ### Comment · Reviewer_2CGe · 2023-08-16
> > > >
> > > > I appreciate the authors' response and would like to inquire about an additional matter. Upon reviewing the references, I observed that paper [26] is listed in the references but not explicitly mentioned in the paper itself. After examining its content, I found it to be both significant and relevant to the topic. Hence, I recommend engaging in a comprehensive discussion that highlights the distinctions between your work and [26]. Could you kindly explain the differences between your work and [26]?
> > > >
> > > > [26] Nathan Kallus and Masatoshi Uehara. Double reinforcement learning for efficient off-policy evaluation in Markov decision processes. J. Mach. Learn. Res., 21(1), June 2022. ISSN 1532-4435.
> > > >
> > > > Furthermore, I came across the following relevant survey that might be of interest:
> > > >
> > > > Uehara, Masatoshi, Chengchun Shi, and Nathan Kallus. "A review of off-policy evaluation in reinforcement learning." arXiv preprint arXiv:2212.06355 (2022).

---

> > > > > ### Author Response · Authors · 2023-08-17
> > > > >
> > > > > We thank the reviewer for their question.
> > > > > > Could you kindly explain the differences between your work and [26]?
> > > > >
> > > > > [26] considers the problem of off-policy evaluation in RL setting. Specifically, the authors derive efficient off-policy estimators separately for Markov Decision Process (MDP) and non-Markov Decision Process (NMDP) settings. The OPE estimator proposed in MDP setting utilizes marginalised importance weights however, the main difference between our methodology and that of [26] is that MR uses ratios of marginal densities of rewards $Y$ whereas [26] utilises the ratios of marginal densities of state-action pair $(s_t, a_t)$ at time $t$. As we explain next, this choice of marginal densities of $(s_t, a_t)$ is specific to MDP settings and helps alleviate the problem of 'curse of horizon', a problem specific to reinforcement learning. In a contextual bandits setup (i.e. single decision setting), the ratio of marginal densities of $(s_t, a_t)$ in fact reduces to the policy ratios.
> > > > >
> > > > > In an MDP, the data comprises of trajectories $\tau = (s_0, a_0, r_0, s_1, a_1, r_1, \dots, s_t, a_t, r_t)$, where for the sake of simplicity we assume that $a_i \mid s_i \sim \pi^b(\cdot\mid s_i )$. Given a target policy $\pi^*$ (which we assume is stationary for the sake of simplicity again), the traditional IPW estimator considers the ratios of joint distributions of the whole trajectories under target and behaviour policies, which reduces to the product of policy ratios:
> > > > > $$\frac{p_{\pi^*}(\tau)}{p_{\pi^b}(\tau)} = \prod_{i=0}^t \frac{\pi^*(a_i \mid s_i) }{\pi^b(a_i \mid s_i) }$$
> > > > >
> > > > > However, in an MDP, all relevant information regarding $r_t$ in the past trajectory is captured by the state-action pair $(s_t, a_t).$
> > > > > Technically, this property can be expressed as
> > > > > $$r_t \perp (s_0, a_0, r_0, s_1, a_1, r_1, \dots, s_{t-1}, a_{t-1}, r_{t-1}) \mid (s_t, a_t)$$
> > > > >
> > > > > Therefore, when estimating $E_{\pi^*}[r_t]$, instead of weighting the rewards by the joint distribution over the entire trajectory (as in traditional IPW estimator), the methodology in [26] only focuses on the shift in the marginal distribution of $(s_t, a_t)$, which leads to a more efficient off-policy estimator. The MR estimator would instead directly focus on the marginal shift in the rewards $r_t$.
> > > > >
> > > > > To summarise, we would like to reiterate that while [26] also considers marginalized density ratios, the importance ratios considered in [26] are different than the ones used in MR. The methodology considered in [26] is aimed more towards addressing the problem of OPE in RL settings where the trajectories might span a large horizon and the challenges faced are different than the ones in contextual bandits.
> > > > >
> > > > > We hope the above answers the reviewer's question. We will include a more detailed discussion regarding this comparison in our paper.
> > > > >
> > > > > > Furthermore, I came across the following relevant survey that might be of interest:
> > > > > Uehara, Masatoshi, Chengchun Shi, and Nathan Kallus. "A review of off-policy evaluation in reinforcement learning." arXiv preprint arXiv:2212.06355 (2022).
> > > > >
> > > > > We thank the reviewer for this reference and will incorporate this in the updated version of our paper.

---

> > > > > > ### Comment · Reviewer_2CGe · 2023-08-17
> > > > > >
> > > > > > I appreciate the authors for answering my question. I recommend including this discussion in either the main paper or the appendix.

---

### Author Rebuttal · Authors · 2023-08-09

Firstly, we would like to thank the reviewers for taking the time to review our paper, appreciating the quality of our work and providing many insightful comments regarding it. To address some of the concerns raised, we have conducted additional experiments and included the results in the attached PDF file. We hope that our rebuttal along with the additional results answer the reviewers' questions, and that the reviewers will consider increasing their scores.

---

### Decision · Program_Chairs · 2023-09-21

**Decision:**

Accept (poster)

**Comment:**

This paper explores the off-policy evaluation problem in bandits. In an effort to reduce the variance of existing off-policy evaluation (OPE) estimators, the authors propose a new estimator, the Marginal Ratio (MR) estimator, which focuses on shifts in the marginal distribution of outcomes. The authors offer robust analysis of the proposed method, demonstrating that the MR achieves lower variance than existing methodologies.

The reviewers unanimously recommend acceptance of this paper, a stance with which I concur. However, this endorsement is conditional on the authors incorporating all the changes they've promised during the rebuttal phase into the final version of the paper. These include a comparison with overlooked related works, additional experiments on image datasets like CIFAR-100, among others.